# GeoEvolve: Automating Geospatial Model Discovery via Multi-Agent Large Language Models

## Abstract

Geospatial modeling provides critical solutions for pressing global challenges such as sustainability and climate change. Existing large language model (LLM)–based algorithm discovery frameworks, such as AlphaEvolve, excel at generic code evolution but lack the domain knowledge required for complex geospatial problems. We introduce GeoEvolve, a multi-agent LLM framework that couples evolutionary search with dynamic geospatial domain knowledge. GeoEvolve operates in nested loops: an inner code evolver generates candidate solutions, while an outer agentic controller—supported by Automated Knowledge Construction and Code-to-Formula agents—queries a Dynamic GeoKnowRAG module to inject theoretical priors. This architecture addresses the challenges of spatial heterogeneity and temporal non-stationarity. We evaluate GeoEvolve on three classical tasks: spatial interpolation (Kriging), uncertainty quantification (GeoCP), and spatial regression (GWR). Across 9 datasets, GeoEvolve discovers novel algorithms that incorporate geospatial theory. It achieves significant gains, such as a 29.5% increase in regression $R^2$ and a 13–21% reduction in interpolation error. Furthermore, extensive ablation studies confirm GeoEvolve's robustness across diverse foundation models (GPT, Gemini, Qwen) and its spatiotemporal generalizability, validating that domain-guided retrieval is essential for stable evolution. Collectively, these results offer a scalable path toward trustworthy, automated geospatial modeling, opening new avenues for efficient AI-for-Science discovery.

## 1 Introduction

Beyond building powerful AI models that help us analyze data and understand the world, enabling AI models to evolve on their own and autonomously extract knowledge stands as the next important and promising frontier. It usually involves a prolonged procedure of asking a research question, gathering relevant information, analyzing it to identify patterns or insights, and communicating the results as new knowledge. The rise of the large language models (LLMs), such as GPT-4 (Achiam et al., 2023) and Gemini (Comanici et al., 2025), presents the possibility of accelerating and automating this knowledge discovery procedure. The confidence in this direction is supported by the breakthroughs in LLMs, such as retrieval augmented generation (RAG) that enhances the output of LLMs (Lewis et al., 2020; Jiang et al., 2023) and agents that execute complex tasks autonomously (Li et al., 2023; Qian et al., 2024). In fact, the integration of LLMs into this procedure has already boosted the performance of a range of discovery-oriented tasks, such as drug repurposing (Huang et al., 2024), hypothesis generation (Kumbhar et al., 2025; Xiong et al., 2024), chip design (Ho & Ren, 2024), urban planning (Zhou et al., 2024). Recently, Google introduced AlphaEvolve, which has demonstrated remarkable capabilities in automating algorithm discovery across diverse domains, such as tackling complex mathematical optimization problems. Building on this foundation, OpenEvolve has been developed as an open-source implementation of Google DeepMind's AlphaEvolve, providing the research community with accessible tools for further exploration and application.

Despite these advances, the domain of geospatial modeling remains relatively underexplored in the context of LLM-driven knowledge discovery. Geospatial problems are inherently complicated,

characterized by spatial autocorrelation (Miller, 2004), spatial heterogeneity (Cheng et al., 2024), scale effect (Chen et al., 2019), and diverse modalities (e.g., maps, remote sensing imagery, spatial network, and textual description) (Mai et al., 2023), etc. Moreover, addressing geospatial problems also demands synthesizing knowledge across different disciplines, from environmental science to urban studies, making it difficult for single-agent systems to provide comprehensive solutions.

In this paper, we introduce GeoEvolve, an advanced agent combining the evolutionary process with LLM-based code generation and geospatial knowledge-informed RAG (GeoKnowRAG) to automatically investigate optimal geospatial modeling. GeoEvolve operates in two complementary loops. As is shown in Figure 1, the inner loop runs OpenEvolve (Sharma, 2025) for a limited number of evolutionary steps, generating candidates of discovery. The outer loop is governed by an agentic controller, which evaluates the best solutions, retains global elites to prevent performance degradation, and invokes the GeoKnowRAG module. This module will query a structured geospatial knowledge database, thus producing refined, domain-informed prompts that guide the next round evolution. We show that GeoEvolve can obviously improve the geospatial modeling.

In summary, the contributions of our work are as follows:

1. **Knowledge-guided evolution.** We integrate evolutionary search with domain knowledge by coupling GeoEvolve's evolutionary code generation (via OpenEvolve) with retrieval-augmented geospatial knowledge. This grounds discovery in established geospatial theories and classical methods rather than random mutations, steering evolution toward theoretically meaningful and practically effective directions.

2. **Automated, scalable pipeline.** We develop an automated and scalable geospatial modeling pipeline that can continuously evolve, adapt, and refine geospatial algorithms, providing a robust methodology for diverse geospatial tasks.

3. **State-of-the-art performance and efficiency.** We demonstrate state-of-the-art performance on two spatial modeling cases—spatial interpolation and spatial uncertainty quantification—supported by an ablation study verifying the role of domain knowledge.

## 2 RELATED WORK

**LLM-driven Algorithm Discovery** Driven by LLMs, many studies aim to accelerate the discovery of algorithms with better performance, simpler implementation, and higher computational efficiency. A common approach is evolutionary search, which explores the algorithmic space via mutations and recombinations guided by performance metrics (Surina et al., 2025), enabling breakthroughs across diverse applications (Lu et al., 2024; Ma et al., 2024; Veličković et al., 2024; Morris et al., 2024). Among the most influential methods is FunSearch—searching in the function space—which fosters creative algorithmic solutions while guarding against confabulations (Romera-Paredes et al., 2024), but is limited to evolving a single function rather than an entire codebase. AlphaEvolve, a substantially enhanced successor, leverages LLMs to solve complex problems at scale (Novikov et al., 2025). Yet addressing specialized challenges, particularly in geospatial domains, requires domain-specific knowledge, multi-step reasoning, and iterative refinement guided by evaluation feedback (Chen et al., 2024).

**Retrieval-augmented generation RAG for scientific discovery.** RAG has emerged as a standard strategy to ground LLM outputs in external knowledge, improving factual accuracy and controllability (Lewis et al., 2020; Gao et al., 2023). Recent advances such as RAG-Fusion (Rackauckas, 2024) and reciprocal rank fusion (RRF) (Cormack et al., 2009) demonstrate that expanding and fusing multiple reformulated queries can substantially enhance retrieval coverage and downstream reasoning quality. Moreover, RAG has recently been applied in the geospatial domain to support knowledge discovery and contribute to downstream tasks such as spatial reasoning (Yu et al., 2025). However, to the best of our knowledge, no prior work has leveraged RAG to extract geospatial knowledge specifically for geospatial model construction, leaving an important gap for integrating structured geographic knowledge into model design.

**LLM-based Autonomous Agents** Recent advances in LLM-based autonomous agents have substantially expanded their capacity for solving complex tasks through multi-agent collaboration and

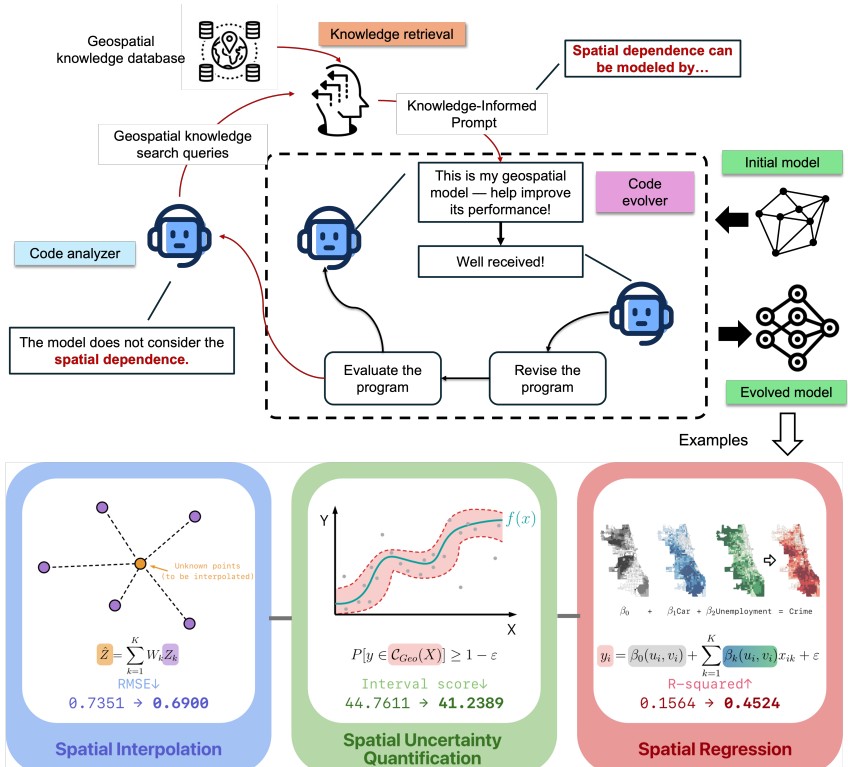

Figure 1: An illustration of the code-evolution trajectory of a geospatial model integrating domain knowledge. The dashed inner box represents the code evolver, a general algorithmic code-generation engine. The surrounding workflow depicts the knowledge-guided code generation proposed in this paper, specifically tailored for geospatial modeling.

structured role-playing (Wang et al., 2024). Frameworks such as MetaGPT (Hong et al., 2023) and ChatDev (Qian et al., 2024) emulate the Standard Operating Procedures of software companies, assigning roles such as product managers and engineers to automate large portions of the software development lifecycle. In the geography domain, LLM-based agents have also been introduced to automate geospatial modeling workflows—including data ingestion, processing, analysis, and visualization—greatly lowering the technical barrier for using domain-specific tools (Li & Ning, 2023). However, while these systems are highly effective at executing linear engineering workflows with well-defined requirements, they are generally not designed for scientific discovery, which requires open-ended objectives, evolving hypotheses, and exploration within large and uncertain search spaces.

## 3 GEOEVOLVE

GeoEvolve is designed to automate geospatial model discovery by integrating evolutionary code generation with structured geospatial knowledge. Unlike general-purpose code agents, GeoEvolve incorporates domain-specific knowledge from spatial modeling literature and classical algorithms, enabling the discovery of geospatial algorithms. Figure 2 illustrates the overall framework of Geo-Evolve. It consists of four main components: (1) a code evolver, (2) an evolved code analyzer, (3) a geospatial knowledge retriever, and (4) a geo-informed prompt generator. Together, these components orchestrate a closed-loop process of code generation, evaluation, and refinement, leading to the emergence of geospatial model discovery.

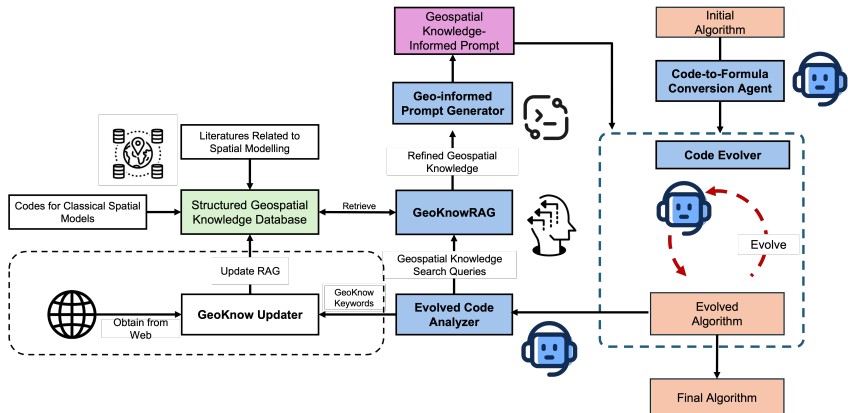

Figure 2: The workflow of GeoEvolve

### 3.1 CODE-TO-FORMULA AGENT

To streamline the transition from user-defined geospatial models to evolutionary search spaces, Geo-Evolve incorporates an Code-to-Formula agent. Instead of requiring users to manually configure the complex input specifications—comprising the initial program, evaluator logic, and instructional prompts—this agent employs an LLM-based semantic parser to automate the initialization process.

Guided by a set of pre-defined heuristic templates and few-shot exemplars derived from classical geospatial algorithms (e.g., Kriging, GeoCP), the agent analyzes the user's raw code to extract core algorithmic logic. It then encapsulates this logic into a standardized triplet format required by the evolutionary engine. This design effectively decouples the user's domain implementation from the framework's internal search protocols, allowing researchers to focus on model logic rather than configuration details. These standardized templates and illustrative exemplars are encapsulated as built-in assets within the GeoEvolve codebase, enabling an out-of-the-box experience for users.

### 3.2 CODE EVOLVER

The central engine of GeoEvolve is the code evolver, an evolutionary coding agent that generates and iteratively refines candidate algorithms. Beginning with an initial algorithm, the evolver performs a fully autonomous pipeline of mutation, evaluation, and selection relying on the power of LLMs. Candidate algorithms are represented as a group of executable code fragments. Mutations can be parameter changes, operator substitutions, or structure modifications to the algorithm. Abstractly, given a task-specific objective function $\mathcal{L}$, the evolver seeks to optimize an algorithm $A$ such that

$$A* = \arg \min_{A \in \mathcal{A}} \mathcal{L}(A; \mathcal{D}), \tag{1}$$

where $\mathcal{A}$ is the search space and $\mathcal{D}$ is the dataset. Here, we use OpenEvolve as the code evolver, which is the open-source equivalent of AlphaEvolve.

### 3.3 EVOLVED CODE ANALYZER

The evolved code analyzer is an LLM-powered diagnostic agent that interprets both the evolved code and associated metrics (e.g., RMSE for regression tasks). Its role is not limited to evaluating task outcomes, but also to providing semantic analysis of the code, thus identifying potential weaknesses or missing knowledge. To be specific, the LLM is required to achieve two tasks. First, it identifies missing or problematic knowledge from the evolved code. Second, it suggests search queries for retrieving useful geospatial knowledge from GeoKnowRAG. The diagnostic feedback given by this agent will be passed to the geospatial knowledge retriever to obtain related knowledge. This design allows GeoEvolve to reason about why the evolved algorithm fails and what kind of domain knowledge is needed to improve it. The template and an example of the code analyzer can be found at Figure 6.

## 3.4 GEOSPATIAL KNOWLEDGE RETRIEVER

To prevent the evolutionary search from drifting into non-meaningful algorithmic space, GeoEvolve incorporates domain-specific geospatial knowledge through a dedicated Geospatial Knowledge Retrieval module (GeoKnowRAG). We construct a structured knowledge base by collecting literature on core geospatial modeling concepts (e.g., spatial autocorrelation) and classical algorithms (e.g., geographically weighted regression) from Wikipedia, arXiv, and GitHub, using curated keywords (Figure 7, Appendix A.3.1). To ensure high-quality and comprehensive knowledge coverage, RAG-Fusion (Rackauckas, 2024) is applied to merge results from multiple reformulated queries, enabling the system to capture both precise theoretical matches and semantically related concepts. Geo-KnowRAG transforms these diverse resources into a structured RAG system that delivers domain-aware prompts directly to the code evolver, providing the theoretical grounding and classical geospatial methods required for effective algorithmic refinement. As shown in Figure 3, GeoKnowRAG comprises four steps:

**Source Identification and Acquisition** Unlike previous approaches that rely on manually curated static topic lists, GeoEvolve employs a fully automated, agent-driven pipeline to construct and continuously evolve its knowledge base. This process operates in two phases: automated initialization and dynamic expansion.

First, to establish the foundational knowledge base, we introduce an Automated Knowledge Base Construction Agent. Upon receiving the user's baseline geospatial code, this agent performs semantic analysis to extract core algorithmic concepts and automatically identifies an initial set of search keywords (defaulting to 5 key terms). These keywords drive the initial retrieval from three complementary corpora—peer-reviewed papers (arXiv), encyclopedic entries (Wikipedia), and open-source repositories (GitHub)—downloading up to 150 documents to form a task-specific, normalized UTF-8 knowledge repository.

Second, to address theoretical gaps that emerge during evolution, we implement a Dynamic Knowledge Update loop. In each outer iteration, the Evolved Code Analyzer scrutinizes the evolved code and performance metrics. Acting as a diagnostic gatekeeper, it determines whether the current algorithmic bottleneck stems from a lack of domain knowledge. If a deficit is identified, the agent generates precise search queries and triggers the GeoKnow Updater, which fetches high-relevance literature from the web (capped at 5 new documents per cycle) to augment the database in real-time. Conversely, if the knowledge base is deemed sufficient, the system adaptively reverts to static retrieval to conserve resources. Finally, the Geo-informed Prompt Generator synthesizes the updated knowledge with the current code to steer the next evolutionary step.

**Text Chunking and Pre-processing** First, each document is semantically segmented into 300-word chunks with a 50-word overlap to preserve contextual continuity across chunk boundaries and improve downstream retrieval accuracy. Second, all PDF, Markdown, and HTML sources are stripped of formatting, de-duplicated, and tokenized into a clean corpus ready for embedding.

**Vectorization and Knowledge Indexing** First, every chunk is encoded using the `text-embedding-3-small` model from OpenAI to obtain high-dimensional semantic vectors. Second, these embeddings are stored in a **Chroma** vector database, which supports approximate nearest-neighbor search and metadata filtering by topic or source type. Third, this indexed database forms the persistent memory of GeoKnowRAG and enables millisecond-scale retrieval across the geospatial knowledge space.

**RAG-Fusion Query and Prompt Generation** First, GeoKnowRAG employs multi-angle question expansion, where each input query from the GeoEvolve controller is reformulated into several sub-questions emphasizing different semantic aspects such as theory, implementation, and evaluation. Second, each sub-question is independently embedded and used for vector search to retrieve top-$k$ relevant chunks from the Chroma index. Third, the retrieved results are re-ranked using RRF, which scores passages based on the reciprocal of their ranks across sub-queries so that consistently high-scoring chunks surface to the top. Fourth, the highest-ranked passages are aggregated and summarized into a geo-informed prompt encoding key formulas, algorithmic structures, and empir-

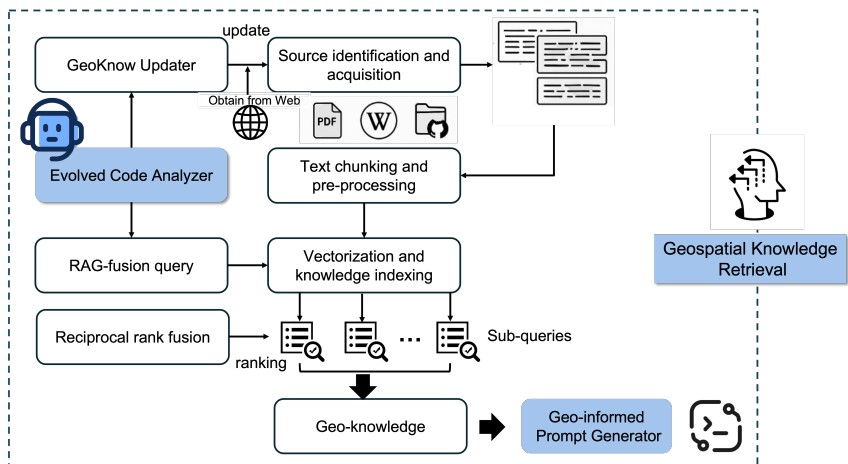

Figure 3: The workflow of GeoKnowRAG

ical heuristics, which is then supplied to the GeoEvolve code evolver to guide the next round of algorithmic mutation and evaluation.

### 3.5 GEO-INFORMED PROMPT GENERATOR

The information, from retrieved geospatial knowledge to evolved code, and associated metrics, is then processed together by the geo-informed prompt generator, which will translate it into a structured prompt for the code evolver. This prompt refines the search by introducing domain constraints, suggesting algorithmic structures, or incorporating empirical heuristics. The generator leverages LLMs as reasoning and translation engines, transforming abstract geospatial knowledge into actionable modifications of candidate code.

The LLMs are required to generate a prompt that includes four key elements. First, algorithmic fixes or improvements suggesting how the current algorithm could be revised. Second, new operators or parameters that may improve performance in subsequent evolutionary iterations. Third, geospatial knowledge, including the direction of exploration, theoretical or empirical conditions, and expected outputs. Fourth, maximum tokens control, which helps maintain efficiency and reduce hallucination.

## 4 EXPERIMENTS

To evaluate GeoEvolve's capability for improving and discovering geospatial models, we focus on three fundamental topics: spatial interpolation, uncertainty quantification, and spatial regression. We detail the first two in the main text and present the spatial regression results in Appendix. For each topic, we select the most representative and classical baseline model, and employ a GPT-4–based evolutionary engine as the core evolve agent to autonomously search, mutate, and refine candidate algorithms.

We use OpenEvolve as the primary baseline. In addition, we conduct an ablation study with two variants. First, OpenEvolve with GeoKnowledge Prompt, where domain knowledge is incorporated as additional prompts. The prompt template is: *"You are allowed to refer to advanced methods in the field of spatial interpolation and consider some important settings of spatial models, such as localized variogram, automatic variogram parameter selection, or stratified strategy, etc."* Second, GeoEvolve without GeoKnowledge, where the GeoKnowRAG module is removed. For every algorithm, after each evolutionary step the generated code is first analyzed by the code analyzer and then directly passed to the knowledge-prompt generator to create new prompts.

For the OpenEvolve-based algorithms, we perform ten iterations of evolutionary search. For the GeoEvolve algorithms, we run ten outer-loop cycles—each consisting of the code analyzer, Geo-KnowRAG, and geo-informed prompt generator—and within every outer cycle we conduct ten

inner-loop evolutions. This results in a total of one hundred evolutionary iterations. For every experiment, the dataset is split into training, validation, and test sets in an 8:1:1 ratio.

Crucially, we extend our evaluation to the unique challenge of geospatial generalizability, testing the framework across 9 datasets. Geospatial modeling demands robustness across three dimensions: Domain Generalizability (transferring logic between disparate fields, e.g., socioeconomic vs. environmental data), Spatial Generalizability (adapting to spatial heterogeneity across regions), and Temporal Generalizability (mitigating non-stationarity over time).

## 4.1 SPATIAL INTERPOLATION MODEL

**Task- Spatial interpolation** Spatial interpolation is one of the most important applications in geospatial analysis and a key approach for humans to observe the Earth's surface environment and understand the planet (Lam, 1983). Its task is to model discrete sample points collected across geographic space—such as climate observation stations, biodiversity observation points, or mineral sampling sites—and to predict the continuous spatial surface of the geographic variables of interest based on these observations.

**Model- Oridinary Kriging** We selected ordinary kriging, the most classical geostatistical spatial interpolation model, as the first case study for GeoEvolve to automatically improve and evaluate. Since its invention, many studies have attempted to extend kriging, for example by integrating regression models in regression kriging (Hengl et al., 2007) or by accounting for spatially stratified heterogeneity in stratified kriging (Luo et al., 2023). However, ordinary kriging remains the fundamental core of the entire kriging family and of geostatistics itself. Because it was developed long ago and has a relatively simple structure, direct algorithmic innovations to ordinary kriging have become increasingly rare. More details about ordinary kriging can be found at Appendix A.3.1.

If GeoEvolve can demonstrably enhance ordinary kriging, it would greatly revitalize geostatistical methods and provide fundamental improvements that can propagate to all kriging-based models and applications. This rationale underpins our choice of ordinary kriging as the first benchmark algorithm in this study.

**Evaluator** For the kriging interpolation task, we use the root mean squared error (RMSE) as the evaluation metric. Our objective is to obtain a kriging model that achieves a lower RMSE, indicating higher predictive accuracy.

**Datasets** In this study, we use trace-element observations of copper (Cu), lead (Pb), and zinc (Zn) collected from a representative region of Australia (with concentrations expressed in parts per million, ppm) to conduct spatial interpolation and geostatistical modeling experiments. These three heavy metals have important indicative significance in environmental geochemistry: on the one hand, they serve as key factors for assessing regional environmental pollution levels and soil heavy-metal accumulation. Details of the data acquisition and processing procedures can be found in (Luo et al., 2025).

### 4.1.1 EVOLVED ALGORITHM OF ORDINARY KRIGING.

GeoEvolve preserves the ordinary-kriging core but augments it with (i) an expanded variogram family (Exponential, Gaussian, Linear, and Matérn) with automatic model selection via AIC/BIC, capturing a wider range of spatial smoothness; (ii) an adaptive empirical variogram using quantile/Silverman binning, trimmed means, and an automatic $n_{\text{lags}} \in [8, 20] \propto \sqrt{n}$ to stabilize nugget/sill/range estimation; (iii) robust multi-start fitting with L1 or weighted least squares and bin-based weights to avoid local minima and keep parameters physically meaningful; (iv) localized kriging that solves a $K$-NN system with condition-number–aware diagonal adjustment, reducing complexity from $O(n^3)$ to $O(K^3)$ and improving numerical stability; and (v) an adaptive log transform with a data-driven offset to reduce skew and ensure valid back-transformation. Together, these changes retain unbiasedness and best-linear prediction while delivering lower RMSE/MAE, tighter residuals, and greater computational robustness across heterogeneous spatial settings. The detailed development of GeoEvolve–Kriging can be found at Appendix A.4.1.

Table 1: Performance comparison across different methods. For each metal, lower is better for RMSE/MAE, and higher is better for $R^2$.

| Method | Cu | | | Pb | | | Zn | | |
|---|---|---|---|---|---|---|---|---|---|
| | RMSE ↓ | MAE ↓ | $R^2$ ↑ | RMSE ↓ | MAE ↓ | $R^2$ ↑ | RMSE ↓ | MAE ↓ | $R^2$ ↑ |
| Original | 0.9139 | 0.6752 | 0.3751 | 0.6619 | 0.4580 | 0.3563 | 0.6294 | 0.4689 | 0.4304 |
| OpenEvolve (No GeoKnowledge) | 0.8727 | 0.6557 | **0.4302** | 0.6413 | 0.4441 | **0.3957** | 0.6245 | 0.4712 | 0.4395 |
| OpenEvolve (General GeoKnowledge) | 0.9264 | 0.6519 | 0.3755 | 0.6519 | 0.4598 | 0.3755 | 0.6332 | 0.4725 | 0.4235 |
| OpenEvolve (Specific GeoKnowledge) | 0.9139 | 0.6761 | 0.3752 | 0.6632 | 0.4579 | 0.3537 | 0.6337 | 0.4716 | 0.4227 |
| GeoEvolve (No RAG)) | 0.9139 | 0.7321 | 0.2889 | 0.6619 | 0.5871 | 0.0298 | 0.6294 | 0.5905 | 0.1723 |
| GeoEvolve (Static RAG)) | 0.8602 | 0.6423 | 0.3596 | 0.5927 | 0.4390 | 0.3025 | 0.5941 | 0.4475 | **0.4433** |
| GeoEvolve (Dynamic RAG) | **0.8718** | **0.6418** | 0.3721 | **0.6131** | **0.4299** | 0.3492 | **0.5852** | **0.4388** | 0.4363 |

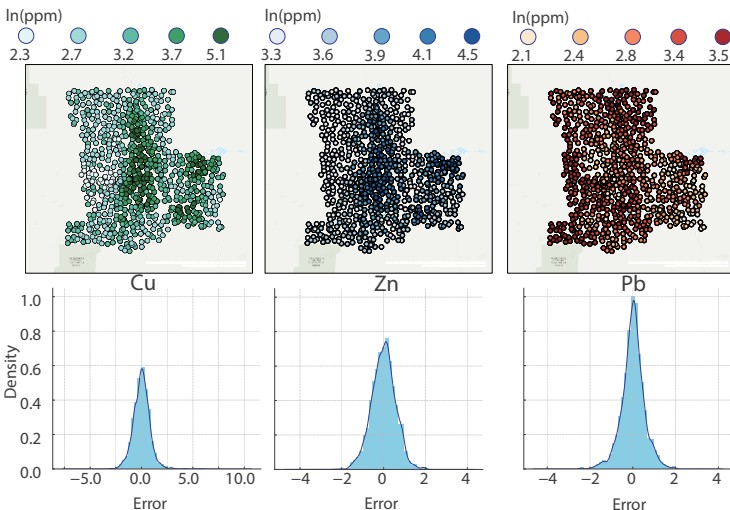

Figure 4: The spatial distribution of predicted concentrations and the error distribution of three elements, Cu, Zn, and Pb obtained from Evolved Kriging

### 4.1.2 MODEL EVALUATION

Table 1 reports the kriging accuracy obtained by different methods. GeoEvolve–kriging consistently achieves the lowest RMSE and MAE across the prediction of Cu, Pb, and Zn, while the original kriging baseline performs worst. Applying OpenEvolve to kriging improves the prediction of Cu and Pb but slightly degrades the performance on Zn. Introducing GeoKnowledge prompts into OpenEvolve does not lead to further gains, possibly because the injected knowledge lacks direct relevance to variogram estimation or spatial covariance structures that govern kriging performance. GeoEvolve without GeoKnowRAG already outperforms OpenEvolve, yet still falls short of the full GeoEvolve model, underscoring the critical role of structured geospatial domain knowledge in guiding algorithm evolution.

Compared with OpenEvolve–kriging, GeoEvolve–kriging reduces RMSE by 11.3%, 20.9%, and 13.5% on Cu, Pb, and Zn predictions, respectively. Relative to the original kriging, the reductions are 15.4%, 21.2%, and 13.0%, further highlighting GeoEvolve's ability to automatically discover and refine spatial interpolation algorithms with substantially improved predictive accuracy.

Figure 4 illustrates the spatial distributions of the predicted concentrations and the associated error maps for Cu, Pb, and Zn obtained by GeoEvolve–kriging, clearly demonstrating its capability to capture fine-scale spatial variability while maintaining low residual errors.

## 4.2 SPATIAL UNCERTAINTY QUANTIFICATION MODEL

**Task- Spatial UQ**  In spatial predictive modeling, it is not sufficient merely to develop more accurate models for point predictions; an equally critical task is to quantify and communicate the uncertainty of predictions, as this directly shapes the reliability and legitimacy of geography-based decisions such as flood evacuation planning and public facility site selection. Therefore, incorporating rigorous uncertainty quantification into spatial prediction is essential not only for improving scientific credibility, but also for supporting transparent, fair, and ethically sound spatial planning and policy making.

**Model- GeoCP**  In geography, the task of assessing the reliability of spatial prediction results is commonly addressed through uncertainty quantification (UQ). In this study, we adopt geospatial conformal prediction (GeoCP)—a model-agnostic algorithm for estimating the uncertainty of spatial prediction models—as the target method for enhancement using GeoEvolve (Lou et al., 2025b). More details about GeoCP can be found at Appendix A.3.2.

**Evaluator**  For GeoCP uncertainty estimation, we use the *interval score*

$$\text{IS}_i = \max(U_i - L_i, \epsilon) + \frac{2}{\alpha}\big[(L_i - y_i)\mathbb{I}(y_i < L_i) + (y_i - U_i)\mathbb{I}(y_i > U_i)\big], \tag{2}$$

where $L_i, U_i$ are prediction bounds, $y_i$ the observation, and $\alpha$ the significance level (e.g., 0.1 for 90% intervals). The first term measures interval width (with $\epsilon \approx 10^{-6}$ to avoid zero width), and the second penalizes coverage violations, scaled by $1/\alpha$. Smaller IS indicates tighter and better-calibrated intervals.

**Datasets**  The housing price dataset used in this study originates from the GeoDa Lab repository[1]. The original data include 21,613 residential transactions and 21 attributes from Seattle and King County, Washington (May 2014–May 2015). For our analysis, we focus on the Greater Seattle urban core and retain 11 key variables, with housing sale price (in \$10,000s) as the dependent variable. Eight non-spatial predictors capture structural and quality characteristics—bathrooms, living-space and lot size, grade, condition, waterfront proximity, view quality, and property age—while two spatial predictors are geographic coordinates expressed in UTM (universal transverse mercator). Further details of the dataset are documented in (Lou et al., 2025b;a).

### 4.2.1 EVOLVED ALGORITHM OF GEOCP

GeoEvolve–GeoCP preserves the fundamental conformal prediction framework of GeoCP while introducing two major methodological advances. First, it refines the geographic weighting scheme: still employing a Gaussian kernel, but re-optimizing the bandwidth parameter through multi-start global search with adaptive clipping to ensure numerical stability and faithfully capture local spatial heterogeneity. Second, it enhances the weighted quantile computation by unifying earlier adaptive strategies into a simplified yet robust stepwise estimator with improved vectorization and conditioning checks, thereby delivering higher accuracy and better scalability on large test sets.The detailed analysis of GeoEvolve-GeoCP can be found at Appendix A.4.2.

### 4.2.2 MODEL EVALUATION

To perform GeoCP, we first build a house-price prediction model using a base predictor with eight explanatory variables and two spatial variables as inputs. The trained model is then assessed with GeoCP to quantify predictive uncertainty, and the final output is the uncertainty of house-price predictions on the test set. In this study, we choose XGBoost as the base predictor, which achieves an $R^2$ of 0.871 and an RMSE of 7.362 (10,000 USD). The results are presented in Figure 5. The predicted uncertainty exhibits a clear spatial pattern: it is highest around Lake Washington in downtown Seattle, slightly lower in suburban areas, and lowest in the rural southern region. A scatter plot of predicted uncertainty versus predicted price further reveals that uncertainty increases with house price, peaking at approximately 125 (10,000 USD) and then leveling off with a slight decline.

---

[1]https://geodacenter.github.io/data-and-lab/KingCounty-HouseSales2015/

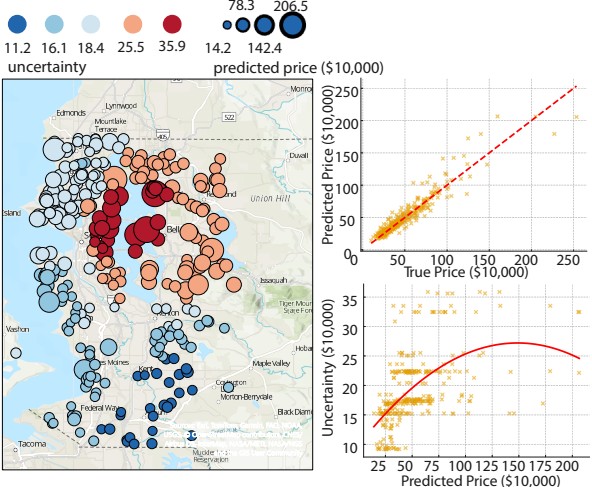

Figure 5: The spatial distribution of estimated uncertainty for the housing price prediction task in Seattle using the evolved GeoCP.

Table 2: Comparison of conformal prediction metrics. Smaller Average Interval Size and Interval Score indicate sharper and more efficient intervals.

| Method | Average Interval Size ↓ | Interval Score ↓ |
|---|---|---|
| Original | 18.3254 | 44.7611 |
| OpenEvolve (No GeoKnowledge) | 16.9139 | 43.1823 |
| OpenEvolve (General GeoKnowledge) | 17.1508 | 42.8343 |
| OpenEvolve (Specific GeoKnowledge) | 13.9557 | 41.4267 |
| GeoEvolve (No RAG) | 17.5586 | 44.2545 |
| GeoEvolve (Static RAG) | 18.5818 | 45.2738 |
| GeoEvolve (Dynamic RAG) | **13.7750** | **41.2389** |

We apply GeoCP in seven configurations, original, OpenEvolve without GeoKnowledge Prompt, OpenEvolve with General GeoKnowledge Prompt, OpenEvolve with Specific GeoKnowledge Prompt, GeoEvolve without GeoKnowRAG, GeoEvolve with Static GeoKnowRAG, and GeoEvolve with Dynamic GeoKnowRAG–to quantify uncertainty on the same test set. able 2 reports the GeoCP performance obtained by different methods. As shown, different variants of OpenEvolve reduces the interval score to 43.1823, 42.8343, and 41.4267, respectively. In comparison with OpenEvolve, the three variants of GeoEvolve achieves an interval score of 44.2545, 45.2738, 41.2389, respectively. The performance of GeoEvolve with static GeoKnowRAG even degrades, this may suggest that GeoKnowRAG fails to provide useful geographical knowledge for evolution. However, when dynamically updating new geographical knowledge, GeoEvolve shows unprecedented performance.

## 5 CONCLUSION

We presented GeoEvolve, a multi-agent LLM framework that couples evolutionary code search with geospatial domain knowledge via GeoKnowRAG to automate geospatial model discovery. Across three fundamental tasks, GeoEvolve consistently improved upon classical baselines and strong OpenEvolve variants. Ablations confirm that structured, domain-guided retrieval is pivotal: removing GeoKnowRAG degrades performance despite identical evolutionary budgets, underscoring the value of grounding algorithm evolution in geospatial theory.

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

# A APPENDIX

## A.1 SPATIAL REGRESSION MODEL

**Task- Spatial regression** Spatial regression explicitly introduces geospatial context into the statistical framework of regression. One wants to combine space with the statistical models when he or she thinks geospatial space can play an essential role in the data generation process or use space as a proxy for some factors difficult to obtain.

**Model- GWR** In this work, we selected geographically weighted regression (GWR) (Fotheringham et al., 2009), one of the most famous spatial regression models. For GWR, the regression coefficients are not fixed, but depend on the geographical coordinates of observations, which is defined as follows:

$$y_i = \beta_0(u_i, v_i) + \sum_{k=1}^{K} \beta_k(u_i, v_i)x_{ik} + \varepsilon_i \tag{3}$$

where $(u_i, v_i)$ are the geographical coordinates.

**Evaluator** As for GWR, we use the coefficient of determination ($R^2$) as the evaluation metric. Our objective is to obtain an evolved GWR model that has the highest $R^2$.

**Datasets** The Georgia census data[2] is extracted from GWmodel, a R package that contains a group of geographically weighted models. The original data contains 7 variables and 2 pairs of geographical coordinates expressed in geodetic and projected coordinate systems, respectively. In this work, we employ the percentage of the county population with a bachelor's degree as the target variable, and the other 6 variables (total population, rural population percentage, elderly (65+) population percentage, foreign-born population percentage, population living below the poverty line percentage, black population percentage) as explanatory variables.

## A.2 USE OF LLMS

We use LLMs to polish selected paragraphs and to automatically extract differences between algorithms (e.g., Kriging and GeoCP) produced by different code-generation methods (e.g., OpenEvolve and GeoEvolve), thereby facilitating the analysis of GeoEvolve's specific improvements and their underlying causes. All research ideas were independently conceived by the authors.

## A.3 CODE ANALYZER

Figure 6 shows the template of the Code Analyzer and an example output.

---

[2]https://r-packages.io/datasets/Georgia

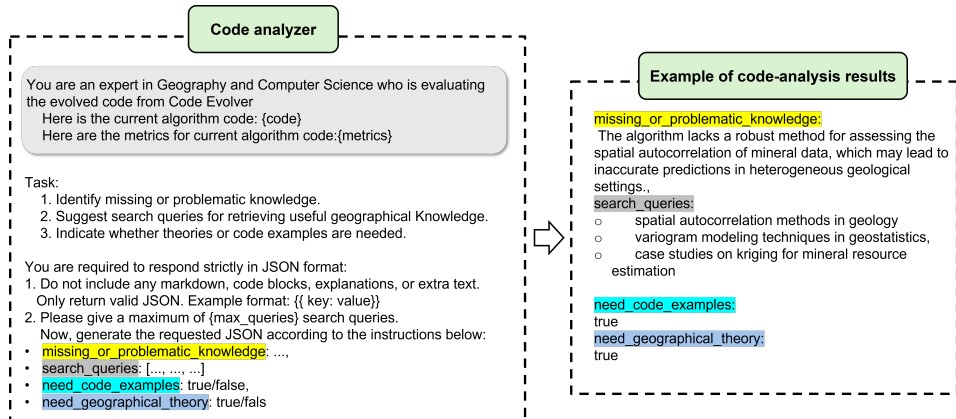

Figure 6: The template and an example of code analyzer

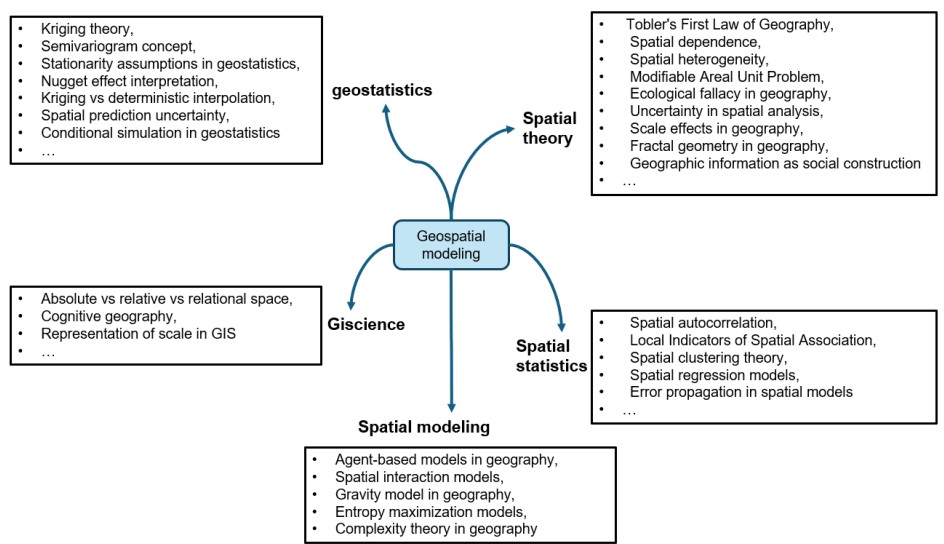

Figure 7: The keywords used for constructing geospatial knowledge database

## A.4 GEOSPATIAL KNOWLEDGE DATABASE

The geospatial knowledge is initialized automatically from web (e.g., arxiv, wikipedia, github, etc.) according to the user-defined keywords and can be updated according to the requirements dynamically during evolution. Figure 7 shows an example of constructed geospatial knowledge database, the five categories are geostatistics, spatial theory, GIScience, spatial statistics, and spatial modeling.

It should be noted that the construction of a geospatial knowledge base can include many more keywords, enabling a much larger scale—potentially comprising thousands of documents or developed through more sophisticated processes. In the present experiments, however, we intentionally created a small-scale knowledge base to validate the effectiveness of GeoEvolve on three algorithmic tasks. We expect that GeoEvolve will achieve even greater performance gains when combined with a larger and more comprehensive geospatial knowledge base in future work.

## A.5 BENCHMARK METHODS

Figure 8 illustrates the GeoEvolve without RAG version used in our ablation study. The algorithm still consists of an outer loop and an inner loop. After the agent proposes an improvement to the algorithm in the inner loop, the code analyzer evaluates the updated code. In this version, the system

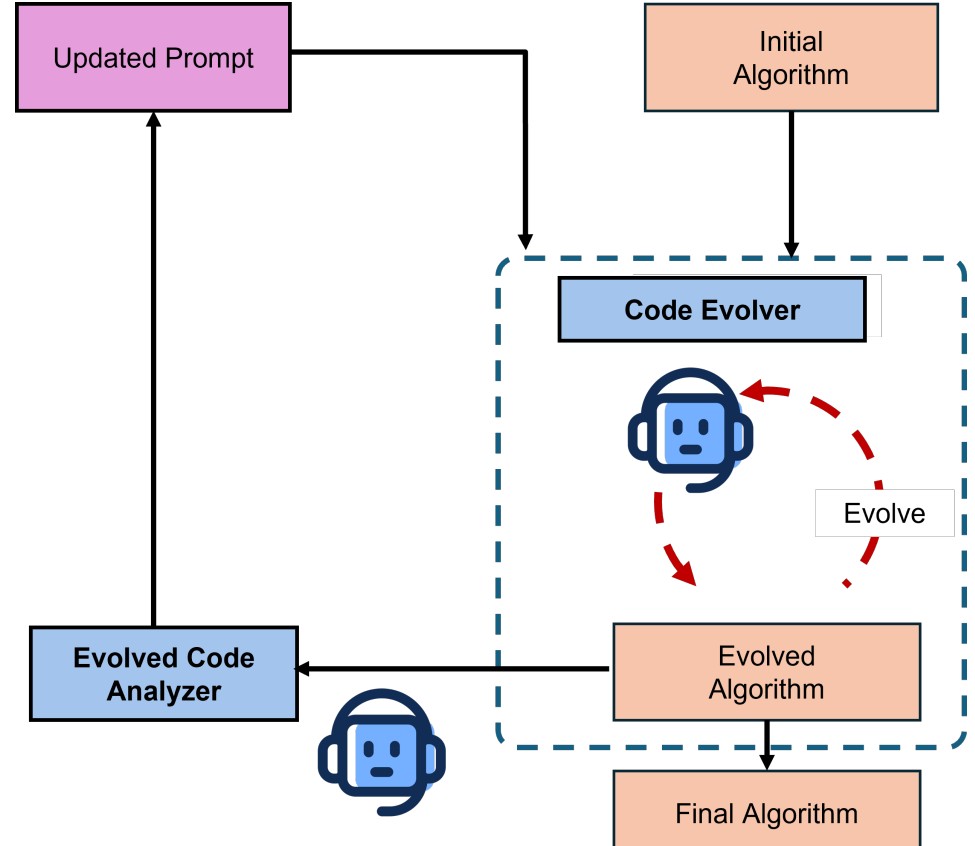

Figure 8: GeoEvolve without RAG

does not retrieve any information from the GeoKnowRAG geospatial knowledge base; instead, it directly updates the prompt and returns to the inner loop to further evolve the algorithm.

### A.6 LLM CONFIGURATION

To ensure reproducibility and a fair comparison across model families, we report the exact large language models (LLMs) used in all components of our system, including the OpenEvolve baseline, the GeoEvolve framework, the GeoKnowRAG retrieval module, and the outer-loop agentic controller. Across all LLM families (GPT, Gemini, Qwen), we adopt a consistent two-tier strategy: a *primary* model for code mutation and generation, and a *secondary* model for validation, refinement, and fallback reasoning. Retrieval modules use the corresponding embedding model for vectorization, and the agent controller employs a lightweight but reasoning-capable model to support outer-loop decision making.

**GPT family.** OpenEvolve uses `GPT-4o` as the primary evolver and `GPT-4.1` as the secondary validator. GeoEvolve adopts the same configuration. GeoKnowRAG embeds all knowledge documents using `text-embedding-3-large`. The outer-loop agent controller also operates on `GPT-4.1`, balancing reasoning depth and runtime efficiency.

**Gemini family.** Both OpenEvolve and GeoEvolve use `Gemini-2.5-flash` as the primary evolver and `Gemini-2.5-pro` as the secondary model. GeoKnowRAG employs `gemini-embedding-001`, and the agent controller runs on `Gemini-2.5-flash`.

**Qwen family.** For the Qwen models, OpenEvolve and GeoEvolve use `Qwen3-235B` (primary) and `Qwen3-32B` (secondary). GeoKnowRAG uses `qwen3-embedding-8B`, and the agent controller also runs on `Qwen3-32B`.

Table 3: LLM configuration for all components of OpenEvolve and GeoEvolve.

| Component | GPT | Gemini | Qwen |
|---|---|---|---|
| OpenEvolve (primary) | GPT-4o | Gemini-2.5-flash | Qwen3-235B |
| OpenEvolve (secondary) | GPT-4.1 | Gemini-2.5-pro | Qwen3-32B |
| GeoEvolve (primary) | GPT-4o | Gemini-2.5-flash | Qwen3-235B |
| GeoEvolve (secondary) | GPT-4.1 | Gemini-2.5-pro | Qwen3-32B |
| GeoKnowRAG embeddings | text-embedding-3-large | gemini-embedding-001 | qwen3-embedding-8B |
| Agent controller | GPT-4.1 | Gemini-2.5-flash | Qwen3-32B |

Table 4: Runtime comparison of GeoEvolve and OpenEvolve variants using GPT-4.1 across three geospatial tasks.

| RAG Setting | Task | Dataset | Time (s) | Hours |
|---|---|---|---|---|
| GeoEvolve (Dynamic RAG) | Kriging | Australia Minerals | 3085.37 | 0.86 |
| | GeoCP | Seattle House Price | 4546.91 | 1.26 |
| | GWR | Georgia Census | 1862.72 | 0.52 |
| GeoEvolve (Static RAG) | Kriging | Australia Minerals | 2730.55 | 0.76 |
| | GeoCP | Seattle House Price | 4750.32 | 1.32 |
| | GWR | Georgia Census | 2446.12 | 0.68 |
| GeoEvolve (No RAG) | Kriging | Australia Minerals | 2065.91 | 0.57 |
| | GeoCP | Seattle House Price | 2235.45 | 0.62 |
| | GWR | Georgia Census | 1786.66 | 0.50 |
| OpenEvolve (No GeoKnowledge) | Kriging | Australia Minerals | 1192.98 | 0.33 |
| | GeoCP | Seattle House Price | 402.89 | 0.15 |
| | GWR | Georgia Census | 343.55 | 0.10 |
| OpenEvolve (General GeoKnowledge) | Kriging | Australia Minerals | 1670.90 | 0.46 |
| | GeoCP | Seattle House Price | 380.93 | 0.11 |
| | GWR | Georgia Census | 100.61 | 0.03 |
| OpenEvolve (Specific GeoKnowledge) | Kriging | Australia Minerals | 1437.91 | 0.40 |
| | GeoCP | Seattle House Price | 539.68 | 0.15 |
| | GWR | Georgia Census | 667.19 | 0.19 |

This unified LLM configuration is crucial for interpreting our ablations. The GeoEvolve without GeoKnowRAG variant keeps the *identical* primary and secondary models, ensuring that performance differences arise solely from the absence of structured domain knowledge rather than changes in model capacity. Similarly, using matched primary/secondary pairs across OpenEvolve and GeoEvolve removes confounding effects from heterogeneous model dependencies. In our experiments, replacing the secondary models with weaker reasoning engines leads to noticeably less stable evolutionary trajectories, confirming that fallback validation is essential for preventing code drift and maintaining interpretable improvements. Thus, the LLM design is not merely an implementation detail but a controlled experimental factor that enables clean causal attribution in our ablation study.

## A.7 TIME

Table 4 reports the full runtime comparison of GeoEvolve and OpenEvolve using GPT-4.1 across the three geospatial tasks. Overall, GeoEvolve incurs additional computational cost due to its two-level agentic control loop and the GeoKnowRAG retrieval mechanism, but the overhead is consistent and interpretable. For Dynamic RAG, GeoEvolve requires 0.52–1.26 hours per task, while Static RAG slightly reduces the overhead to 0.68–1.32 hours. Removing RAG reduces the runtime further to 0.50–0.62 hours, confirming that a substantial portion of the overhead comes from knowledge retrieval rather than code evolution itself. In contrast, OpenEvolve—without geospatial knowledge integration—runs considerably faster (0.03–0.33 hours), but this speed comes at the cost of weaker algorithmic improvements. These results reflect a clear trade-off: integrating structured geospatial knowledge increases runtime but enables GeoEvolve to produce substantially stronger and more stable algorithmic improvements.

Figure 9 presents the average runtime of GeoEvolve across three LLM families (GPT-4.1, Gemini-2.5, and Qwen3-32B) under both Dynamic and Static RAG settings. Two clear patterns emerge from the results.

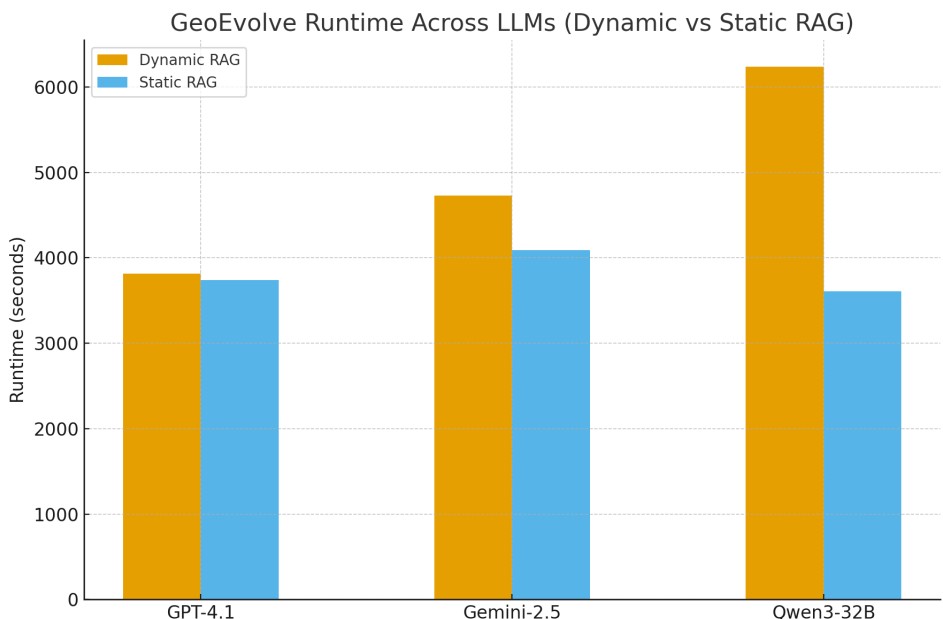

Figure 9: Average runtime of GeoEvolve across three LLM families (GPT-4.1, Gemini-2.5, and Qwen3-32B) under Dynamic and Static RAG.

First, the choice of LLM strongly affects computational cost. GPT-4.1 achieves the lowest runtime (approximately 3.7k seconds), Gemini-2.5 is moderately slower (around 4.1–4.7k seconds), and Qwen3-32B is the slowest (over 6.2k seconds). This ordering reflects the inherent inference latency of each model, indicating that GeoEvolve's execution time scales proportionally with the underlying LLM's response speed.

Second, Static RAG consistently outperforms Dynamic RAG in runtime across all LLMs. Static RAG avoids repeated retrieval–summarization cycles in the outer loop, whereas Dynamic RAG regenerates the knowledge context at every iteration, leading to additional overhead. The effect is particularly pronounced for Qwen3-32B, where Dynamic RAG incurs nearly 70% more latency compared with Static RAG.

Overall, these results highlight a practical trade-off: Dynamic RAG provides higher retrieval adaptivity at the cost of increased runtime, while Static RAG offers more efficient execution with slightly reduced flexibility. This confirms that (i) geospatial algorithm evolution is sensitive to LLM inference speed, and (ii) users may balance computational efficiency and retrieval precision by choosing between Static and Dynamic RAG modes.

## A.8 GENERALIZATION EXPERIMENT

### A.8.1 DATASETS

We selected 3 datasets per task (9 total datasets) to ensure comprehensive coverage.

- Kriging: Australian Minerals, Ocean Chlorophyll, Temperature Station Data.
- GeoCP: Seattle Housing Price, US Life Expectancy, China PM2.5.
- GWR (New): Georgia Education, NYC Income, Chicago Health.

The detailed descriptions about the datasets used in thie work is displayed in Table 5.

### A.8.2 DOMAIN GENERALIZATION

Our goal is to demonstrate robust cross-domain transferability. We expect the models evolved on a source domain (e.g., housing prices) to effectively generalize to target domains (e.g., minerals).

| Model | Name | Description |
|-------|------|-------------|
| Kriging | Australian Minerals | Spatial measurements of Cu, Pb, and Zn from a region in Australia, selected for their significance as indicators of environmental contamination and ecological health. |
| Kriging | Ocean Chlorophyll | The Ocean Chlorophyll dataset includes 4,136 highly clustered chlorophyll observations collected near Townsville, Australia. |
| Kriging | Temperature Station Data | 90-day ambient temperature covering Los Angeles County from 1st January to 31st March, 2019, collected from Weather Underground. |
| GeoCP | Seattle Housing | Home sales prices and characteristics for Seattle. |
| GeoCP | US Life Expectancy | Life expectancy and related sociodemographic variables for US counties. |
| GeoCP | China PM2.5 | PM2.5 concentration and related variables for China cities |
| GWR | Georgia Education | Census data about education from the county of Georgia, USA |
| GWR | NYC Income | Block-level Earnings New York City (2002-14) from Longitudinal Employer-Household Dynamics (LEHD). |
| GWR | Chicago Health | Public health and socio-economic indicators for the 77 community areas of Chicago, IL, 2014. |

Table 5: Details about datasets employed in the domain generalization experiment

Specifically, the evolution phase was conducted on the Australian Minerals dataset for Kriging, Seattle Housing for GeoCP, and Georgia Education for GWR. The results for three models are as follows.

**Kriging** Kriging Task (RMSE ↓): OpenEvolve baselines exhibit catastrophic failure (divergence) on the Ocean Chlorophyll dataset, whereas GeoEvolve remains robust (see Table 6).

Table 6: Model performance comparison of evolved Kriging

| Model | Australian Minerals | | | Ocean | Temperature |
|-------|------|------|------|------------|-------------|
| | **Cu** | **Pb** | **Zn** | **Chlorophyll** | |
| Original | 0.9139 | 0.6619 | 0.6294 | 0.9949 | 1.1567 |
| GeoEvolve (Dynamic RAG) | 0.8718 | 0.6131 | **0.5852** | 0.9916 | 1.1634 |
| GeoEvolve (Static RAG) | **0.8602** | **0.5927** | 0.5941 | 0.6179 | **0.5417** |
| GeoEvolve (No RAG) | **0.8602** | **0.5927** | 0.5941 | **0.5441** | 1.0499 |
| OpenEvolve (No GeoKnow) | 0.8727 | 0.6413 | 0.6245 | **0.5296** | 1.1083 |
| OpenEvolve (General GeoKnow) | 0.9264 | 0.6519 | 0.6333 | 0.6158 | 2.0221 |
| OpenEvolve (Specific GeoKnow) | 0.9139 | 0.6632 | 0.6338 | Fail (460.4035) | 5.7484 |

**GeoCP** GeoCP Task (Interval Score ↓): Dynamic RAG achieves the best scores across all datasets, significantly reducing uncertainty compared to baselines (see Table 7).

**GWR** GWR Task ($R^2$ ↑): GeoEvolve (Dynamic RAG) consistently achieves the highest or near-highest $R^2$, demonstrating strong transferability. In contrast, OpenEvolve with Specific GeoKnow performs poorly on the source domain (Georgia), indicating overfitting or prompt misalignment (see Figure 8).

Table 7: Model performance comparison of evolved GeoCP

| Model | Dataset | Interval Score ↓ | Coverage | Avg Interval Size |
|---|---|---|---|---|
| Original | Seattle House Price | 44.7611 | 0.9533 | 18.3254 |
| | US Life Expectancy | 121.4673 | 0.9098 | 50.2762 |
| | China PM2.5 | 23.4409 | 0.9295 | 10.6332 |
| GeoEvolve (Dynamic RAG) | Seattle House Price | **41.2389** | 0.9000 | 13.7750 |
| | US Life Expectancy | **113.2176** | 0.8922 | 41.5560 |
| | China PM2.5 | **21.0186** | 0.9507 | 9.3529 |
| GeoEvolve (Static RAG) | Seattle House Price | 45.2738 | 0.9600 | 18.7862 |
| | US Life Expectancy | 123.5681 | 0.9424 | 53.4537 |
| | China PM2.5 | **21.0186** | 0.9507 | 9.3529 |
| GeoEvolve (No RAG) | Seattle House Price | 44.2545 | 0.9433 | 17.5586 |
| | US Life Expectancy | 141.2780 | 0.9424 | 67.3331 |
| | China PM2.5 | 24.0525 | 0.9437 | 10.9554 |
| OpenEvolve (No GeoKnow) | Seattle House Price | 43.1823 | 0.9333 | 16.9139 |
| | US Life Expectancy | 123.4440 | 0.8897 | 45.9977 |
| | China PM2.5 | 26.1946 | 0.9146 | 9.4270 |
| OpenEvolve (General GeoKnow) | Seattle House Price | 42.8343 | 0.9333 | 17.1508 |
| | US Life Expectancy | 124.3828 | 0.9023 | 51.4155 |
| | China PM2.5 | 27.0417 | 0.9085 | 10.0772 |
| OpenEvolve (Specific GeoKnow) | Seattle House Price | 41.4267 | 0.9033 | 13.9557 |
| | US Life Expectancy | 124.0236 | 0.8671 | 42.2759 |
| | China PM2.5 | 26.7654 | 0.8732 | 8.1674 |

Table 8: Model performance comparison of evolved GWR

| Model | Georgia Education | NYC Income | Chicago Health |
|---|---|---|---|
| Original | 0.1564 | 0.7065 | 0.5999 |
| GeoEvolve (Dynamic RAG) | 0.3556 | **0.7385** | **0.6221** |
| GeoEvolve (Static RAG) | **0.4524** | 0.5039 | 0.4927 |
| GeoEvolve (No RAG) | **0.4524** | 0.5038 | 0.5388 |
| OpenEvolve (No GeoKnow) | 0.2287 | 0.6238 | 0.5388 |
| OpenEvolve (General GeoKnow) | 0.2353 | 0.6317 | 0.6008 |
| OpenEvolve (Specific GeoKnow) | 0.1367 | 0.7297 | 0.6074 |

### A.8.3 SPATIAL GENERALIZATION

For geospatial models, spatial generalization is equally important. Taking GWR for New York income dataset as an example, we use a Spatial Leave-One-Out (SpatialLOO) approach (training on $N-1$ regions, testing on held-out region). Figure 10 offers a general illustration of SpatialLOO. GeoEvolve with Dynamic RAG achieved the lowest RMSE and standard deviation, proving it adapts best to unseen spatial distributions. The performance of spatial generalization is shown in the Table 9.

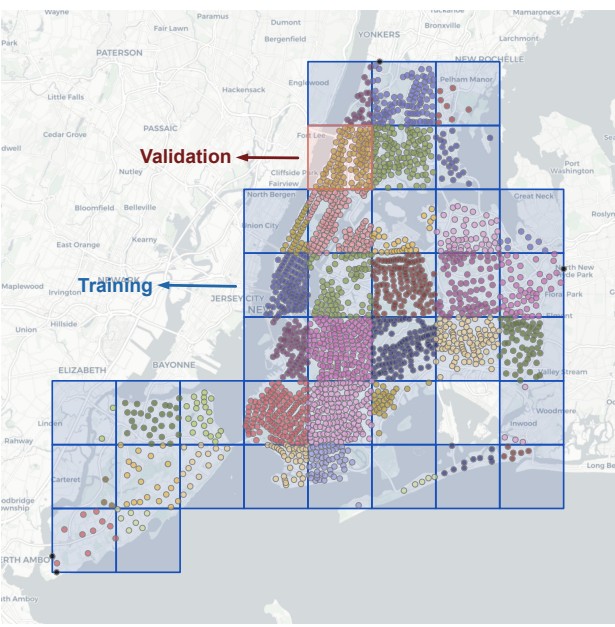

Figure 10: Spatial Leave-One-Out sampling for evaluating spatial generalization

Table 9: Performance of spatial generalization

| Model | Mean RMSE | Std RMSE |
|---|---|---|
| Original | 2.165 | 1.66 |
| GeoEvolve with Dynamic RAG | **1.968** ↓ | **1.532** ↓ |
| GeoEvolve with Static RAG | 2.3 ↑ | 1.571 ↓ |
| GeoEvolve without RAG | 2.289 ↑ | 1.687 ↑ |
| OpenEvolve without GeoKnow | 2.432 ↑ | 1.707 ↑ |
| OpenEvolve with General GeoKnow | 2.059 ↑ | 1.641 ↑ |
| OpenEvolve with Specific GeoKnow | 2.378 ↑ | 2.366 ↑ |

### A.8.4 TEMPORAL GENERALIZATION

Temporal generalization can also be vital in tasks involving spatiotemporal prediction, so we design experiments for evaluating temporal generalization performance: ensuring that a geospatial model evolved on data from one specific time period (e.g., 2024) maintains high performance when applied to datasets from a different time period (e.g., 2025).

Taking Kriging for temperature interpolation as an example, we trained on historical data (Jan 2019) and tested on future data (next 100 days). As shown in Table 10, GeoEvolve with Static RAG yielded the highest mean improvement (0.39), while OpenEvolve variants caused performance degradation (negative improvement).

Table 10: Performance of temporal generalization

| Model | Avg. Improvement | Med. Improvement | Std | Min | Max |
|---|---|---|---|---|---|
| GeoEvolve with Dynamic RAG | -0.00325 | -0.00312 | 0.00125 | -0.00698 | 0.00012 |
| GeoEvolve with Static RAG | **0.39243** | **0.44375** | 0.22398 | -0.40564 | 0.71676 |
| GeoEvolve without RAG | 0.05269 | 0.05169 | 0.02014 | -0.00796 | 0.10649 |
| OpenEvolve with General GeoKnow | -0.38191 | -0.33568 | 0.17768 | -0.86385 | -0.16534 |
| OpenEvolve with Specific GeoKnow | -3.02455 | -2.93563 | 0.75308 | -4.49663 | -1.75598 |
| OpenEvolve without GeoKnow | 0.02625 | 0.01741 | 0.03027 | -0.01006 | 0.12728 |

## A.9 ORIGINAL ALGORITHM

### A.9.1 ORIGINAL ALGORITHM OF ORIDINARY KRIGING.

Kriging is a geostatistical spatial interpolation method that provides the *best linear unbiased estimator* (BLUE) of an unknown value at a location by optimally weighting surrounding observations. It assumes that the spatial process $Z(s)$ can be represented as

$$Z(s) = \mu + \varepsilon(s), \tag{4}$$

where $\mu$ is an unknown constant mean and $\varepsilon(s)$ is a zero-mean, second-order stationary random field. The key assumption of second-order stationarity requires that the mean is constant and that the covariance depends only on the lag vector $h$, i.e.,

$$\mathrm{Cov}\big[Z(s), Z(s+h)\big] = C(h), \tag{5}$$

or equivalently through the semivariogram $\gamma(h)$.

Ordinary kriging predicts the value at an unsampled location $s_0$ as a weighted linear combination of the observed data:

$$\hat{Z}(s_0) = \sum_{i=1}^{n} \lambda_i Z(s_i), \tag{6}$$

subject to the unbiasedness constraint

$$\sum_{i=1}^{n} \lambda_i = 1. \tag{7}$$

The kriging weights $\lambda_i$ are determined by minimizing the estimation variance

$$\sigma_k^2 = \mathrm{Var}\big[\hat{Z}(s_0) - Z(s_0)\big] \tag{8}$$

using the spatial covariance or variogram model.

### A.9.2 ORIGINAL ALGORITHM OF GEOCP

GeoConformal Prediction (GeoCP) is a model-agnostic framework for quantifying spatial prediction uncertainty by extending *conformal prediction (CP)* with explicit geographic weighting. Conformal prediction provides finite-sample, distribution-free prediction intervals by computing nonconformity scores on a calibration set and selecting the $(1 - \varepsilon)$ quantile to guarantee coverage. However, standard CP assumes data exchangeability and yields intervals of constant width, which is violated in geospatial settings where strong spatial heterogeneity and covariate shift are common.

To overcome these limitations, GeoCP integrates spatial dependence directly into the conformal framework. Given a geospatial model $f : \mathcal{X} \to \mathcal{Y}$ trained on a set of observations and a calibration set $\{(X_i, y_i)\}_{i=1}^{m}$, let $a(\cdot)$ be a nonconformity score (e.g., absolute residual) and $a_i = a(f(X_i), y_i)$ for calibration point $i$. For a test location $X_{\text{test}}$ with geographic coordinates $(u_{\text{test}}, v_{\text{test}})$, GeoCP assigns each calibration point $i$ a geographic weight

$$w_i(u_{\text{test}}, v_{\text{test}}) = \frac{K_\sigma\big(d((u_{\text{test}}, v_{\text{test}}), (u_i, v_i))\big)}{\sum_{j=1}^{m} K_\sigma\big(d((u_{\text{test}}, v_{\text{test}}), (u_j, v_j))\big)}, \tag{9}$$

where $d(\cdot, \cdot)$ is the geographic distance and $K_\sigma$ is a distance-decay kernel (e.g., Gaussian). These weights reflect Tobler's first law of geography—that nearby observations are more similar—thus relaxing the exchangeability requirement of classical CP.

The GeoCP prediction interval for $X_{\text{test}}$ is then defined as

$$C_{\text{geo}}(X_{\text{test}}) = \left\{ y : \ a\big(f(X_{\text{test}}), y\big) \leq Q_{1-\varepsilon}^{\text{geo}}(\{a_i\}, \{w_i(u_{\text{test}}, v_{\text{test}})\}) \right\}, \tag{10}$$

where $Q_{1-\varepsilon}^{\text{geo}}$ is the geographically weighted $(1-\varepsilon)$-quantile computed as

$$Q_{1-\varepsilon}^{\text{geo}} = \inf\left\{ q : \sum_{i=1}^{m} w_i(u_{\text{test}}, v_{\text{test}}) \, \mathbf{1}\{a_i \leq q\} \geq 1 - \varepsilon \right\}. \tag{11}$$

Algorithmically, GeoCP proceeds as follows: (1) split the dataset into training, calibration, and test sets; (2) fit the spatial prediction model $f$ on the training set; (3) compute nonconformity scores $\{a_i\}$ on the calibration set; (4) for each test point, calculate geographic weights $w_i$ via (9); (5) determine the geographically weighted quantile (11) and form the prediction interval (10).

By construction, GeoCP inherits the rigorous finite-sample coverage guarantee of conformal prediction,

$$\mathbb{P}[y_{\text{test}} \in C_{\text{geo}}(X_{\text{test}})] \geq 1 - \varepsilon,$$

while producing *spatially varying* prediction intervals that directly reflect local heterogeneity. Because it does not require modifying the underlying predictive model, GeoCP can be applied seamlessly to classical geostatistical methods (e.g., Kriging) and modern GeoAI models, providing a unified and interpretable framework for uncertainty quantification and supporting fair, responsible geographic decision-making.

## A.10 Evolved Kriging model

### A.10.1 GeoEvolve-Kriging (our model)

Compared with the original Ordinary Kriging, GeoEvolve–Kriging preserves the core structure while introducing the following key innovations:

- **Expanded and automatically selected variogram family.** Instead of a single non-standard exponential model, GeoEvolve fits a flexible family

$$\gamma_\theta(h) = \theta_0 + \theta_1\left[1 - \exp\big(-(h/\theta_2)^p\big)\right], \tag{12}$$

where $p = 1$ yields the exponential model, $p = 2$ the Gaussian model, and $p \in (0, 2)$ the *Matérn* family (with smoothness $\nu$). Candidate models $\{\text{Exponential}, \text{Gaussian}, \text{Linear}, \text{Matérn}\}$ are compared using information criteria such as

$$\text{AIC} = 2k - 2\log L, \qquad \text{BIC} = k\log n - 2\log L, \tag{13}$$

and the optimal variogram is selected by minimum AIC/BIC. This multi-model, multi-start search avoids local minima and captures a wide spectrum of spatial smoothness.

- **Adaptive empirical variogram estimation.** GeoEvolve constructs the empirical semi-variogram using adaptive binning based on Silverman's rule or quantiles:

$$\hat{\gamma}(h_k) = \frac{1}{2|N(h_k)|} \sum_{(i,j) \in N(h_k)} [Z(x_i) - Z(x_j)]^2, \tag{14}$$

where $N(h_k)$ is the set of pairs with distances in the $k$th adaptive bin. Robust trimmed means and an automatic choice of $n_{\text{lags}} \in [8, 20] \propto \sqrt{n}$ reduce the impact of outliers and distance heterogeneity.

- **Robust model fitting.** Parameter estimation in (12) is performed via multi-start global optimization with either

$$\min_\theta \sum_k w_k \, |\hat{\gamma}(h_k) - \gamma_\theta(h_k)| \tag{15}$$

(robust L1 loss) or weighted least squares, depending on empirical residual patterns, where $w_k$ are bin-based weights. This strategy guards against local minima and ensures sill $\theta_1$ and range $\theta_2$ remain physically meaningful.

- **Localized kriging with adaptive regularization.** To improve scalability and stability, GeoEvolve restricts the kriging system to the $K$ nearest neighbors (e.g., $K = 25$) of $x_0$ using a cKDTree and adds a condition-number–dependent diagonal adjustment:

$$\mathbf{K}_{\text{loc}}\lambda = \mathbf{k}_{\text{loc}}, \qquad \mathbf{K}_{\text{loc}} \leftarrow \mathbf{K}_{\text{loc}} + \epsilon(\kappa)\mathbf{I}, \tag{16}$$

  where $\epsilon(\kappa)$ is an adaptive nugget (e.g., $10^{-10}$ to $10^{-4}$) determined by the matrix condition number $\kappa$. This reduces computational cost from $O(n^3)$ to $O(K^3)$ and stabilizes inversion in ill-conditioned settings.

- **Adaptive data transformation.** GeoEvolve applies an adaptive log transform

$$Z' = \log(Z + \delta), \tag{17}$$

  where the offset $\delta$ is chosen from the 1st percentile of positive values plus a small $\epsilon$ to reduce skewness and ensure valid back-transformation.

### A.10.2 Comparison of Evolved Kriging from Different Models

In this section, we analyze the main technical components of different algorithm:

**Variogram family.** Original uses only the exponential variogram with a non-standard form $nugget + sill(1 - e^{-h \cdot range})$. OpenEvolve standardizes the form to $e^{-h/range}$ and adds Gaussian and Linear options. OpenEvolve with GeoKnowledge adopts the same set but applies automatic model selection among candidate models. GeoEvolve further introduces the Matern family ($\nu = 0.2$–$3.0$) with full AIC/BIC-based automatic selection and multi-start optimization.

**Empirical variogram.** Original employs 12 equal-width bins including zero distance and is unweighted. OpenEvolve truncates distances to 85% of the maximum and removes NaN bins. OpenEvolve with GeoKnowledge follows the same procedure but adds minimal pair control. GeoEvolve uses adaptive binning via Silverman's rule or quantiles, applies a robust trimmed mean, and automatically sets $n_{\text{lags}} = 8$–$20 \propto \sqrt{n}$.

**Model fitting.** Original applies an L1 loss with a single L-BFGS-B run. OpenEvolve still uses L1 but adds parameter bounds, smart initialization, and a fallback strategy. OpenEvolve with GeoKnowledge switches to L2 loss and selects the best model by minimum MSE. GeoEvolve adopts a robust L1 loss, multi-start global search, Matern smoothness grid, and AIC/BIC complexity penalties.

**Kriging solver.** Original builds a global system without neighborhood selection. OpenEvolve introduces diagonal regularization ($10^{-10}$) and a pseudo-inverse fallback. OpenEvolve with GeoKnowledge is identical. GeoEvolve employs localized kriging using cKDTree nearest 25 neighbors and condition-number–adaptive regularization ($10^{-10}$–$10^{-4}$), with mean fallback if the system is singular.

### A.10.3 Knowledge discovery from GeoEvolve

We summarize the key geospatial knowledge underlying the improved GeoEvolve algorithm, which can contribute to geospatial modeling.

**Expanded variogram family with automatic selection.** Fits appropriate smoothness and range, lowering RMSE/MAE and improving $R^2$.

**Adaptive empirical variogram (trimmed mean, quantile bins).** Stabilizes nugget/sill/range estimates and reduces run-to-run variance.

**Multi-start with parameter bounds in optimization.** Improves convergence and avoids negative or degenerate parameter estimates.

**Localized kriging with condition-based regularization.** Reduces computational cost (from $O(n^3)$ to local operations) and improves robustness for ill-conditioned systems.

**Geo-knowledge injection.** Provides informative priors and narrows the search space, improving small-sample and non-stationary performance.

### A.11 EVOLVED GEOCP MODEL

#### A.11.1 GEOEVOLVE-GEOCP (OUR MODEL)

The fundamental conformal construction is preserved, but the following modifications are introduced:

- **Refined geographic weighting.** While keeping the Gaussian kernel form

$$w_i(u_{\text{test}}, v_{\text{test}}) = \frac{\exp\left[-\frac{1}{2}\left(\frac{d((u_{\text{test}}, v_{\text{test}}),(u_i, v_i))}{\sigma}\right)^2\right]}{\sum_{j=1}^{m}\exp\left[-\frac{1}{2}\left(\frac{d((u_{\text{test}}, v_{\text{test}}),(u_j, v_j))}{\sigma}\right)^2\right]}, \quad (18)$$

GeoEvolve reoptimizes the bandwidth parameter $\sigma$ through multi-start global search and adaptive clipping

$$\sigma \in [\sigma_{\min}, \sigma_{\max}], \quad (19)$$

ensuring both numerical stability and fidelity to local spatial heterogeneity.

- **Enhanced weighted quantile computation.** GeoEvolve consolidates earlier adaptive strategies into a simplified yet robust stepwise quantile estimator:

$$Q_{1-\varepsilon}^{\text{geo}} = \inf\left\{q : \sum_{i=1}^{m} w_i(u_{\text{test}}, v_{\text{test}})\,\mathbf{1}\{a_i \leq q\} \geq 1 - \varepsilon\right\}. \quad (20)$$

The algorithmic implementation uses improved vectorization and conditioning checks, guaranteeing accuracy and scalability on large test sets.

#### A.11.2 COMPARISON OF EVOLVED GEOCP FROM DIFFERENT MODELS

We summarize the key technical elements of the different code-evolution algorithms.

**Original GeoCP.** This version uses a fixed-bandwidth Gaussian kernel $e^{-0.5d^2}$ without weight normalization. It computes weighted quantiles with a *stepwise* rule, selecting the index where cumulative weights exceed $q$ without interpolation, and adopts the quantile level $q = \lceil(1-\alpha)(N+1)\rceil/N$, which is slightly conservative. Only the mean interval score is reported as the uncertainty metric. As a result, the method may produce overly wide or miscalibrated intervals in regions with strong spatial heterogeneity or sparse sampling.

**OpenEvolve.** This stage introduces adaptive bandwidth, dynamically adjusting kernel width for each test location based on its $k$-nearest neighbor distance and row-wise distance dispersion. It replaces the stepwise weighted quantile with interpolated weighted quantiles, avoiding discontinuous interval endpoints.

**OpenEvolve with GeoKnowledge.** Here the bandwidth is eo-knowledge guided: per-test $k$-NN bandwidths are clipped to the empirical range $[0.05, 0.5]$. Weight normalization ensures that each test point's kernel weights sum to one, providing numerical stability and spatial consistency. The quantile level is refined to $q = (1-\alpha)(N+1)/N$ (without ceiling), reducing conservativeness and shortening intervals. Furthermore, comprehensive UQ metrics are reported, including mean interval length, empirical coverage, and deviation from nominal coverage. Overall, this stage further shortens intervals and achieves near-nominal coverage while remaining robust at boundaries and in sparse areas.

**GeoEvolve.** GeoEvolve–GeoCP remains faithful to the core conformal prediction framework while sharpening spatial weighting and quantile estimation, the two pillars of interval construction. The refined geographic weighting adaptively tunes bandwidth to local heterogeneity, ensuring that conformal scores reflect the true spatial dependence and avoid instability.

#### A.11.3 KNOWLEDGE DISCOVERY FROM GEOEVOLVE

We distill the geospatial knowledge that underlies the improved GeoCP algorithm produced by Geo-Evolve.

**Adaptive bandwidth.** This mechanism adjusts kernel width to local calibration-point density, preventing overly wide intervals in dense regions and overly narrow ones in sparse regions. It drives the interval score down and keeps empirical coverage near $(1 - \alpha)$.

**Interpolated weighted quantile.** By eliminating discrete jumps when cumulative weights cross the quantile threshold, this refinement produces smoother, more stable prediction interval endpoints and lowers variance.

**Refined quantile level without ceiling.** This adjustment avoids the conservative upward bias from the ceiling function, shortens interval length, and keeps empirical coverage close to the nominal level.

