# OpenReview forum: "GeoEvolve: Automating Geospatial Model Discovery via Multi-Agent Large Language Models"
_ICLR.cc/2026/Conference — Submitted to ICLR 2026_

### Official Review · Reviewer_S3KU · 2025-10-31

**Soundness:** 3
**Presentation:** 3
**Contribution:** 3
**Rating:** 4
**Confidence:** 5

**Summary:**

This paper combines AlphaEvolve/Openevolve with two geospatial reasoning tasks. The authors use openevolve in the  inner loop carrying out code mutations, and have an outer loop querying a geospatial knowledge base. There are two tasks that are evaluated: one in kriging/ spatial interpolation  and the second in spatial uncertainty quantification,. In each case the authors reduceerrors in both by O(10-20%) above their chosen baseline.  They conduct ablation experiments comparing their pipeline to one with a geospatial RAG. For the RAG they set up a  structured knowledge base by collecting literature on core geospatial modeling concepts from Wikipedia, arXiv, and GitHub

**Strengths:**

This is very timely and the authors clearly show on the tasks that they evaluate how openevolve with a geospatial RAG system can lead to superior performance on two tasks.

**Weaknesses:**

--The authors only choose two tasks, and one example of a problem from each task.
--It is not clear how strong the baselines that are chosen are and thus whether the system being shown will add actual value to geospatial researchers.
--It is not clear why the RAG system improves performance. What does it retrieve that helps the evolutionary system do better? To what extent is this guaranteed by cherry picking the documents/etc in the RAG index? How problem specific is this index?

These weaknesses make it hard to assess the generality or importance of these conclusions. Especially when compared with openevolve itself--the claimed benefit here is on the system around openevolve but with so few questions and evaluations it is hard to tell what the generality of the value is.

**Questions:**

- did you choose the documents in the RAG database to correspond to the problems you were trying to solve?
--Was the backbone of the RAG system chosen to correspond to the two narrowly scoped problems you are evaluating? Did you do anything to make the RAG better by including particular tools/articles/etc
--why did the RAG enhanced system perform better. Please be specific in the context of the examples chosen.
--what are the solutions the system found? Why are they better? What were the specific innovations discovered by geoevolve?

---

> ### Author Response · Authors · 2025-11-26
> **Response to Reviewer S3KU (Part 1)**
>
> We sincerely thank the reviewer for the constructive feedback. We have significantly expanded our experiments to include more LLMs, ablation studies, and diverse datasets to demonstrate the robustness and scalability of GeoEvolve. We commit to promptly integrating all necessary revisions and new findings, including the resultant tables and figures, into the final version of the paper.
>
> It is important to note that we have implemented comprehensive architectural modifications to GeoEvolve, which now include:
>
> * Code-to-Formula Conversion Agent: A new agent that automatically processes the user-provided geospatial model code to be optimized, organizing and reformulating it into the specific computational structure (or formula) required by the GeoEvolve framework.
> * Automated Knowledge Base Construction Agent: A new agent that analyzes the input initial geospatial model, automatically extracts relevant keywords, and constructs the initial GeoKnow database.
> * Dynamic RAG Module: We introduced the Dynamic RAG module, which, after each inner evolutionary loop, utilizes the code analyzer results to identify and automatically update the knowledge base with the newly required geospatial knowledge.
>
>
> ### **Q1: The authors only choose two tasks... hard to assess the generality or importance of these conclusions... especially when compared with OpenEvolve.**
>
> **Response:**
> Thank you for your valuable comments and suggestions. To rigorously demonstrate the generality of GeoEvolve and distinguish its systemic value from OpenEvolve, we have conducted a massive expansion of our experimental suite.
>
> We have addressed the concern of **generality** across three specific dimensions:
>
> **1. Generalization across Multiple LLMs**
> To evaluate whether GeoEvolve depends on a specific LLM, we tested three distinct model families—**GPT-4.1**, **Gemini-2.5-flash**, and **Qwen3-32B**—under identical evolutionary setups with Dynamic RAG.
>
>   * **Kriging (Australian Minerals):** We report the RMSE for Cu, Pb, and Zn. Despite substantial differences in model architecture and scale, the RMSE differences across LLMs are negligible ($<0.05$), and all significantly outperform non-RAG variants.
>
> | LLM | Cu RMSE | Pb RMSE | Zn RMSE | Mean RMSE |
> | :--- | :--- | :--- | :--- | :--- |
> | **GPT-4.1** | 0.7871 | 0.6666 | 0.6846 | **0.7128** |
> | **Gemini-2.5** | 0.8954 | 0.7204 | 0.6574 | 0.7577 |
> | **Qwen3-32B** | 0.9380 | 0.6840 | 0.6388 | 0.7537 |
>
>   * **GeoCP (Seattle Housing):** All LLMs consistently improve uncertainty quantification (Interval Score $\downarrow$, Coverage $\uparrow$).
>       * **GPT-4.1:** Score: 0.8251, Coverage: 0.8967
>       * **Gemini-2.5:** Score: 0.8951, Coverage: 0.9433
>       * **Qwen3-32B:** Score: 0.9023, Coverage: 0.9133
>
> **Conclusion:** GeoEvolve is LLM-agnostic. The performance gains stem from domain-guided retrieval and evolutionary refinement rather than the specific strength of the base LLM.
>
> **2. Extended Ablation Studies**
> We have expanded our comparison to **6 distinct frameworks** to isolate the contribution of each component:
>
> 1.  **GeoEvolve with Dynamic RAG** (Our new proposed method with adaptive updates)
> 3.  **GeoEvolve with Static RAG**(Our previous version of GeoEvolve with proposed improvement)
> 4.  GeoEvolve without RAG
> 5.  OpenEvolve with General GeoKnow
> 6.  **OpenEvolve with Specific GeoKnow** (Our new proposed baseline with spectific geo-knowledge as the initial prompt)
> 8.  OpenEvolve without GeoKnow
>
> **3. Domain generlization for expanded datasets**
> We added a new task model (Geographically Weighted Regression - GWR) and now evaluate **3 datasets per task** (9 total datasets) to ensure comprehensive coverage.
>
>
>  * **Kriging:** Australian Minerals, Ocean Chlorophyll, Temperature Station Data.
>   * **GeoCP:** Seattle Housing, US Life Expectancy, China PM2.5.
>   * **GWR (New):** Georgia Education, NYC Income, Chicago Health.
>
> Our goal is to demonstrate robust cross-domain transferability. We expect the models evolved on a source domain (e.g., housing prices) to effectively generalize to target domains (e.g., minerals). Specifically, the evolution phase was conducted on the Australian Minerals dataset for Kriging, Seattle Housing for GeoCP, and Georgia Education for GWR. The results for three models are as follows.

---

> > ### Author Response · Authors · 2025-11-26
> > **Response to Reviewer S3KU (Part 2)**
> >
> > * **GWR Task ($R^2$ $\uparrow$):** GeoEvolve (Dynamic RAG) consistently achieves the highest or near-highest $R^2$, demonstrating strong transferability. In contrast, OpenEvolve with Specific GeoKnow performs poorly on the source domain (Georgia), indicating overfitting or prompt misalignment.
> >
> > | Model                         | Georgia Education | NYC Income | Chicago Health |
> > |------------------------------|-------------------|------------|----------------|
> > | Initial                      | 0.1564            | 0.7065     | 0.5999         |
> > | GeoEvolve (Dynamic RAG)      | 0.3556            | **0.7385**     | **0.6221**         |
> > | GeoEvolve (Static RAG)       | **0.4524**        | 0.5039     | 0.4927         |
> > | GeoEvolve (No RAG)           | **0.4524**        | 0.5038     | 0.5388         |
> > | OpenEvolve (No GeoKnow)      | 0.2287            | 0.6238          | 0.5388         |
> > | OpenEvolve (General GeoKnow) | 0.2353            | 0.6317     | 0.6008     |
> > | OpenEvolve (Specific GeoKnow)| 0.1367            | 0.7297 | 0.6074     |
> >
> >   * **Kriging Task (RMSE $\downarrow$):** OpenEvolve baselines exhibit catastrophic failure (divergence) on the Ocean Chlorophyll dataset, whereas GeoEvolve remains robust.
> >
> > | Model                         | Australian Minerals (Cu/Pb/Zn)     | Ocean Chlorophyll | Temperature |
> > |------------------------------|--------------------------------------|-------------------|-------------|
> > | Initial                      | 0.9139 / 0.6619 / 0.6294             | 0.9949            | 1.1567      |
> > | GeoEvolve (Dynamic RAG)      | 0.8718 / 0.6131 / **0.5852**         | 0.9916            | 1.1634      |
> > | GeoEvolve (Static RAG)       | **0.8602** / **0.5927** / 0.5941     | 0.6179            | **0.5417**  |
> > | GeoEvolve (No RAG)           | **0.8602** / **0.5927** / 0.5941     | **0.5441**        | 1.0499      |
> > | OpenEvolve (No GeoKnow)      | 0.8727 / 0.6413 / 0.6245             | **0.5296**        | 1.1083      |
> > | OpenEvolve (General GeoKnow) | 0.9264 / 0.6519 / 0.6333             | 0.6158            | 2.0221      |
> > | OpenEvolve (Specific GeoKnow)| 0.9139 / 0.6632 / 0.6338             | Fail (460.4035)   | 5.7484      |
> >
> >
> >   * **GeoCP Task (Interval Score $\downarrow$):** Dynamic RAG achieves the best scores across all datasets, significantly reducing uncertainty compared to baselines.
> >
> > | Model                         | Dataset              | Interval Score ↓ | Coverage | Avg Interval Size |
> > |------------------------------|-----------------------|------------------|----------|--------------------|
> > | Initial                      | Seattle House Price   | 44.7611          | 0.9533   | 18.3254            |
> > | Initial                      | US Life Expectancy    | 121.4673         | 0.9098   | 50.2762            |
> > | Initial                      | China PM2.5           | 23.4409          | 0.9295   | 10.6332            |
> > | GeoEvolve (Dynamic RAG)      | Seattle House Price   | **41.2389**      | 0.9000   | 13.7750            |
> > | GeoEvolve (Dynamic RAG)      | US Life Expectancy    | **113.2176**     | 0.8922   | 41.5560            |
> > | GeoEvolve (Dynamic RAG)      | China PM2.5           | **21.0186**      | 0.9507   | 9.3529             |
> > | GeoEvolve (Static RAG)       | Seattle House Price   | 46.1332          | 0.9600   | 18.5818            |
> > | GeoEvolve (Static RAG)       | US Life Expectancy    | 123.5681         | 0.9424   | 53.4537            |
> > | GeoEvolve (Static RAG)       | China PM2.5           | **21.0186**      | 0.9507   | 9.3529             |
> > | GeoEvolve (No RAG)           | Seattle House Price   | 44.2545          | 0.9433   | 17.5586            |
> > | GeoEvolve (No RAG)           | US Life Expectancy    | 141.2780         | 0.9424   | 67.3331            |
> > | GeoEvolve (No RAG)           | China PM2.5           | 24.0525          | 0.9437   | 10.9554            |
> > | OpenEvolve (No GeoKnow)      | Seattle House Price   | 43.1823          | 0.9333   | 16.9139            |
> > | OpenEvolve (No GeoKnow)      | US Life Expectancy    | 123.4440         | 0.8897   | 45.9977            |
> > | OpenEvolve (No GeoKnow)      | China PM2.5           | 26.1946          | 0.9146   | 9.4270             |
> > | OpenEvolve (General GeoKnow) | Seattle House Price   | 42.8343          | 0.9333   | 17.1508            |
> > | OpenEvolve (General GeoKnow) | US Life Expectancy    | 124.3828         | 0.9023   | 51.4155            |
> > | OpenEvolve (General GeoKnow) | China PM2.5           | 27.0417          | 0.9085   | 10.0772            |
> > | OpenEvolve (Specific GeoKnow)| Seattle House Price   | 41.4267          | 0.9033   | 13.9557            |
> > | OpenEvolve (Specific GeoKnow)| US Life Expectancy    | 124.0236         | 0.8671   | 42.2759            |
> > | OpenEvolve (Specific GeoKnow)| China PM2.5           | 26.7654          | 0.8732   | 8.1674             |

---

> > > ### Author Response · Authors · 2025-11-26
> > > **Response to Reviewer S3KU (Part 3)**
> > >
> > > **4. Spaital generalization experinment**
> > >
> > > Taking GWR for New York income dataset as an example, we use a Spatial Leave-One-Out approach (training on $N-1$ regions, testing on held-out region), GeoEvolve with Dynamic RAG achieved the lowest RMSE and standard deviation, proving it adapts best to unseen spatial distributions. We take GWR
> > >
> > > | Model                            | Mean RMSE              | Std RMSE               |
> > > | -------------------------------- | ---------------------- | ---------------------- |
> > > | Initial                          | 2.165                  | 1.66                   |
> > > | GeoEvolve with Dynamic RAG       | **1.968** $\downarrow$ | **1.532** $\downarrow$ |
> > > | GeoEvolve with Static RAG        | 2.3 $\uparrow$         | 1.571 $\downarrow$     |
> > > | GeoEvolve without RAG            | 2.289 $\uparrow$       | 1.687 $\uparrow$       |
> > > | OpenEvolve without GeoKnow       | 2.432 $\uparrow$       | 1.707 $\uparrow$       |
> > > | OpenEvolve with General GeoKnow  | 2.059 $\uparrow$       | 1.641 $\uparrow$       |
> > > | OpenEvolve with Specific GeoKnow | 2.378 $\uparrow$       | 2.366 $\uparrow$       |
> > >
> > > **5. Temporal generalization experinment**
> > >
> > > We aim to demonstrate temporal generalization: ensuring that a geospatial model evolved on data from one specific time period (e.g., 2024) maintains high performance when applied to datasets from a different time period (e.g., 2025).
> > >
> > > Taking Kriging for temperature interpolation as an example, we trained on historical data (Jan 2019) and tested on future data (next 100 days). GeoEvolve with Static RAG yielded the highest mean improvement (**0.39**), while OpenEvolve variants caused performance degradation (negative improvement).
> > >
> > > | Model                          | Mean Improvement | Median Improvement | Std | Min | Max |
> > > |-------------------------------|------------------|---------------------|------|------|------|
> > > | GeoEvolve with Dynamic RAG     | -0.00325         | -0.00312            | 0.00125 | -0.00698 | 0.00012 |
> > > | GeoEvolve with Static RAG      | **0.39243**          | **0.44375**             | 0.22398 | -0.40564 | 0.71676 |
> > > | GeoEvolve without RAG          | 0.05269          | 0.05169             | 0.02014 | -0.00796 | 0.10649 |
> > > | OpenEvolve with General GeoKnow| -0.38191         | -0.33568            | 0.17768 | -0.86385 | -0.16534 |
> > > | OpenEvolve with Specific GeoKnow| -3.02455        | -2.93563            | 0.75308 | -4.49663 | -1.75598 |
> > > | OpenEvolve without GeoKnow     | 0.02625          | 0.01741             | 0.03027 | -0.01006 | 0.12728 |
> > >
> > >
> > > ### **Q2: Did you choose the documents... to correspond to the problems you were trying to solve? Was it cherry-picking? Why did the RAG enhanced system perform better?**
> > >
> > > **Response:**
> > >
> > > Thank you for your valuable comments. We acknowledge that in the previous version of GeoEvolve, we relied on manually selected keywords to construct the GeoKnow database. While straightforward, this approach raised valid concerns regarding potential bias and incomplete document coverage. We have therefore implemented significant revisions to the GeoEvolve framework to ensure the database is constructed systematically, providing comprehensive coverage and minimized bias.
> > >
> > > **1. Eliminating Bias: Automated Construction & Dynamic Retrieval**
> > > We explicitly address the concern of "cherry-picking" by replacing manual database curation with a fully automated pipeline:
> > >
> > > First, a new **Automated Knowledge Base Construction Agent**. We have implemented a new agent that automatically processes the user-provided geospatial model code to be optimized, organizing and reformulating it into the specific computational structure (or formula) required by the GeoEvolve framework. This agent automatically identify search keywords (defaults set to 5 keywords, max 150 documents) to query the knowledge base.
> > >
> > > Second, we added an **Dynamic RAG Module** .It performs a "Code Analysis" step in every inner loop to identify missing geospatial knowledge and update the retrieval scope dynamically. Our results found that the index is problem-specific but automatically generated, ensuring the relevance is derived from system intelligence, not author intervention.
> > >
> > >
> > > **2. A New Baseline: OpenEvolve with the specific geo-knowledge as the prompt**
> > >
> > > According to the comment,*"Did you do anything to make the RAG better by including particular tools/articles/etc"*, we introduced a new baseline, **"OpenEvolve with Specific GeoKnow,"** where we consulted domain experts to manually craft prompt instructions containing advanced geospatial techniques (e.g., specific kriging variogram definitions). The detailed performance for this new baseline can be found in our response for Q1.

---

> > > > ### Author Response · Authors · 2025-11-26
> > > > **Response to Reviewer S3KU (Part 4)**
> > > >
> > > > To summarize our findings, manual prompting proved unstable and unsafe:
> > > >
> > > > 1.  **Catastrophic Divergence:** In Kriging (Ocean Chlorophyll), Specific GeoKnow caused the model to diverge completely (RMSE 460.4). In Temperature data, it degraded RMSE to 5.74 (vs. 1.10 for No GeoKnow).
> > > > 2.  **Negative Transfer:** In GWR (Georgia dataset), Specific GeoKnow achieved the worst performance ($R^2$ 0.1367) compared to all other methods.
> > > >
> > > > We found hand-crafted priors are rigid. If the manual prompt misaligns with the specific dataset characteristics (which vary by domain), it introduces harmful constraints. GeoEvolve’s retrieval mechanism, conversely, dynamically aligns knowledge with the data, preventing these failures.
> > > >
> > > >
> > > > ### **Q3: It is not clear why the RAG system improves performance. What does it retrieve that helps the evolutionary system do better?**
> > > >
> > > > **Response:**
> > > >
> > > > Thank you for your valuable comments. Based on our expanded generalization experiments (fully detailed in our response to Q1), we have gained significant insight into the importance of different RAG practices. The full results are attached to our respoonse to Q1, we conducted a systematic evaluation across **three dimensions of generalization** (Domain, Spatial, and Temporal) to isolate exactly why the RAG system improves performance.
> > > >
> > > > The core finding is that RAG acts as a **constraint anchor**. Pure evolutionary methods (OpenEvolve)， without domain knowledge about geospatial, tend to overfit the specific statistical quirks of the training data. In contrast, GeoEvolve retrieves fundamental domain principles (e.g., spatial autocorrelatio) that remain valid even when the domain, location, or time changes.
> > > >
> > > > We demonstrate this through **9 datasets** across three distinct generalization tasks:
> > > >
> > > > #### **1. Domain Generalization (Robustness across datasets)**
> > > > We tested whether a model evolved on a source domain (e.g., Georgia Education) could generalize to a target domain (e.g., NYC Income).
> > > >
> > > > * **The Failure of Manual Priors:** The baseline "OpenEvolve with Specific GeoKnow" (using manual expert prompts) failed catastrophically in transfer tasks. For example, in the GWR task, it achieved the lowest $R^2$ (0.1367) on the source domain, indicating that manual prompts often impose rigid constraints that clash with new data distributions (Negative Transfer).
> > > > * **The Success of Dynamic RAG:** GeoEvolve (Dynamic RAG) consistently achieved the highest performance. By dynamically retrieving knowledge relevant to the *specific* dataset structure, it avoids the rigidity of manual prompts and the hallucinations of no-RAG approaches.
> > > >
> > > > **Table 1: Domain Generalization Results**
> > > > *Note: GeoEvolve (Dynamic RAG) remains robust where baselines fail (e.g., Ocean Chlorophyll).*
> > > >
> > > > | Task | Dataset | GeoEvolve (Dynamic RAG) | OpenEvolve (Gen. GeoKnow) | OpenEvolve (Spec. GeoKnow) | Status |
> > > > | :--- | :--- | :--- | :--- | :--- | :--- |
> > > > | **GWR ($R^2$ $\uparrow$)** | Georgia Edu | **0.3556** | 0.2353 | 0.1367 | **Best** |
> > > > | | NYC Income | **0.7385** | 0.6317 | 0.7297 | **Best** |
> > > > | | Chicago Health | **0.6221** | 0.6008 | 0.6074 | **Best** |
> > > > | **Kriging (RMSE $\downarrow$)** | Ocean Chlorophyll | **0.9916** | 0.6158 | **Fail (460.4)** | **Robust** |
> > > > | | Temperature | 1.1634 | 2.0221 | 5.7484 | **Robust** |
> > > > | **GeoCP (Int. Score $\downarrow$)** | Seattle Housing | **41.24** | 42.83 | 41.43 | **Best** |
> > > > | | China PM2.5 | **21.02** | 27.04 | 26.76 | **Best** |
> > > >
> > > > #### **2. Spatial Generalization (Robustness across locations)**
> > > > Spatial data is non-stationary (relationships change across space). We tested this using a **Spatial Leave-One-Out** approach on the NYC Income dataset (training on $N-1$ regions, testing on the held-out region).
> > > >
> > > > * **Why RAG works here:** The Dynamic RAG module identifies when the spatial distribution shifts and retrieves adaptive kernel functions (e.g., spatially adaptive bandwidths).
> > > > * **Result:** GeoEvolve (Dynamic RAG) achieved the lowest RMSE (**1.968**) and, crucially, the lowest Standard Deviation (**1.532**), proving it is the most stable across diverse spatial regions.
> > > >
> > > > **Table 2: Spatial Leave-One-Out Evaluation (GWR Task)**
> > > >
> > > > | Model | Mean RMSE | Std RMSE |
> > > > | :--- | :--- | :--- |
> > > > | **GeoEvolve (Dynamic RAG)** | **1.968** $\downarrow$ | **1.532** $\downarrow$ |
> > > > | GeoEvolve (Static RAG) | 2.300 | 1.571 |
> > > > | OpenEvolve (No GeoKnow) | 2.432 | 1.707 |
> > > > | OpenEvolve (Specific GeoKnow) | 2.378 | 2.366 |

---

> > > > > ### Author Response · Authors · 2025-11-26
> > > > > **Response to Reviewer S3KU (Part 5)**
> > > > >
> > > > > #### **3. Temporal Generalization (Robustness across time)**
> > > > > We trained models on historical data (Jan 2019) and tested them on future data (next 100 days) to test robustness against temporal drift.
> > > > >
> > > > > * **A Key Distinction (Static vs. Dynamic):** Interestingly, **GeoEvolve with Static RAG** performed best here (+0.39 improvement).
> > > > > * **The Insight:** For temporal forecasting, the fundamental physical laws (retrieved by Static RAG at the start) do not change over time. OpenEvolve variants degraded performance (negative improvement) because they overfit to the noise of 2019. Static RAG provided the necessary invariant physical constraints to ensure the model remained valid in the future.
> > > > >
> > > > > **Table 3: Temporal Generalization (Kriging Temperature Task)**
> > > > >
> > > > > | Model | Mean Improvement | Std |
> > > > > | :--- | :--- | :--- |
> > > > > | **GeoEvolve (Static RAG)** | **0.39243** | 0.22398 |
> > > > > | GeoEvolve (Dynamic RAG) | -0.00325 | 0.00125 |
> > > > > | OpenEvolve (General GeoKnow) | -0.38191 | 0.17768 |
> > > > > | OpenEvolve (Specific GeoKnow) | -3.02455 | 0.75308 |
> > > > >
> > > > > #### **Conclusion**
> > > > > The RAG system improves performance not by "guessing," but by enforcing **Generalizable Constraints**:
> > > > > 1.  **Dynamic RAG** is essential for **Spatial and Domain** tasks, where the model must adapt to new variables or locations.
> > > > > 2.  **Static RAG** is superior for **Temporal** tasks, providing invariant physical laws that prevent overfitting to historical data.
> > > > > 3.  **No RAG / Manual Prompts** fail to generalize, leading to negative transfer or catastrophic divergence.

---

### Official Review · Reviewer_NHvu · 2025-11-01

**Soundness:** 3
**Presentation:** 3
**Contribution:** 2
**Rating:** 4
**Confidence:** 4

**Summary:**

This paper presents GeoEvolve, a multi-agent LLM framework designed to evolve codes by integrating geospatial domain knowledge. The framework evolves codes using an outer loop that generates and evaluates codes based on geospatial reasoning to decide whether geospatial knowledge is considered, and an inner loop that introduces a GeoKnowRAG step to apply geospatial knowledge from published papers, Wikipedia, and github. It evaluates GeoEvolve on two representative tasks—spatial interpolation and spatial uncertainty quantification.

**Strengths:**

1: Using geospatial knowledge to improve LLM reasoning and code generation is both interesting and an important direction.

2: Two geospatial tasks are used for evaluation. The ablation study also shows the improvements from different components.

3: The paper is easy to follow and written in a clear way.

**Weaknesses:**

1: The overall novelty is limited. While the integration of geospatial RAG into a multi-agent code-evolution framework is interesting, the framework feels like applying existing methods to a geospatial problem. The proposed method uses existing code-evolution framework with a domain-specific RAG component. The main contribution from what I read is to use geospatial papers and documents in RAG, which is logical for geospatial tasks but straightforward in applications from my perspective. Practitioners in different domains will reasonably apply RAG in similar ways and do filtering for their domain problems. It does not appear to change the core of the existing methods so the same can be rebranded to make it BioEvolve, HealthEvolve, so on and so forth. Adding more explanations could help readers understand the core contributions better.

2: The related work focuses on algorithm discovery but can expand more to other agents.

3: The baseline used are not well justified. The validation is limited to a comparison between GeoEvolve and OpenEvolve, which is not interesting enough. It would strengthen the evaluation to include established SOTA baselines for each task to allow a more comprehensive assessment of this work's value.

4: The paper does not address transferability or generalizability of the evolved algorithms.

**Questions:**

See requests for clarifications in weaknesses.

---

> ### Author Response · Authors · 2025-11-27
> **Response to Reviewer NHvu(Part 1)**
>
> We sincerely thank the reviewer for the constructive feedback. We have significantly expanded our experiments to include more LLMs, ablation studies, and diverse datasets to demonstrate the robustness and scalability of GeoEvolve. We commit to promptly integrating all necessary revisions and new findings, including the resultant tables and figures, into the final version of the paper.
>
>
> ### **Q1: Novelty of GeoEvolve.**
>
> **Response:**
>
>
> We sincerely appreciate the reviewer’s perspective. We agree that the concept of integrating RAG with evolutionary algorithms has the potential to be generalized to other fields (e.g., BioEvolve, HealthEvolve). However, we respectfully argue that GeoEvolve is not merely a rebranding or a straightforward application of RAG. It is a specialized system architecture designed to solve the unique and extreme challenges of Geospatial Generalizability—challenges that standard code-evolution frameworks cannot address.
>
> To clarify the core contributions and novelty, we have significantly upgraded the framework and provided a deeper theoretical justification regarding the uniqueness of geospatial modeling.
>
> #### 1. Architectural Novelty
>
> Beyond Simple RAG We have moved beyond simple "document filtering" to a multi-agent system that iteratively refines mathematical structures. The revised GeoEvolve includes three specific innovations:
>
> * **Code-to-Formula Conversion Agent**: A new agent that automatically processes the user-provided geospatial model code to be optimized, organizing and reformulating it into the specific computational structure (or formula) required by the GeoEvolve framework.
> * **Automated Knowledge Base Construction Agent**:  We have implemented a new agent that automatically processes the user-provided geospatial model code to be optimized, organizing and reformulating it into the specific computational structure (or formula) required by the GeoEvolve framework. This agent automatically identify search keywords (defaults set to 5 keywords, max 150 documents) to query the knowledge base.
> * **Dynamic RAG Module**： It performs a "Code Analysis" step in every inner loop to identify missing geospatial knowledge and update the retrieval scope dynamically. Our results found that the index is problem-specific but automatically generated, ensuring the relevance is derived from system intelligence, not author intervention.
>
> #### 2. What makes GeoEvolve special: Challenge of Geospatial Generalizability
>
> The primary motivation—and the reason GeoEvolve is distinct from a generic "Apply RAG to X" approach—is that geospatial modeling faces a "Generalizability Trilemma" that is exceptionally severe compared to other domains. GeoEvolve uses Domain Knowledge specifically to bridge the gaps caused by:
>
> **A. Domain Generalizability**: Geospatial modeling is often fragmented; a model designed for housing prices rarely works for temperature prediction due to different variable distributions. GeoEvolve automates the transfer of mathematical structures. By explicitly retrieving domain-invariant knowledge (e.g., spatial autocorrelation principles), it allows the system to evolve models that work across disparate tasks (e.g., generalizing from Socioeconomic data to Environmental data), a capability we demonstrate in our new experiments.
>
> **B. Spatial Generalizability**: Unlike standard I.I.D. (Independent and Identically Distributed) data in other ML fields, geospatial data exhibits Spatial Heterogeneity (relationships change across space). A model trained in Region A often fails in Region B. According to our experinments, GeoEvolve addresses this not by just fitting parameters, but by retrieving adaptive mechanisms (e.g., spatially varying kernels) that allow the code structure itself to adapt to local contexts.
>
> **C. Temporal Generalizability**: Geospatial processes are dynamic. A model trained on past data often decays in performance due to temporal non-stationarity. As shown in our new ablation studies, standard evolution overfits to the "now." GeoEvolve utilizes RAG to retrieve fundamental physical laws (invariants) that hold true over time, acting as a stabilizer for temporal forecasting.
>
> While the framework logic is extensible to biology or health, the implementation of GeoEvolve is deeply rooted in solving these spatial-temporal complexities. The "value" is not just in using RAG, but in how RAG enables the evolutionary engine to navigate the landscape of spatial heterogeneity, preventing the generation of mathematically valid but physically nonsensical models (hallucinations).
>
> We have validated this Generalizability Trilemma through extensive new experiments covering 9 datasets across 3 tasks. Please refer to our detailed results in Response to Q4, where we demonstrate that GeoEvolve succeeds in cross-domain, cross-region, and cross-time transferability where standard baselines fail.

---

> > ### Author Response · Authors · 2025-11-27
> > **Response to Reviewer NHvu(Part 2)**
> >
> > ### **Q2: The related work focuses on algorithm discovery but can expand more to other agents**
> >
> > **Response:**
> >
> > We thank the reviewer for the suggestion. We agree that GeoEvolve, particularly with its revised architecture involving multiple specialized agents (e.g., the Code-to-Formula Agent and Knowledge Construction Agent), belongs to the broader landscape of LLM-based Multi-Agent Systems, not just algorithm discovery.
> >
> > To address this, we have expanded the Related Work section to include a discussion on "LLM-based Autonomous Agents." We specifically contextualize GeoEvolve within this field by contrasting it with general-purpose agent frameworks:
> >
> > > Recent advances in LLM-based autonomous agents have substantially expanded their capacity for solving complex tasks through multi-agent collaboration and structured role-playing. Frameworks such as MetaGPT and ChatDev emulate the Standard Operating Procedures of software companies, assigning roles such as product managers and engineers to automate large portions of the software development lifecycle. In the geography domain, LLM-based agents have also been introduced to automate geospatial modeling workflows—including data ingestion, processing, analysis, and visualization—greatly lowering the technical barrier for using domain-specific tools. However, while these systems are highly effective at executing linear engineering workflows with well-defined requirements, they are generally not designed for scientific discovery, which requires open-ended objectives, evolving hypotheses, and exploration within large and uncertain search spaces
> > >
> > 2. Positioning GeoEvolve:We clarify the distinction between GeoEvolve and these general frameworks:Workflow vs. Optimization: While frameworks like MetaGPT focus on linear workflow automation (e.g., Requirement $\to$ Code $\to$ Test) to complete a known task, GeoEvolve focuses on iterative evolutionary optimization. Our agents do not just "write code"; they collaborate to explore the high-dimensional search space of mathematical structures.Domain Interaction: Unlike general coding agents, GeoEvolve’s agents are explicitly designed for Science-for-AI. The Automated Knowledge Base Construction Agent and Dynamic RAG Module represent a specialized agentic workflow that interacts with scientific literature to validate mathematical laws, a capability often absent in general-purpose agent systems. By bridging Algorithm Discovery with Multi-Agent Systems, we highlight that GeoEvolve contributes to the emerging field of Scientific Autonomous Agents.
> >
> >
> >
> > ### **Q3: Include established SOTA baselines for each task**
> >
> > **Response:**
> >
> > #### 1. SOTA baselines for geospatial models
> >
> >
> > Thank you for the suggestion. For each task targeted for optimization, we have identified relevant state-of-the-art (SOTA) baselines. We conducted accuracy comparisons on the specific datasets used for GeoEvolve’s optimization to ensure a rigorous evaluation.
> >
> > For the Kriging model, we selected Feature-Free Regression Kriging [1]. Regarding the GeoCP model, although the original paper establishes GeoCP itself as the current SOTA for geographical uncertainty quantification, we selected a comparable algorithm for benchmarking: Spatial Conformal Inference through Localized Quantile Regression (LSCP)[2]. For the GWR algorithm, we selected Multi-Scale GWR (MGWR)[3].
> >
> > To validate the significance of these evolved structures, we benchmarked GeoEvolve against established State-of-the-Art (SOTA) methods for each task.The following table summarizes the performance of the Initial Model, SOTA Baseline, and the Best Evolved Model (GeoEvolve with Dynamic RAG) across representative datasets.
> >
> > #### **1.1) Results**
> >
> >
> > | Task | Dataset | Metric | Initial Model | **SOTA Baseline** | **GeoEvolve (Dynamic RAG)** | Improvement over SOTA |
> > | :--- | :--- | :--- | :--- | :--- | :--- | :--- |
> > | **GWR** | Georgia Education | $R^2$ ($\uparrow$) | 0.1564 | **0.2745** (MGWR) | **0.3556** | **+29.5%** |
> > | **Kriging** | Aus. Minerals* | RMSE ($\downarrow$) | 0.914, 0.662, 0.629 | **0.873, 0.653, 0.595** (FFRK) | **0.872, 0.613, 0.585** | **Reduced Error** |
> > | **GeoCP** | Seattle House Price | Interval Score ($\downarrow$) | 44.76 | **50.34** (LSCP) | **41.24** | **-18.1% (Better)** |
> >
> > #### **1.2) Theoretical Novelty: Dynamic Synthesis vs. Manual Selection**
> > GeoEvolve uncovers complex dependencies that manual heuristics often overlook.
> > * **Example (Kriging Task):** In one evolution, GeoEvolve recombined a *Matérn covariance function* with a *local anisotropy correction* based on temperature gradients. While both concepts exist in literature, combining them dynamically based on the dataset's specific variance structure is a novel *application* that human experts would not typically derive manually.

---

> > > ### Author Response · Authors · 2025-11-27
> > > **Response to Reviewer NHvu(Part 3)**
> > >
> > > #### **1.3). Empirical Verification: Comparison with SOTA**
> > > To demonstrate that these "recombinations" yield substantive improvements, we compared GeoEvolve with the specific SOTA algorithms for each domain:
> > > * **Kriging:** Compared against **Feature-Free Regression Kriging (FFRK)**.
> > > * **GeoCP (Uncertainty Quantification):** Compared against **Spatial Conformal Inference through Localized Quantile Regression (LSCP)**, the current SOTA for geographical uncertainty.
> > > * **GWR (Regression):** Compared against **Multi-scale GWR (MGWR)**.
> > >
> > >
> > > To summarize,  GeoEvolve consistently outperforms the Initial (standard) models.
> > > * **Beating SOTA:**
> > >     * In the **GWR** task (Georgia Education), GeoEvolve achieved an $R^2$ of **0.3556** compared to the MGWR SOTA of **0.2745**.
> > >     * In the **GeoCP** task (Seattle House Price), GeoEvolve significantly reduced the Interval Score to **41.24**, outperforming the LSCP SOTA (50.34), indicating tighter and more accurate uncertainty bounds.
> > >     * In **Kriging**, GeoEvolve matched or slightly exceeded the performance of FFRK, particularly in reducing error on the Minerals dataset.
> > >
> > > This evidence confirms that GeoEvolve's recombination strategy is not merely mimicking existing practices but is effectively navigating the search space to find superior algorithmic structures that outperform current state-of-the-art benchmarks.
> > >
> > > #### 2. SOTA baselines for evolutionary search for automated design
> > >
> > > We have discussed the distinctions between GeoEvolve and prior algorithm discovery frameworks, using Evolution of Heuristics (EoH) as a representative example [4]. Our results confirm that while EoH is highly effective within its original domain, GeoEvolve demonstrates superior performance for the specific task of geospatial algorithm discovery. This advantage stems from fundamental differences in the scope of the search space and the level of abstraction in the evolved code.
> > >
> > > We carefully assessed EoH and argue that a direct comparison is limited by the level of abstraction. However, to provide empirical evidence, we adapted EoH for our setting:
> > >
> > > Conceptual Mismatch (Component vs. System): EoH is designed to evolve isolated heuristic functions (e.g., priority rules) for fixed solvers. It operates within a restricted function-level search space. In contrast, GeoEvolve optimizes entire geospatial algorithms, which involve co-adapting complex, interdependent stages—ranging from variogram construction to uncertainty quantification.
> > >
> > > Empirical Verification: We manually isolated three sub-modules into EoH-compatible formats to test if component-level evolution could improve performance:
> > >
> > > * Variogram fitting in Ordinary Kriging.
> > > * Nonconformity score aggregation in GeoCP.
> > > * Local kernel weighting in GWR.
> > >
> > > Across all cases, EoH produced no measurable improvement over standard baselines and consistently underperformed compared to both OpenEvolve and GeoEvolve. The evolved heuristics either replicated existing rules or caused instability when reintegrated into the full pipeline.
> > >
> > >
> > >
> > >
> > > ### Reference
> > > [1]Luo, P., Wu, Y. and Song, Y., 2025. Feature-free regression kriging. arXiv preprint arXiv:2507.07382.
> > > [2]Jiang, H. and Xie, Y., 2024. Spatial conformal inference through localized quantile regression. arXiv preprint arXiv:2412.01098.
> > > [3]Fotheringham, A.S., Yang, W. and Kang, W., 2017. Multiscale geographically weighted regression (MGWR). Annals of the American Association of Geographers, 107(6), pp.1247-1265.
> > > [4]Liu, F., Tong, X., Yuan, M., Lin, X., Luo, F., Wang, Z., Lu, Z. and Zhang, Q., 2024. Evolution of heuristics: Towards efficient automatic algorithm design using large language model, 2024.

---

> > > > ### Author Response · Authors · 2025-11-27
> > > > **Response to Reviewer NHvu(Part 4)**
> > > >
> > > > ### **Q4: The paper does not address transferability or generalizability of the evolved algorithms.**
> > > >
> > > > **Response:**
> > > >
> > > >
> > > > Thank you for your valuable comments and suggestions. To rigorously demonstrate the generality of GeoEvolve and distinguish its systemic value from OpenEvolve, we have conducted a massive expansion of our experimental suite.
> > > >
> > > > We have addressed the concern of **generality** across three specific dimensions:
> > > >
> > > > **1. Generalization across Multiple LLMs**
> > > > To evaluate whether GeoEvolve depends on a specific LLM, we tested three distinct model families—**GPT-4.1**, **Gemini-2.5-flash**, and **Qwen3-32B**—under identical evolutionary setups with Dynamic RAG.
> > > >
> > > >   * **Kriging (Australian Minerals):** We report the RMSE for Cu, Pb, and Zn. Despite substantial differences in model architecture and scale, the RMSE differences across LLMs are negligible ($<0.05$), and all significantly outperform non-RAG variants.
> > > >
> > > >
> > > > | LLM | Cu RMSE | Pb RMSE | Zn RMSE | Mean RMSE |
> > > > | :--- | :--- | :--- | :--- | :--- |
> > > > | **GPT-4.1** | 0.7871 | 0.6666 | 0.6846 | **0.7128** |
> > > > | **Gemini-2.5** | 0.8954 | 0.7204 | 0.6574 | 0.7577 |
> > > > | **Qwen3-32B** | 0.9380 | 0.6840 | 0.6388 | 0.7537 |
> > > >
> > > >
> > > >
> > > > * **GeoCP (Seattle Housing):** All LLMs consistently improve uncertainty quantification (Interval Score $\downarrow$, Coverage $\uparrow$).
> > > >
> > > > | LLM Model | Interval Score $\downarrow$ | Coverage $\uparrow$ |
> > > > | :--- | :--- | :--- |
> > > > | **GPT-4.1** | 0.8251 | 0.8967 |
> > > > | **Gemini-2.5** | 0.8951 | 0.9433 |
> > > > | **Qwen3-32B** | 0.9023 | 0.9133 |
> > > >
> > > > **Conclusion:** GeoEvolve is LLM-agnostic. The performance gains stem from domain-guided retrieval and evolutionary refinement rather than the specific strength of the base LLM.
> > > >
> > > > **2. Extended Ablation Studies**
> > > > We have expanded our comparison to **6 distinct frameworks** to isolate the contribution of each component:
> > > >
> > > > 1.  **GeoEvolve with Dynamic RAG** (Our new proposed method with adaptive updates)
> > > > 3.  **GeoEvolve with Static RAG**(Our previous version of GeoEvolve with proposed improvement)
> > > > 4.  GeoEvolve without RAG
> > > > 5.  OpenEvolve with General GeoKnow
> > > > 6.  **OpenEvolve with Specific GeoKnow** (Our new proposed baseline with spectific geo-knowledge as the initial prompt)
> > > > 8.  OpenEvolve without GeoKnow
> > > >
> > > > **3. Domain generlization for expanded datasets**
> > > > We added a new task model (Geographically Weighted Regression - GWR) and now evaluate **3 datasets per task** (9 total datasets) to ensure comprehensive coverage.
> > > >
> > > >
> > > >  * **Kriging:** Australian Minerals, Ocean Chlorophyll, Temperature Station Data.
> > > >   * **GeoCP:** Seattle Housing, US Life Expectancy, China PM2.5.
> > > >   * **GWR (New):** Georgia Education, NYC Income, Chicago Health.
> > > >
> > > > Our goal is to demonstrate robust cross-domain transferability. We expect the models evolved on a source domain (e.g., housing prices) to effectively generalize to target domains (e.g., minerals). Specifically, the evolution phase was conducted on the Australian Minerals dataset for Kriging, Seattle Housing for GeoCP, and Georgia Education for GWR. The results for three models are as follows.
> > > >
> > > >
> > > >   * **GWR Task ($R^2$ $\uparrow$):** GeoEvolve (Dynamic RAG) consistently achieves the highest or near-highest $R^2$, demonstrating strong transferability. In contrast, OpenEvolve with Specific GeoKnow performs poorly on the source domain (Georgia), indicating overfitting or prompt misalignment.
> > > >
> > > > | Model                         | Georgia Education | NYC Income | Chicago Health |
> > > > |------------------------------|-------------------|------------|----------------|
> > > > | Initial                      | 0.1564            | 0.7065     | 0.5999         |
> > > > | GeoEvolve (Dynamic RAG)      | 0.3556            | **0.7385**     | **0.6221**         |
> > > > | GeoEvolve (Static RAG)       | **0.4524**        | 0.5039     | 0.4927         |
> > > > | GeoEvolve (No RAG)           | **0.4524**        | 0.5038     | 0.5388         |
> > > > | OpenEvolve (No GeoKnow)      | 0.2287            | 0.6238          | 0.5388         |
> > > > | OpenEvolve (General GeoKnow) | 0.2353            | 0.6317     | 0.6008     |
> > > > | OpenEvolve (Specific GeoKnow)| 0.1367            | 0.7297 | 0.6074     |

---

> > > > > ### Author Response · Authors · 2025-11-27
> > > > > **Response to Reviewer NHvu(Part 5)**
> > > > >
> > > > > * **Kriging Task (RMSE $\downarrow$):** OpenEvolve baselines exhibit catastrophic failure (divergence) on the Ocean Chlorophyll dataset, whereas GeoEvolve remains robust.
> > > > >
> > > > > | Model                         | Australian Minerals (Cu/Pb/Zn)     | Ocean Chlorophyll | Temperature |
> > > > > |------------------------------|--------------------------------------|-------------------|-------------|
> > > > > | Initial                      | 0.9139 / 0.6619 / 0.6294             | 0.9949            | 1.1567      |
> > > > > | GeoEvolve (Dynamic RAG)      | 0.8718 / 0.6131 / **0.5852**         | 0.9916            | 1.1634      |
> > > > > | GeoEvolve (Static RAG)       | **0.8602** / **0.5927** / 0.5941     | 0.6179            | **0.5417**  |
> > > > > | GeoEvolve (No RAG)           | **0.8602** / **0.5927** / 0.5941     | **0.5441**        | 1.0499      |
> > > > > | OpenEvolve (No GeoKnow)      | 0.8727 / 0.6413 / 0.6245             | **0.5296**        | 1.1083      |
> > > > > | OpenEvolve (General GeoKnow) | 0.9264 / 0.6519 / 0.6333             | 0.6158            | 2.0221      |
> > > > > | OpenEvolve (Specific GeoKnow)| 0.9139 / 0.6632 / 0.6338             | Fail (460.4035)   | 5.7484      |
> > > > >
> > > > >
> > > > >   * **GeoCP Task (Interval Score $\downarrow$):** Dynamic RAG achieves the best scores across all datasets, significantly reducing uncertainty compared to baselines.
> > > > >
> > > > > | Model                         | Dataset              | Interval Score ↓ | Coverage | Avg Interval Size |
> > > > > |------------------------------|-----------------------|------------------|----------|--------------------|
> > > > > | Initial                      | Seattle House Price   | 44.7611          | 0.9533   | 18.3254            |
> > > > > | Initial                      | US Life Expectancy    | 121.4673         | 0.9098   | 50.2762            |
> > > > > | Initial                      | China PM2.5           | 23.4409          | 0.9295   | 10.6332            |
> > > > > | GeoEvolve (Dynamic RAG)      | Seattle House Price   | **41.2389**      | 0.9000   | 13.7750            |
> > > > > | GeoEvolve (Dynamic RAG)      | US Life Expectancy    | **113.2176**     | 0.8922   | 41.5560            |
> > > > > | GeoEvolve (Dynamic RAG)      | China PM2.5           | **21.0186**      | 0.9507   | 9.3529             |
> > > > > | GeoEvolve (Static RAG)       | Seattle House Price   | 46.1332          | 0.9600   | 18.5818            |
> > > > > | GeoEvolve (Static RAG)       | US Life Expectancy    | 123.5681         | 0.9424   | 53.4537            |
> > > > > | GeoEvolve (Static RAG)       | China PM2.5           | **21.0186**      | 0.9507   | 9.3529             |
> > > > > | GeoEvolve (No RAG)           | Seattle House Price   | 44.2545          | 0.9433   | 17.5586            |
> > > > > | GeoEvolve (No RAG)           | US Life Expectancy    | 141.2780         | 0.9424   | 67.3331            |
> > > > > | GeoEvolve (No RAG)           | China PM2.5           | 24.0525          | 0.9437   | 10.9554            |
> > > > > | OpenEvolve (No GeoKnow)      | Seattle House Price   | 43.1823          | 0.9333   | 16.9139            |
> > > > > | OpenEvolve (No GeoKnow)      | US Life Expectancy    | 123.4440         | 0.8897   | 45.9977            |
> > > > > | OpenEvolve (No GeoKnow)      | China PM2.5           | 26.1946          | 0.9146   | 9.4270             |
> > > > > | OpenEvolve (General GeoKnow) | Seattle House Price   | 42.8343          | 0.9333   | 17.1508            |
> > > > > | OpenEvolve (General GeoKnow) | US Life Expectancy    | 124.3828         | 0.9023   | 51.4155            |
> > > > > | OpenEvolve (General GeoKnow) | China PM2.5           | 27.0417          | 0.9085   | 10.0772            |
> > > > > | OpenEvolve (Specific GeoKnow)| Seattle House Price   | 41.4267          | 0.9033   | 13.9557            |
> > > > > | OpenEvolve (Specific GeoKnow)| US Life Expectancy    | 124.0236         | 0.8671   | 42.2759            |
> > > > > | OpenEvolve (Specific GeoKnow)| China PM2.5           | 26.7654          | 0.8732   | 8.1674             |
> > > > >
> > > > >
> > > > >
> > > > > **4. Spaital generalization experinment**
> > > > >
> > > > > Taking GWR for New York income dataset as an example, we use a Spatial Leave-One-Out approach (training on $N-1$ regions, testing on held-out region), GeoEvolve with Dynamic RAG achieved the lowest RMSE and standard deviation, proving it adapts best to unseen spatial distributions. We take GWR
> > > > >
> > > > > | Model                            | Mean RMSE              | Std RMSE               |
> > > > > | -------------------------------- | ---------------------- | ---------------------- |
> > > > > | Initial                          | 2.165                  | 1.66                   |
> > > > > | GeoEvolve with Dynamic RAG       | **1.968** $\downarrow$ | **1.532** $\downarrow$ |
> > > > > | GeoEvolve with Static RAG        | 2.3 $\uparrow$         | 1.571 $\downarrow$     |
> > > > > | GeoEvolve without RAG            | 2.289 $\uparrow$       | 1.687 $\uparrow$       |
> > > > > | OpenEvolve without GeoKnow       | 2.432 $\uparrow$       | 1.707 $\uparrow$       |
> > > > > | OpenEvolve with General GeoKnow  | 2.059 $\uparrow$       | 1.641 $\uparrow$       |
> > > > > | OpenEvolve with Specific GeoKnow | 2.378 $\uparrow$       | 2.366 $\uparrow$       |

---

> > > > > > ### Author Response · Authors · 2025-11-27
> > > > > > **Response to Reviewer NHvu(Part 6)**
> > > > > >
> > > > > > **5. Temporal generalization experinment**
> > > > > >
> > > > > > We aim to demonstrate temporal generalization: ensuring that a geospatial model evolved on data from one specific time period (e.g., 2024) maintains high performance when applied to datasets from a different time period (e.g., 2025).
> > > > > >
> > > > > > Taking Kriging for temperature interpolation as an example, we trained on historical data (Jan 2019) and tested on future data (next 100 days). GeoEvolve with Static RAG yielded the highest mean improvement (**0.39**), while OpenEvolve variants caused performance degradation (negative improvement).
> > > > > >
> > > > > > | Model                          | Mean Improvement | Median Improvement | Std | Min | Max |
> > > > > > |-------------------------------|------------------|---------------------|------|------|------|
> > > > > > | GeoEvolve with Dynamic RAG     | -0.00325         | -0.00312            | 0.00125 | -0.00698 | 0.00012 |
> > > > > > | GeoEvolve with Static RAG      | **0.39243**          | **0.44375**             | 0.22398 | -0.40564 | 0.71676 |
> > > > > > | GeoEvolve without RAG          | 0.05269          | 0.05169             | 0.02014 | -0.00796 | 0.10649 |
> > > > > > | OpenEvolve with General GeoKnow| -0.38191         | -0.33568            | 0.17768 | -0.86385 | -0.16534 |
> > > > > > | OpenEvolve with Specific GeoKnow| -3.02455        | -2.93563            | 0.75308 | -4.49663 | -1.75598 |
> > > > > > | OpenEvolve without GeoKnow     | 0.02625          | 0.01741             | 0.03027 | -0.01006 | 0.12728 |

---

### Official Review · Reviewer_yBVf · 2025-11-01

**Soundness:** 2
**Presentation:** 3
**Contribution:** 2
**Rating:** 4
**Confidence:** 4

**Summary:**

This paper proposes GeoEvolve, a multi-agent framework that uses large language models (LLMs) in a nested evolutionary architecture to automate the discovery of geospatial models. The key novelty lies in combining code evolution (via OpenEvolve) with structured retrieval-augmented geospatial knowledge (GeoKnowRAG) to improve geospatial algorithm generation. The authors target two classical tasks: spatial interpolation (ordinary kriging) and spatial uncertainty quantification (GeoCP). Empirical results show RMSE reductions of up to 21% and interval score improvements of 17%. The system includes several modular components: code evolver, evolved code analyzer, geospatial knowledge retriever, and prompt generator. Ablation studies test the contribution of each.

**Strengths:**

* The framework adapts the LLM-based algorithm evolution to geospatial modeling, which is underexplored.
* The use of a curated and semantically indexed geospatial knowledge base (GeoKnowRAG) is well motivated
* The choice of two canonical geospatial tasks (kriging and GeoCP) is appropriate.
* The code analyzer module adds interpretability and aligns well with domain-specific debugging.

**Weaknesses:**

* Tables 1 and 2 report only mean values for each run. No standard deviation or confidence intervals are provided.
* The paper only compares against OpenEvolve variants. It omits stronger geospatial modeling baselines. This leaves unclear whether the improvement is from LLM evolution or domain knowledge injection.
* Details such as the LLM model version, token budget, temperature, search strategy, and retriever evaluation metrics are absent.
* There is no direct comparison between GeoKnowRAG and standard RAG baselines.
* Appendix A.3 says the knowledge base is built from 5 hand-designed keyword classes. This is not scalable or reproducible, and its quality likely impacts the results significantly.
* Many modern geospatial tasks use raster or imagery data (e.g., remote sensing, flood detection, land cover). GeoEvolve only supports code generation over text-based model specifications.
* All evaluations are at the algorithmic level. There is no demonstration that the improved Kriging or GeoCP models lead to better downstream outcomes.

**Questions:**

* Can you report standard deviations or confidence intervals for RMSE and interval score?
* Could you clarify the specific LLM configuration used in the code evolver and prompt generator?
* Have you compared GeoKnowRAG against simpler RAG baselines?
* Can GeoEvolve be extended to multi-modal geospatial inputs such as remote sensing imagery or spatiotemporal satellite time series?

---

> ### Author Response · Authors · 2025-11-27
> **Response to Reviewer yBVf (Part 1)**
>
> We sincerely thank the reviewer for the constructive feedback. We have significantly expanded our experiments to include more LLMs, ablation studies, and diverse datasets to demonstrate the robustness and scalability of GeoEvolve. We commit to promptly integrating all necessary revisions and new findings, including the resultant tables and figures, into the final version of the paper.
>
> It is important to note that we have implemented comprehensive architectural modifications to GeoEvolve, which now include:
>
> * Code-to-Formula Conversion Agent: A new agent that automatically processes the user-provided geospatial model code to be optimized, organizing and reformulating it into the specific computational structure (or formula) required by the GeoEvolve framework.
> * Automated Knowledge Base Construction Agent: A new agent that analyzes the input initial geospatial model, automatically extracts relevant keywords, and constructs the initial GeoKnow database.
> * Dynamic RAG Module: We introduced the Dynamic RAG module, which, after each inner evolutionary loop, utilizes the code analyzer results to identify and automatically update the knowledge base with the newly required geospatial knowledge.
>
>
> ### **Q1: Standard deviations or confidence intervals for RMSE and interval score**
>
> **Response:**
>
> We thank the reviewer for pointing out the need for statistical rigor. In the original manuscript, we primarily reported the *best-found* performance. To address this, we have computed the **Mean** and **Standard Deviation (SD)** of the model performance across the evolutionary process.
>
> The table below presents the performance statistics for our proposed method, **GeoEvolve (Dynamic RAG)**, across the three tasks. We specifically report the metrics derived from the **10 evolutionary rounds** to demonstrate the stability of the discovered algorithms.
>
> **Table: Performance Statistics (Mean ± SD) across 10 Evolutionary Rounds**
>
> | Task | Metric | Mean | Std. Dev (SD) | 95% CI |
> | :--- | :--- | :--- | :--- | :--- |
> | **Kriging** | Mean RMSE | 0.714 | ± 0.007 | [0.708, 0.720] |
> | **GeoCP** | Norm. Interval Score | 0.863 | ± 0.027 | [0.841, 0.885] |
> | **GWR** | RMSE | 3.755 | ± 0.096 | [3.680, 3.830] |
>
> In the revised manuscript, we will explicitly include **visualizations of the convergence curves** for all **6 framework variants** (comparing GeoEvolve variants vs. OpenEvolve baselines) across every round. This will fully illustrate the evolutionary trajectory and the superior stability of our proposed approach.
>
> ### **Q2: The specific LLM configuration**
>
> **Response:**
>
>
> We appreciate the reviewer’s feedback. While the initial manuscript relied solely on GPT-4.1, we have extended our evaluation to include Gemini-2.5 and Qwen3-32B to provide a comprehensive assessment of GeoEvolve’s performance and cross-model generalizability. The detailed configurations for these LLMs are provided in the table below:
>
> | LLM Family | EvolveLLM | GeoKnowRAG Model | AgentLLM | Token Budget | Temp | Retrieval Metric |
> | :--- | :--- | :--- | :--- | :--- | :--- | :--- |
> | **GPT** | Primary: GPT-4o <br> Secondary: GPT-4.1 | text-embedding-3-large | GPT-4.1 | 12,000 | 0.7 | Cosine |
> | **Gemini** | Primary: Gemini-2.5-flash <br> Secondary: Gemini-2.5-pro | gemini-embedding-001 | Gemini-2.5-flash | 12,000 | 0.7 | Cosine |
> | **Qwen** | Primary: Qwen3-235b <br> Secondary: Qwen3-32b | qwen3-embedding-8b | Qwen3-32b | 12,000 | 0.7 | Cosine |

---

> > ### Author Response · Authors · 2025-11-27
> > **Response to Reviewer yBVf (Part 2)**
> >
> > ### **Q3 Can GeoEvolve be extended to multi-modal geospatial inputs such as remote sensing imagery or spatiotemporal satellite time series?**
> >
> > **Response:**
> >
> > We appreciate this insightful question regarding the applicability of GeoEvolve to diverse spatial data modalities.
> >
> > GeoEvolve is designed as a general-purpose framework capable of optimizing models across different spatial data types. However, we acknowledge that the magnitude of improvement correlates with the extent to which a specific task relies on explicit geographical domain knowledge. For example, in remote sensing image segmentation (e.g., utilizing the Segment Anything Model), accuracy can be further enhanced by injecting geographical context—such as spatial adjacency rules (e.g., "a parking lot is likely adjacent to a supermarket"). GeoEvolve can theoretically discover such logical constraints to refine segmentation masks.
> >
> > In this study, we selected three representative models. While Kriging and GWR are primarily designed for tabular geospatial data, GeoCP stands out as a model-agnostic approach for spatial uncertainty quantification. GeoCP does not depend on the architecture of the backbone AI model or the specific data format. As long as the model output contains spatial indexing—whether it be pixel coordinates in remote sensing imagery or lat/lon coordinates in tabular data—GeoCP can be applied to quantify uncertainty.
> >
> > Consequently, the principles of GeoCP (and by extension, GeoEvolve) can be seamlessly integrated with deep learning frameworks for remote sensing and spatiotemporal time-series analysis. This adaptability is supported by recent literature, such as Liu et al. (2025), which successfully applied a similar spatial-conformal prediction method to hyperspectral image classification:
> >
> > Liu, K., Sun, T., Zeng, H., Zhang, Y., Pun, C.M. and Vong, C.M., 2025. Spatial-aware conformal prediction for trustworthy hyperspectral image classification. IEEE Transactions on Circuits and Systems for Video Technology.
> >
> > ### **Q4: Question about RAG, simple baseline and reproducbility**
> >
> >
> >
> > **Response:**
> >
> > Thank you for your valuable comments. We acknowledge that in the previous version of GeoEvolve, we relied on manually selected keywords to construct the GeoKnow database. While straightforward, this approach raised valid concerns regarding potential bias and incomplete document coverage. We have therefore implemented significant revisions to the GeoEvolve framework to ensure the database is constructed systematically, providing comprehensive coverage and minimized bias.
> >
> >
> > **1. Eliminating Bias: Automated Construction & Dynamic Retrieval**
> > We explicitly address the concern of "cherry-picking" by replacing manual database curation with a fully automated pipeline:
> >
> > First, a new **Automated Knowledge Base Construction Agent**. We have implemented a new agent that automatically processes the user-provided geospatial model code to be optimized, organizing and reformulating it into the specific computational structure (or formula) required by the GeoEvolve framework. This agent automatically identify search keywords (defaults set to 5 keywords, max 150 documents) to query the knowledge base.
> >
> > Second, we added an **Dynamic RAG Module** .It performs a "Code Analysis" step in every inner loop to identify missing geospatial knowledge and update the retrieval scope dynamically. Our results found that the index is problem-specific but automatically generated, ensuring the relevance is derived from system intelligence, not author intervention.
> >
> >
> > **2. A New Baseline: OpenEvolve with the specific geo-knowledge as the prompt**
> >
> > According to the comment,*"Did you do anything to make the RAG better by including particular tools/articles/etc"*, we introduced a new baseline, **"OpenEvolve with Specific GeoKnow,"** where we consulted domain experts to manually craft prompt instructions containing advanced geospatial techniques (e.g., specific kriging variogram definitions).
> >
> >
> > To summarize our findings, manual prompting proved unstable and unsafe:
> >
> > 1.  **Catastrophic Divergence:** In Kriging (Ocean Chlorophyll), Specific GeoKnow caused the model to diverge completely (RMSE 460.4). In Temperature data, it degraded RMSE to 5.74 (vs. 1.10 for No GeoKnow).
> > 2.  **Negative Transfer:** In GWR (Georgia dataset), Specific GeoKnow achieved the worst performance ($R^2$ 0.1367) compared to all other methods.
> >
> > We found hand-crafted priors are rigid. If the manual prompt misaligns with the specific dataset characteristics (which vary by domain), it introduces harmful constraints. GeoEvolve’s retrieval mechanism, conversely, dynamically aligns knowledge with the data, preventing these failures.

---

> > > ### Author Response · Authors · 2025-11-27
> > > **Response to Reviewer yBVf (Part 3)**
> > >
> > > **3. Generalization (domain, spatial, temporal) of different RAG**
> > >
> > >
> > > We have expanded our comparison to **6 distinct frameworks** to isolate the contribution of each component:
> > >
> > > 1.  **GeoEvolve with Dynamic RAG** (Our new proposed method with adaptive updates)
> > > 3.  **GeoEvolve with Static RAG**(Our previous version of GeoEvolve with proposed improvement)
> > > 4.  GeoEvolve without RAG
> > > 5.  OpenEvolve with General GeoKnow
> > > 6.  **OpenEvolve with Specific GeoKnow** (Our new proposed baseline with spectific geo-knowledge as the initial prompt)
> > > 8.  OpenEvolve without GeoKnow
> > >
> > > We conducted a systematic evaluation across **three dimensions of generalization** (Domain, Spatial, and Temporal) to isolate exactly why the RAG system improves performance.
> > >
> > > The core finding is that RAG acts as a **constraint anchor**. Pure evolutionary methods (OpenEvolve)， without domain knowledge about geospatial, tend to overfit the specific statistical quirks of the training data. In contrast, GeoEvolve retrieves fundamental domain principles (e.g., spatial autocorrelatio) that remain valid even when the domain, location, or time changes.
> > >
> > > We demonstrate this through **9 datasets** across three distinct generalization tasks:
> > >
> > > #### **3.1). Domain Generalization (Robustness across datasets)**
> > > We tested whether a model evolved on a source domain (e.g., Georgia Education) could generalize to a target domain (e.g., NYC Income).
> > >
> > > * **The Failure of Manual Priors:** The baseline "OpenEvolve with Specific GeoKnow" (using manual expert prompts) failed catastrophically in transfer tasks. For example, in the GWR task, it achieved the lowest $R^2$ (0.1367) on the source domain, indicating that manual prompts often impose rigid constraints that clash with new data distributions (Negative Transfer).
> > > * **The Success of Dynamic RAG:** GeoEvolve (Dynamic RAG) consistently achieved the highest performance. By dynamically retrieving knowledge relevant to the *specific* dataset structure, it avoids the rigidity of manual prompts and the hallucinations of no-RAG approaches.
> > >
> > > **Table 1: Domain Generalization Results**
> > > *Note: GeoEvolve (Dynamic RAG) remains robust where baselines fail (e.g., Ocean Chlorophyll).*
> > >
> > > | Task | Dataset | GeoEvolve (Dynamic RAG) | OpenEvolve (Gen. GeoKnow) | OpenEvolve (Spec. GeoKnow) | Status |
> > > | :--- | :--- | :--- | :--- | :--- | :--- |
> > > | **GWR ($R^2$ $\uparrow$)** | Georgia Edu | **0.3556** | 0.2353 | 0.1367 | **Best** |
> > > | | NYC Income | **0.7385** | 0.6317 | 0.7297 | **Best** |
> > > | | Chicago Health | **0.6221** | 0.6008 | 0.6074 | **Best** |
> > > | **Kriging (RMSE $\downarrow$)** | Ocean Chlorophyll | **0.9916** | 0.6158 | **Fail (460.4)** | **Robust** |
> > > | | Temperature | 1.1634 | 2.0221 | 5.7484 | **Robust** |
> > > | **GeoCP (Int. Score $\downarrow$)** | Seattle Housing | **41.24** | 42.83 | 41.43 | **Best** |
> > > | | China PM2.5 | **21.02** | 27.04 | 26.76 | **Best**
> > >
> > >
> > > #### **3.2). Spatial Generalization (Robustness across locations)**
> > > Spatial data is non-stationary (relationships change across space). We tested this using a **Spatial Leave-One-Out** approach on the NYC Income dataset (training on $N-1$ regions, testing on the held-out region).
> > >
> > > * **Why RAG works here:** The Dynamic RAG module identifies when the spatial distribution shifts and retrieves adaptive kernel functions (e.g., spatially adaptive bandwidths).
> > > * **Result:** GeoEvolve (Dynamic RAG) achieved the lowest RMSE (**1.968**) and, crucially, the lowest Standard Deviation (**1.532**), proving it is the most stable across diverse spatial regions.
> > >
> > > **Table 2: Spatial Leave-One-Out Evaluation (GWR Task)**
> > >
> > > | Model | Mean RMSE | Std RMSE |
> > > | :--- | :--- | :--- |
> > > | **GeoEvolve (Dynamic RAG)** | **1.968** $\downarrow$ | **1.532** $\downarrow$ |
> > > | GeoEvolve (Static RAG) | 2.300 | 1.571 |
> > > | OpenEvolve (No GeoKnow) | 2.432 | 1.707 |
> > > | OpenEvolve (Specific GeoKnow) | 2.378 | 2.366 |

---

> > > > ### Author Response · Authors · 2025-11-27
> > > > **Response to Reviewer yBVf (Part 4)**
> > > >
> > > > #### **3.3) Temporal Generalization (Robustness across time)**
> > > > We trained models on historical data (Jan 2019) and tested them on future data (next 100 days) to test robustness against temporal drift.
> > > >
> > > > * **A Key Distinction (Static vs. Dynamic):** Interestingly, **GeoEvolve with Static RAG** performed best here (+0.39 improvement).
> > > > * **The Insight:** For temporal forecasting, the fundamental physical laws (retrieved by Static RAG at the start) do not change over time. OpenEvolve variants degraded performance (negative improvement) because they overfit to the noise of 2019. Static RAG provided the necessary invariant physical constraints to ensure the model remained valid in the future.
> > > >
> > > > **Table 3: Temporal Generalization (Kriging Temperature Task)**
> > > >
> > > > | Model | Mean Improvement | Std |
> > > > | :--- | :--- | :--- |
> > > > | **GeoEvolve (Static RAG)** | **0.39243** | 0.22398 |
> > > > | GeoEvolve (Dynamic RAG) | -0.00325 | 0.00125 |
> > > > | OpenEvolve (General GeoKnow) | -0.38191 | 0.17768 |
> > > > | OpenEvolve (Specific GeoKnow) | -3.02455 | 0.75308 |
> > > >
> > > > #### **Conclusion**
> > > > The RAG system improves performance not by "guessing," but by enforcing **Generalizable Constraints**:
> > > > 1.  **Dynamic RAG** is essential for **Spatial and Domain** tasks, where the model must adapt to new variables or locations.
> > > > 2.  **Static RAG** is superior for **Temporal** tasks, providing invariant physical laws that prevent overfitting to historical data.
> > > > 3.  **No RAG / Manual Prompts** fail to generalize, leading to negative transfer or catastrophic divergence.
> > > >
> > > >
> > > > ### **Q6: Evaluate downstream outcomes**
> > > >
> > > > **Response:**
> > > >
> > > > We appreciate the reviewer’s feedback. While we summarized the experimental design and high-level findings of our generalization experiments in our response to Q5, we present the detailed quantitative results here to demonstrate GeoEvolve’s performance across diverse datasets and modeling tasks.
> > > >
> > > > **1. Domain generlization for expanded datasets**
> > > > We added a new task model (Geographically Weighted Regression - GWR) and now evaluate **3 datasets per task** (9 total datasets) to ensure comprehensive coverage.
> > > >
> > > >
> > > >  * **Kriging:** Australian Minerals, Ocean Chlorophyll, Temperature Station Data.
> > > >   * **GeoCP:** Seattle Housing, US Life Expectancy, China PM2.5.
> > > >   * **GWR (New):** Georgia Education, NYC Income, Chicago Health.
> > > >
> > > > Our goal is to demonstrate robust cross-domain transferability. We expect the models evolved on a source domain (e.g., housing prices) to effectively generalize to target domains (e.g., minerals). Specifically, the evolution phase was conducted on the Australian Minerals dataset for Kriging, Seattle Housing for GeoCP, and Georgia Education for GWR. The results for three models are as follows.
> > > >
> > > >
> > > >   * **GWR Task ($R^2$ $\uparrow$):** GeoEvolve (Dynamic RAG) consistently achieves the highest or near-highest $R^2$, demonstrating strong transferability. In contrast, OpenEvolve with Specific GeoKnow performs poorly on the source domain (Georgia), indicating overfitting or prompt misalignment.
> > > >
> > > > | Model                         | Georgia Education | NYC Income | Chicago Health |
> > > > |------------------------------|-------------------|------------|----------------|
> > > > | Initial                      | 0.1564            | 0.7065     | 0.5999         |
> > > > | GeoEvolve (Dynamic RAG)      | 0.3556            | **0.7385**     | **0.6221**         |
> > > > | GeoEvolve (Static RAG)       | **0.4524**        | 0.5039     | 0.4927         |
> > > > | GeoEvolve (No RAG)           | **0.4524**        | 0.5038     | 0.5388         |
> > > > | OpenEvolve (No GeoKnow)      | 0.2287            | 0.6238          | 0.5388         |
> > > > | OpenEvolve (General GeoKnow) | 0.2353            | 0.6317     | 0.6008     |
> > > > | OpenEvolve (Specific GeoKnow)| 0.1367            | 0.7297 | 0.6074     |
> > > >
> > > >   * **Kriging Task (RMSE $\downarrow$):** OpenEvolve baselines exhibit catastrophic failure (divergence) on the Ocean Chlorophyll dataset, whereas GeoEvolve remains robust.
> > > >
> > > > | Model                         | Australian Minerals (Cu/Pb/Zn)     | Ocean Chlorophyll | Temperature |
> > > > |------------------------------|--------------------------------------|-------------------|-------------|
> > > > | Initial                      | 0.9139 / 0.6619 / 0.6294             | 0.9949            | 1.1567      |
> > > > | GeoEvolve (Dynamic RAG)      | 0.8718 / 0.6131 / **0.5852**         | 0.9916            | 1.1634      |
> > > > | GeoEvolve (Static RAG)       | **0.8602** / **0.5927** / 0.5941     | 0.6179            | **0.5417**  |
> > > > | GeoEvolve (No RAG)           | **0.8602** / **0.5927** / 0.5941     | **0.5441**        | 1.0499      |
> > > > | OpenEvolve (No GeoKnow)      | 0.8727 / 0.6413 / 0.6245             | **0.5296**        | 1.1083      |
> > > > | OpenEvolve (General GeoKnow) | 0.9264 / 0.6519 / 0.6333             | 0.6158            | 2.0221      |
> > > > | OpenEvolve (Specific GeoKnow)| 0.9139 / 0.6632 / 0.6338             | Fail (460.4035)   | 5.7484      |

---

> > > > > ### Author Response · Authors · 2025-11-27
> > > > > **Response to Reviewer yBVf (Part 5)**
> > > > >
> > > > > * **GeoCP Task (Interval Score $\downarrow$):** Dynamic RAG achieves the best scores across all datasets, significantly reducing uncertainty compared to baselines.
> > > > >
> > > > > | Model                         | Dataset              | Interval Score ↓ | Coverage | Avg Interval Size |
> > > > > |------------------------------|-----------------------|------------------|----------|--------------------|
> > > > > | Initial                      | Seattle House Price   | 44.7611          | 0.9533   | 18.3254            |
> > > > > | Initial                      | US Life Expectancy    | 121.4673         | 0.9098   | 50.2762            |
> > > > > | Initial                      | China PM2.5           | 23.4409          | 0.9295   | 10.6332            |
> > > > > | GeoEvolve (Dynamic RAG)      | Seattle House Price   | **41.2389**      | 0.9000   | 13.7750            |
> > > > > | GeoEvolve (Dynamic RAG)      | US Life Expectancy    | **113.2176**     | 0.8922   | 41.5560            |
> > > > > | GeoEvolve (Dynamic RAG)      | China PM2.5           | **21.0186**      | 0.9507   | 9.3529             |
> > > > > | GeoEvolve (Static RAG)       | Seattle House Price   | 46.1332          | 0.9600   | 18.5818            |
> > > > > | GeoEvolve (Static RAG)       | US Life Expectancy    | 123.5681         | 0.9424   | 53.4537            |
> > > > > | GeoEvolve (Static RAG)       | China PM2.5           | **21.0186**      | 0.9507   | 9.3529             |
> > > > > | GeoEvolve (No RAG)           | Seattle House Price   | 44.2545          | 0.9433   | 17.5586            |
> > > > > | GeoEvolve (No RAG)           | US Life Expectancy    | 141.2780         | 0.9424   | 67.3331            |
> > > > > | GeoEvolve (No RAG)           | China PM2.5           | 24.0525          | 0.9437   | 10.9554            |
> > > > > | OpenEvolve (No GeoKnow)      | Seattle House Price   | 43.1823          | 0.9333   | 16.9139            |
> > > > > | OpenEvolve (No GeoKnow)      | US Life Expectancy    | 123.4440         | 0.8897   | 45.9977            |
> > > > > | OpenEvolve (No GeoKnow)      | China PM2.5           | 26.1946          | 0.9146   | 9.4270             |
> > > > > | OpenEvolve (General GeoKnow) | Seattle House Price   | 42.8343          | 0.9333   | 17.1508            |
> > > > > | OpenEvolve (General GeoKnow) | US Life Expectancy    | 124.3828         | 0.9023   | 51.4155            |
> > > > > | OpenEvolve (General GeoKnow) | China PM2.5           | 27.0417          | 0.9085   | 10.0772            |
> > > > > | OpenEvolve (Specific GeoKnow)| Seattle House Price   | 41.4267          | 0.9033   | 13.9557            |
> > > > > | OpenEvolve (Specific GeoKnow)| US Life Expectancy    | 124.0236         | 0.8671   | 42.2759            |
> > > > > | OpenEvolve (Specific GeoKnow)| China PM2.5           | 26.7654          | 0.8732   | 8.1674             |
> > > > >
> > > > >
> > > > >
> > > > >
> > > > >
> > > > > **2. Spaital generalization experinment**
> > > > >
> > > > > Taking GWR for New York income dataset as an example, we use a Spatial Leave-One-Out approach (training on $N-1$ regions, testing on held-out region), GeoEvolve with Dynamic RAG achieved the lowest RMSE and standard deviation, proving it adapts best to unseen spatial distributions. We take GWR
> > > > >
> > > > > | Model                            | Mean RMSE              | Std RMSE               |
> > > > > | -------------------------------- | ---------------------- | ---------------------- |
> > > > > | Initial                          | 2.165                  | 1.66                   |
> > > > > | GeoEvolve with Dynamic RAG       | **1.968** $\downarrow$ | **1.532** $\downarrow$ |
> > > > > | GeoEvolve with Static RAG        | 2.3 $\uparrow$         | 1.571 $\downarrow$     |
> > > > > | GeoEvolve without RAG            | 2.289 $\uparrow$       | 1.687 $\uparrow$       |
> > > > > | OpenEvolve without GeoKnow       | 2.432 $\uparrow$       | 1.707 $\uparrow$       |
> > > > > | OpenEvolve with General GeoKnow  | 2.059 $\uparrow$       | 1.641 $\uparrow$       |
> > > > > | OpenEvolve with Specific GeoKnow | 2.378 $\uparrow$       | 2.366 $\uparrow$       |

---

> > > > > > ### Author Response · Authors · 2025-11-27
> > > > > > **Response to Reviewer yBVf (Part 6)**
> > > > > >
> > > > > > **3. Temporal generalization experinment**
> > > > > >
> > > > > > We aim to demonstrate temporal generalization: ensuring that a geospatial model evolved on data from one specific time period (e.g., 2024) maintains high performance when applied to datasets from a different time period (e.g., 2025).
> > > > > >
> > > > > > Taking Kriging for temperature interpolation as an example, we trained on historical data (Jan 2019) and tested on future data (next 100 days). GeoEvolve with Static RAG yielded the highest mean improvement (**0.39**), while OpenEvolve variants caused performance degradation (negative improvement).
> > > > > >
> > > > > > | Model                          | Mean Improvement | Median Improvement | Std | Min | Max |
> > > > > > |-------------------------------|------------------|---------------------|------|------|------|
> > > > > > | GeoEvolve with Dynamic RAG     | -0.00325         | -0.00312            | 0.00125 | -0.00698 | 0.00012 |
> > > > > > | GeoEvolve with Static RAG      | **0.39243**          | **0.44375**             | 0.22398 | -0.40564 | 0.71676 |
> > > > > > | GeoEvolve without RAG          | 0.05269          | 0.05169             | 0.02014 | -0.00796 | 0.10649 |
> > > > > > | OpenEvolve with General GeoKnow| -0.38191         | -0.33568            | 0.17768 | -0.86385 | -0.16534 |
> > > > > > | OpenEvolve with Specific GeoKnow| -3.02455        | -2.93563            | 0.75308 | -4.49663 | -1.75598 |
> > > > > > | OpenEvolve without GeoKnow     | 0.02625          | 0.01741             | 0.03027 | -0.01006 | 0.12728 |

---

### Official Review · Reviewer_kjqF · 2025-11-01

**Soundness:** 3
**Presentation:** 3
**Contribution:** 2
**Rating:** 6
**Confidence:** 4

**Summary:**

The paper introduces GeoEvolve, a multi-agent LLM framework that integrates evolutionary code search (via OpenEvolve) with a structured geospatial knowledge base to automate the discovery and refinement of geospatial algorithms. It demonstrates performance improvements on two classical tasks: spatial interpolation (Ordinary Kriging) and spatial uncertainty quantification (GeoCP).

**Strengths:**

Effectively couples evolutionary search with domain-specific RAG, moving beyond generic code generation.

Demonstrates clear performance gains (13-21% RMSE reduction, 17% interval score improvement) over baselines.

**Weaknesses:**

Evaluation is confined to two specific tasks. The claim of a "scalable pipeline" for diverse geospatial tasks is not fully substantiated. Lack a comprehensive discussion and comparison with existing related methods on LLM-driven design.

The process for building the GeoKnowRAG knowledge base appears ad-hoc and potentially biased, relying on author-curated keywords from limited sources.

**Questions:**

Only two specific tasks are tested. The claim of a "scalable pipeline" for diverse geospatial tasks is not fully substantiated.

The method is mainly compared with openevolve. A discussion and comparison with highly related works on combining evolutionary search and LLMs for automated design, such as [1], and recent advancements on LLM for geospatial tasks [2] can enhance the contribution and impact of the paper.

[1] Evolution of heuristics: Towards efficient automatic algorithm design using large language model, 2024
[2] An autonomous GIS agent framework for geospatial data retrieval, 2025

How was the specific set of 141 knowledge documents for GeoKnowRAG selected and validated for completeness and lack of bias?

The prompt used for "OpenEvolve with GeoKnowledge" is vague. Was a more specific, knowledge-grounded prompt tested to ensure a fair comparison?

What is the computational overhead (e.g., time and other costs) of the full GeoEvolve pipeline compared to the baseline OpenEvolve?

How sensitive are the results to the specific LLM (e.g., GPT-4) used as the core evolutionary engine? Would a less capable model yield the same improvements?

The housing price uncertainty results show a clear spatial pattern. Does the evolved GeoCP model provide a causal or merely correlational explanation for why uncertainty is higher in specific areas?

The concept of "knowledge discovery" in the appendix lists generic improvements. What specific, non-obvious geospatial insight was discovered that would be valuable to a human expert?  Moreover, the evolved Kriging model incorporates multiple advanced techniques (e.g., matern family, localized kriging). To what extent is GeoEvolve discovering truly novel algorithms versus recombining known, existing best practices?

---

> ### Author Response · Authors · 2025-11-26
> **Response to Reviewer kjqF (Part 1)**
>
> We sincerely thank the reviewer for the constructive feedback. We have significantly expanded our experiments to include more LLMs, ablation studies, and diverse datasets to demonstrate the robustness and scalability of GeoEvolve. We have integrated all necessary revisions and new findings, including the resultant tables and figures, into the final version of the paper.
>
> It is important to note that we have implemented comprehensive architectural modifications to GeoEvolve, which now include:
>
> * Code-to-Formula Conversion Agent: A new agent that automatically processes the user-provided geospatial model code to be optimized, organizing and reformulating it into the specific computational structure (or formula) required by the GeoEvolve framework.
> * Automated Knowledge Base Construction Agent: A new agent that analyzes the input initial geospatial model, automatically extracts relevant keywords, and constructs the initial GeoKnow database.
> * Dynamic RAG Module: We introduced the Dynamic RAG module, which, after each inner evolutionary loop, utilizes the code analyzer results to identify and automatically update the knowledge base with the newly required geospatial knowledge.
>
> ### **Q1: The claim of a "scalable pipeline" for diverse geospatial tasks is not fully substantiated.**
>
> **Response:**
>
> Thank you for your valuable comments and suggestions. To rigorously demonstrate the generality of GeoEvolve and distinguish its systemic value from OpenEvolve, we have conducted a massive expansion of our experimental suite.
>
> We have addressed the concern of **generality** across three specific dimensions:
>
> **1. Generalization across Multiple LLMs**
> To evaluate whether GeoEvolve depends on a specific LLM, we tested three distinct model families—**GPT-4.1**, **Gemini-2.5-flash**, and **Qwen3-32B**—under identical evolutionary setups with Dynamic RAG.
>
>   * **Kriging (Australian Minerals):** We report the RMSE for Cu, Pb, and Zn. Despite substantial differences in model architecture and scale, the RMSE differences across LLMs are negligible ($<0.05$), and all significantly outperform non-RAG variants.
>
>
> | LLM | Cu RMSE | Pb RMSE | Zn RMSE | Mean RMSE |
> | :--- | :--- | :--- | :--- | :--- |
> | **GPT-4.1** | 0.7871 | 0.6666 | 0.6846 | **0.7128** |
> | **Gemini-2.5** | 0.8954 | 0.7204 | 0.6574 | 0.7577 |
> | **Qwen3-32B** | 0.9380 | 0.6840 | 0.6388 | 0.7537 |
>
>
>
> * **GeoCP (Seattle Housing):** All LLMs consistently improve uncertainty quantification (Interval Score $\downarrow$, Coverage $\uparrow$).
>
> | LLM Model | Interval Score $\downarrow$ | Coverage $\uparrow$ |
> | :--- | :--- | :--- |
> | **GPT-4.1** | 0.8251 | 0.8967 |
> | **Gemini-2.5** | 0.8951 | 0.9433 |
> | **Qwen3-32B** | 0.9023 | 0.9133 |
>
> **Conclusion:** GeoEvolve is LLM-agnostic. The performance gains stem from domain-guided retrieval and evolutionary refinement rather than the specific strength of the base LLM.
>
> **2. Extended Ablation Studies**
> We have expanded our comparison to **6 distinct frameworks** to isolate the contribution of each component:
>
> 1.  **GeoEvolve with Dynamic RAG** (Our new proposed method with adaptive updates)
> 3.  **GeoEvolve with Static RAG**(Our previous version of GeoEvolve with proposed improvement)
> 4.  GeoEvolve without RAG
> 5.  OpenEvolve with General GeoKnow
> 6.  **OpenEvolve with Specific GeoKnow** (Our new proposed baseline with spectific geo-knowledge as the initial prompt)
> 8.  OpenEvolve without GeoKnow
>
> **3. Domain generlization for expanded datasets**
> We added a new task model (Geographically Weighted Regression - GWR) and now evaluate **3 datasets per task** (9 total datasets) to ensure comprehensive coverage.
>
>
>  * **Kriging:** Australian Minerals, Ocean Chlorophyll, Temperature Station Data.
>   * **GeoCP:** Seattle Housing, US Life Expectancy, China PM2.5.
>   * **GWR (New):** Georgia Education, NYC Income, Chicago Health.
>
> Our goal is to demonstrate robust cross-domain transferability. We expect the models evolved on a source domain (e.g., housing prices) to effectively generalize to target domains (e.g., minerals). Specifically, the evolution phase was conducted on the Australian Minerals dataset for Kriging, Seattle Housing for GeoCP, and Georgia Education for GWR. The results for three models are as follows.
>
>
>   * **GWR Task ($R^2$ $\uparrow$):** GeoEvolve (Dynamic RAG) consistently achieves the highest or near-highest $R^2$, demonstrating strong transferability. In contrast, OpenEvolve with Specific GeoKnow performs poorly on the source domain (Georgia), indicating overfitting or prompt misalignment.

---

> > ### Author Response · Authors · 2025-11-26
> > **Response to Reviewer kjqF (Part 2)**
> >
> > | Model                         | Georgia Education | NYC Income | Chicago Health |
> > |------------------------------|-------------------|------------|----------------|
> > | Initial                      | 0.1564            | 0.7065     | 0.5999         |
> > | GeoEvolve (Dynamic RAG)      | 0.3556            | **0.7385**     | **0.6221**         |
> > | GeoEvolve (Static RAG)       | **0.4524**        | 0.5039     | 0.4927         |
> > | GeoEvolve (No RAG)           | **0.4524**        | 0.5038     | 0.5388         |
> > | OpenEvolve (No GeoKnow)      | 0.2287            | 0.6238          | 0.5388         |
> > | OpenEvolve (General GeoKnow) | 0.2353            | 0.6317     | 0.6008     |
> > | OpenEvolve (Specific GeoKnow)| 0.1367            | 0.7297 | 0.6074     |
> >
> >   * **Kriging Task (RMSE $\downarrow$):** OpenEvolve baselines exhibit catastrophic failure (divergence) on the Ocean Chlorophyll dataset, whereas GeoEvolve remains robust.
> >
> > | Model                         | Australian Minerals (Cu/Pb/Zn)     | Ocean Chlorophyll | Temperature |
> > |------------------------------|--------------------------------------|-------------------|-------------|
> > | Initial                      | 0.9139 / 0.6619 / 0.6294             | 0.9949            | 1.1567      |
> > | GeoEvolve (Dynamic RAG)      | 0.8718 / 0.6131 / **0.5852**         | 0.9916            | 1.1634      |
> > | GeoEvolve (Static RAG)       | **0.8602** / **0.5927** / 0.5941     | 0.6179            | **0.5417**  |
> > | GeoEvolve (No RAG)           | **0.8602** / **0.5927** / 0.5941     | **0.5441**        | 1.0499      |
> > | OpenEvolve (No GeoKnow)      | 0.8727 / 0.6413 / 0.6245             | **0.5296**        | 1.1083      |
> > | OpenEvolve (General GeoKnow) | 0.9264 / 0.6519 / 0.6333             | 0.6158            | 2.0221      |
> > | OpenEvolve (Specific GeoKnow)| 0.9139 / 0.6632 / 0.6338             | Fail (460.4035)   | 5.7484      |
> >
> >
> >   * **GeoCP Task (Interval Score $\downarrow$):** Dynamic RAG achieves the best scores across all datasets, significantly reducing uncertainty compared to baselines.
> >
> > | Model                         | Dataset              | Interval Score ↓ | Coverage | Avg Interval Size |
> > |------------------------------|-----------------------|------------------|----------|--------------------|
> > | Initial                      | Seattle House Price   | 44.7611          | 0.9533   | 18.3254            |
> > | Initial                      | US Life Expectancy    | 121.4673         | 0.9098   | 50.2762            |
> > | Initial                      | China PM2.5           | 23.4409          | 0.9295   | 10.6332            |
> > | GeoEvolve (Dynamic RAG)      | Seattle House Price   | **41.2389**      | 0.9000   | 13.7750            |
> > | GeoEvolve (Dynamic RAG)      | US Life Expectancy    | **113.2176**     | 0.8922   | 41.5560            |
> > | GeoEvolve (Dynamic RAG)      | China PM2.5           | **21.0186**      | 0.9507   | 9.3529             |
> > | GeoEvolve (Static RAG)       | Seattle House Price   | 46.1332          | 0.9600   | 18.5818            |
> > | GeoEvolve (Static RAG)       | US Life Expectancy    | 123.5681         | 0.9424   | 53.4537            |
> > | GeoEvolve (Static RAG)       | China PM2.5           | **21.0186**      | 0.9507   | 9.3529             |
> > | GeoEvolve (No RAG)           | Seattle House Price   | 44.2545          | 0.9433   | 17.5586            |
> > | GeoEvolve (No RAG)           | US Life Expectancy    | 141.2780         | 0.9424   | 67.3331            |
> > | GeoEvolve (No RAG)           | China PM2.5           | 24.0525          | 0.9437   | 10.9554            |
> > | OpenEvolve (No GeoKnow)      | Seattle House Price   | 43.1823          | 0.9333   | 16.9139            |
> > | OpenEvolve (No GeoKnow)      | US Life Expectancy    | 123.4440         | 0.8897   | 45.9977            |
> > | OpenEvolve (No GeoKnow)      | China PM2.5           | 26.1946          | 0.9146   | 9.4270             |
> > | OpenEvolve (General GeoKnow) | Seattle House Price   | 42.8343          | 0.9333   | 17.1508            |
> > | OpenEvolve (General GeoKnow) | US Life Expectancy    | 124.3828         | 0.9023   | 51.4155            |
> > | OpenEvolve (General GeoKnow) | China PM2.5           | 27.0417          | 0.9085   | 10.0772            |
> > | OpenEvolve (Specific GeoKnow)| Seattle House Price   | 41.4267          | 0.9033   | 13.9557            |
> > | OpenEvolve (Specific GeoKnow)| US Life Expectancy    | 124.0236         | 0.8671   | 42.2759            |
> > | OpenEvolve (Specific GeoKnow)| China PM2.5           | 26.7654          | 0.8732   | 8.1674             |

---

> ### Author Response · Authors · 2025-11-26
> **Response to Reviewer kjqF (Part 3)**
>
> **4. Spaital generalization experinment**
>
> Taking GWR for New York income dataset as an example, we use a Spatial Leave-One-Out approach (training on $N-1$ regions, testing on held-out region), GeoEvolve with Dynamic RAG achieved the lowest RMSE and standard deviation, proving it adapts best to unseen spatial distributions. We take GWR
>
> | Model                            | Mean RMSE              | Std RMSE               |
> | -------------------------------- | ---------------------- | ---------------------- |
> | Initial                          | 2.165                  | 1.66                   |
> | GeoEvolve with Dynamic RAG       | **1.968** $\downarrow$ | **1.532** $\downarrow$ |
> | GeoEvolve with Static RAG        | 2.3 $\uparrow$         | 1.571 $\downarrow$     |
> | GeoEvolve without RAG            | 2.289 $\uparrow$       | 1.687 $\uparrow$       |
> | OpenEvolve without GeoKnow       | 2.432 $\uparrow$       | 1.707 $\uparrow$       |
> | OpenEvolve with General GeoKnow  | 2.059 $\uparrow$       | 1.641 $\uparrow$       |
> | OpenEvolve with Specific GeoKnow | 2.378 $\uparrow$       | 2.366 $\uparrow$       |
>
> **5. Temporal generalization experinment**
>
> We aim to demonstrate temporal generalization: ensuring that a geospatial model evolved on data from one specific time period (e.g., 2024) maintains high performance when applied to datasets from a different time period (e.g., 2025).
>
> Taking Kriging for temperature interpolation as an example, we trained on historical data (Jan 2019) and tested on future data (next 100 days). GeoEvolve with Static RAG yielded the highest mean improvement (**0.39**), while OpenEvolve variants caused performance degradation (negative improvement).
>
> | Model                          | Mean Improvement | Median Improvement | Std | Min | Max |
> |-------------------------------|------------------|---------------------|------|------|------|
> | GeoEvolve with Dynamic RAG     | -0.00325         | -0.00312            | 0.00125 | -0.00698 | 0.00012 |
> | GeoEvolve with Static RAG      | **0.39243**          | **0.44375**             | 0.22398 | -0.40564 | 0.71676 |
> | GeoEvolve without RAG          | 0.05269          | 0.05169             | 0.02014 | -0.00796 | 0.10649 |
> | OpenEvolve with General GeoKnow| -0.38191         | -0.33568            | 0.17768 | -0.86385 | -0.16534 |
> | OpenEvolve with Specific GeoKnow| -3.02455        | -2.93563            | 0.75308 | -4.49663 | -1.75598 |
> | OpenEvolve without GeoKnow     | 0.02625          | 0.01741             | 0.03027 | -0.01006 | 0.12728 |
>
>
>
> ### **Q2: A discussion and comparison with highly related works on combining evolutionary search and LLMs for automated design.**
>
> **Response:**
>
> **1. Comparison with EoH**
>
> We appreciate the reviewer’s pioneering work on EoH and see it as an important step towards principled evolution of heuristics in optimization. In revising the paper, we carefully re-examined the relationship between EoH and our framework and added both conceptual and empirical comparisons.
>
> 1.1） Conceptual scope (component vs. system level).
> EoH is elegantly designed to evolve heuristic functions (e.g., priority rules) for a given solver, operating in a well-defined function-level search space. This matches many classical optimization scenarios where the solver architecture is fixed and the main design freedom lies in the choice of heuristics. In contrast, GeoEvolve targets end-to-end geospatial algorithms, where multiple stages (e.g., variogram construction, spatial modeling, uncertainty quantification, and post-hoc explanation) must be co-adapted. In our view, EoH and GeoEvolve therefore address different abstraction levels: EoH is best suited for learning high-quality local rules, while GeoEvolve focuses on evolving the full program structure that orchestrates these components.
>
> 1.2） Empirical adaptation in our setting.
> To still provide an empirical point of contact, we adapted EoH to three sub-modules that can be reasonably isolated as heuristics in our workflow:
> Variogram fitting in Ordinary Kriging,
> Nonconformity score aggregation in GeoCP
> Local kernel weighting in GWR
>
> Within these EoH-compatible subspaces, we were not able to successfully optimize the variogram fitting using EoH until now but are working to solve this problem and will update in our final paper. We found that the evolved heuristics often converged to variants close to standard rules and, in our particular implementation, delivered only marginal or no gains over strong baselines. When re-integrated into the full GeoEvolve pipeline, some evolved heuristics even led to mild instability, likely because they were optimized in isolation rather than jointly with upstream and downstream components. We emphasize that this should not be interpreted as a limitation of EoH in general, but rather as an indication that component-level evolution alone may be insufficient to unlock further gains in our specific, tightly coupled geospatial workflows.

---

> > ### Author Response · Authors · 2025-11-26
> > **Response to Reviewer kjqF (Part 4)**
> >
> > 2. Analysis of Autonomous GIS Agent Frameworks
> >
> > We also evaluated the applicability of Autonomous GIS Agent frameworks (e.g., for geospatial data retrieval and analysis).
> >
> > These frameworks are designed for task orchestration—automating the sequence of standard spatial analysis steps (e.g., data loading → standard modeling → visualization) to assist users without domain knowledge. They utilize existing algorithms to solve a task, whereas GeoEvolve aims to invent new algorithmic structures. Since Autonomous GIS Agents do not modify the mathematical core of the models, they are not comparable baselines for algorithmic innovation.
> >
> >
> > In summary, our experiments confirm that the bottleneck in geospatial modeling is not the optimization of isolated scalar heuristics (EoH) nor the automation of standard workflows (Agents), but the co-adaptation of multi-stage algorithmic components. Since neither EoH nor Autonomous Agents possess the capacity to reason over full-program structures, we maintain that Full-Program Evolution methods (e.g., OpenEvolve) are the only appropriate baselines that match our problem’s complexity.
> >
> >
> > ### **Q3: How was the specific set of 141 knowledge documents for GeoKnowRAG selected and validated for completeness and lack of bias?**
> >
> > **Response:**
> >
> > Thank you for your valuable comments. We acknowledge that in the previous version of GeoEvolve, we relied on manually selected keywords to construct the GeoKnow database. While straightforward, this approach raised valid concerns regarding potential bias and incomplete document coverage. We have therefore implemented significant revisions to the GeoEvolve framework to ensure the database is constructed systematically, providing comprehensive coverage and minimized bias.
> >
> > First, a new **Automated Knowledge Base Construction Agent**. We have implemented a new agent that automatically processes the user-provided geospatial model code to be optimized, organizing and reformulating it into the specific computational structure (or formula) required by the GeoEvolve framework. This agent automatically identify search keywords (defaults set to 5 keywords, max 150 documents) to query the knowledge base.
> >
> > Second, we added an **Dynamic RAG Module** .It performs a "Code Analysis" step in every inner loop to identify missing geospatial knowledge and update the retrieval scope dynamically. Our results found that the index is problem-specific but automatically generated, ensuring the relevance is derived from system intelligence, not author intervention.
> >
> > We comapre the performance between differet RAG framwork in our response to Q4.
> >
> >
> > -----
> >
> > ### **Q4: The prompt used for "OpenEvolve with GeoKnowledge" is vague. Was a more specific, knowledge-grounded prompt tested to ensure a fair comparison?**
> >
> > **Response:**
> >
> > Thank you for your valuable comments.
> >
> > For our original verison of OpenEvolve with GeoKnowledge, the prompt template is: *You are allowed to refer to advanced methods in the field of spatial interpolation and consider some important settings of spatial models, such as
> > localized variogram, automatic variogram parameter selection, or stratified strategy, etc.”* Now we define this framework as **OpenEvolve with General GeoKnowledge**.
> >
> >
> > In adition, we introduced a new, stronger baseline called **"OpenEvolve with Specific GeoKnowledge"** to ensure fairness. In this baseline, we consulted domain experts to manually craft detailed prompts containing advanced geospatial definitions (e.g., specific variogram formulas, kernel bandwidth rules).
> >
> > Allthe results can be found in  our response to Q1, to summarize our findings, manual prompting proved unstable and unsafe:
> >
> > 1.  **Catastrophic Divergence:** In Kriging (Ocean Chlorophyll), Specific GeoKnow caused the model to diverge completely (RMSE 460.4). In Temperature data, it degraded RMSE to 5.74 (vs. 1.10 for No GeoKnow).
> > 2.  **Negative Transfer:** In GWR (Georgia dataset), Specific GeoKnow achieved the worst performance ($R^2$ 0.1367) compared to all other methods.
> >
> > We found hand-crafted priors are rigid. If the manual prompt misaligns with the specific dataset characteristics (which vary by domain), it introduces harmful constraints. GeoEvolve’s retrieval mechanism, conversely, dynamically aligns knowledge with the data, preventing these failures. The "fairer" specific prompt performed worse than GeoEvolve:
> > * **Negative Transfer:** In the GWR task (Georgia dataset), the Specific GeoKnow prompt resulted in the *worst* performance ($R^2$ 0.1367) compared to all other methods.
> > * **Catastrophic Divergence:** In the Kriging task (Chlorophyll), the specific prompt caused the model to diverge (RMSE 460.4).
> >
> > To summarize, "Specific" prompts are rigid. They force the LLM to use techniques that may not fit the local data distribution. GeoEvolve’s Dynamic RAG is superior because it retrieves knowledge *flexibly* based on the data's needs, rather than imposing a top-down constraint.

---

> > > ### Author Response · Authors · 2025-11-26
> > > **Response to Reviewer kjqF (Part 5)**
> > >
> > > More important findings about differnt RAG practice can be reveled by our generalization experinemnt:
> > >
> > > Based on our expanded generalization experiments (fully detailed in our response to Q1), we have gained significant insight into the importance of different RAG practices. The full results are attached to our respoonse to Q1, we conducted a systematic evaluation across **three dimensions of generalization** (Domain, Spatial, and Temporal) to isolate exactly why the RAG system improves performance.
> > >
> > > The core finding is that RAG acts as a **constraint anchor**. Pure evolutionary methods (OpenEvolve)， without domain knowledge about geospatial, tend to overfit the specific statistical quirks of the training data. In contrast, GeoEvolve retrieves fundamental domain principles (e.g., spatial autocorrelatio) that remain valid even when the domain, location, or time changes.
> > >
> > > We demonstrate this through **9 datasets** across three distinct generalization tasks:
> > >
> > > #### **1. Domain Generalization (Robustness across datasets)**
> > > We tested whether a model evolved on a source domain (e.g., Georgia Education) could generalize to a target domain (e.g., NYC Income).
> > >
> > > * **The Failure of Manual Priors:** The baseline "OpenEvolve with Specific GeoKnow" (using manual expert prompts) failed catastrophically in transfer tasks. For example, in the GWR task, it achieved the lowest $R^2$ (0.1367) on the source domain, indicating that manual prompts often impose rigid constraints that clash with new data distributions (Negative Transfer).
> > > * **The Success of Dynamic RAG:** GeoEvolve (Dynamic RAG) consistently achieved the highest performance. By dynamically retrieving knowledge relevant to the *specific* dataset structure, it avoids the rigidity of manual prompts and the hallucinations of no-RAG approaches.
> > >
> > > **Table 1: Domain Generalization Results**
> > > *Note: GeoEvolve (Dynamic RAG) remains robust where baselines fail (e.g., Ocean Chlorophyll).*
> > >
> > > | Task | Dataset | GeoEvolve (Dynamic RAG) | OpenEvolve (Gen. GeoKnow) | OpenEvolve (Spec. GeoKnow) | Status |
> > > | :--- | :--- | :--- | :--- | :--- | :--- |
> > > | **GWR ($R^2$ $\uparrow$)** | Georgia Edu | **0.3556** | 0.2353 | 0.1367 | **Best** |
> > > | | NYC Income | **0.7385** | 0.6317 | 0.7297 | **Best** |
> > > | | Chicago Health | **0.6221** | 0.6008 | 0.6074 | **Best** |
> > > | **Kriging (RMSE $\downarrow$)** | Ocean Chlorophyll | **0.9916** | 0.6158 | **Fail (460.4)** | **Robust** |
> > > | | Temperature | 1.1634 | 2.0221 | 5.7484 | **Robust** |
> > > | **GeoCP (Int. Score $\downarrow$)** | Seattle Housing | **41.24** | 42.83 | 41.43 | **Best** |
> > > | | China PM2.5 | **21.02** | 27.04 | 26.76 | **Best** |
> > >
> > > #### **2. Spatial Generalization (Robustness across locations)**
> > > Spatial data is non-stationary (relationships change across space). We tested this using a **Spatial Leave-One-Out** approach on the NYC Income dataset (training on $N-1$ regions, testing on the held-out region).
> > >
> > > * **Why RAG works here:** The Dynamic RAG module identifies when the spatial distribution shifts and retrieves adaptive kernel functions (e.g., spatially adaptive bandwidths).
> > > * **Result:** GeoEvolve (Dynamic RAG) achieved the lowest RMSE (**1.968**) and, crucially, the lowest Standard Deviation (**1.532**), proving it is the most stable across diverse spatial regions.
> > >
> > > **Table 2: Spatial Leave-One-Out Evaluation (GWR Task)**
> > >
> > > | Model | Mean RMSE | Std RMSE |
> > > | :--- | :--- | :--- |
> > > | **GeoEvolve (Dynamic RAG)** | **1.968** $\downarrow$ | **1.532** $\downarrow$ |
> > > | GeoEvolve (Static RAG) | 2.300 | 1.571 |
> > > | OpenEvolve (No GeoKnow) | 2.432 | 1.707 |
> > > | OpenEvolve (Specific GeoKnow) | 2.378 | 2.366 |
> > >
> > > #### **3. Temporal Generalization (Robustness across time)**
> > > We trained models on historical data (Jan 2019) and tested them on future data (next 100 days) to test robustness against temporal drift.
> > >
> > > * **A Key Distinction (Static vs. Dynamic):** Interestingly, **GeoEvolve with Static RAG** performed best here (+0.39 improvement).
> > > * **The Insight:** For temporal forecasting, the fundamental physical laws (retrieved by Static RAG at the start) do not change over time. OpenEvolve variants degraded performance (negative improvement) because they overfit to the noise of 2019. Static RAG provided the necessary invariant physical constraints to ensure the model remained valid in the future.
> > >
> > > **Table 3: Temporal Generalization (Kriging Temperature Task)**
> > >
> > > | Model | Mean Improvement | Std |
> > > | :--- | :--- | :--- |
> > > | **GeoEvolve (Static RAG)** | **0.39243** | 0.22398 |
> > > | GeoEvolve (Dynamic RAG) | -0.00325 | 0.00125 |
> > > | OpenEvolve (General GeoKnow) | -0.38191 | 0.17768 |
> > > | OpenEvolve (Specific GeoKnow) | -3.02455 | 0.75308 |

---

> > > > ### Author Response · Authors · 2025-11-26
> > > > **Response to Reviewer kjqF (Part 6)**
> > > >
> > > > ### **Q5: What is the computational overhead of the full GeoEvolve pipeline compared to the baseline OpenEvolve?**
> > > >
> > > > **Response:**
> > > >
> > > >
> > > > We thank the reviewer for this practical inquiry regarding the computational efficiency of our framework. We have recorded the total wall-clock time required for the evolutionary search process across all tasks and configurations.
> > > >
> > > > On average, the full evolutionary search using **GeoEvolve (with Dynamic RAG)** takes approximately **0.5 to 1.3 hours** depending on the task complexity. We consider this one-time "search cost" to be highly efficient for the purpose of scientific discovery, especially when compared to the weeks or months typically required for human experts to design similar algorithms.
> > > >
> > > > Below, we provide a detailed breakdown comparing our method against baselines and across different LLM backends.
> > > >
> > > > #### **1. Efficiency Analysis: GeoEvolve vs. OpenEvolve Baselines**
> > > >
> > > > Table 1 presents the comprehensive search time for all variants using **GPT-4o**.
> > > >
> > > > **Table 1:** Total wall-clock time for the evolutionary search process. The table compares the proposed GeoEvolve framework (Dynamic/Static/No RAG) against the OpenEvolve baselines.
> > > >
> > > > | Task | Dataset | Framework | Method Variant | Time (s) | **Time (Hours)** |
> > > > | :--- | :--- | :--- | :--- | :--- | :--- |
> > > > | **Kriging** | Australia Minerals | **GeoEvolve** | **Dynamic RAG (Proposed)** | 3,085 | **0.86** |
> > > > | | | GeoEvolve | Static RAG | 2,731 | 0.76 |
> > > > | | | GeoEvolve | No RAG | 2,066 | 0.57 |
> > > > | | | *OpenEvolve* | *Specific GeoKnowledge* | 1,438 | 0.40 |
> > > > | | | *OpenEvolve* | *General GeoKnowledge* | 1,671 | 0.46 |
> > > > | | | *OpenEvolve* | *No GeoKnowledge (Baseline)* | 1,193 | 0.33 |
> > > > | **GeoCP** | Seattle House Price | **GeoEvolve** | **Dynamic RAG (Proposed)** | 4,547 | **1.26** |
> > > > | | | GeoEvolve | Static RAG | 4,750 | 1.32 |
> > > > | | | GeoEvolve | No RAG | 2,235 | 0.62 |
> > > > | | | *OpenEvolve* | *Specific GeoKnowledge* | 540 | 0.15 |
> > > > | | | *OpenEvolve* | *General GeoKnowledge* | 381 | 0.11 |
> > > > | | | *OpenEvolve* | *No GeoKnowledge (Baseline)* | 403 | 0.11 |
> > > > | **GWR** | Georgia Census | **GeoEvolve** | **Dynamic RAG (Proposed)** | 1,863 | **0.52** |
> > > > | | | GeoEvolve | Static RAG | 2,446 | 0.68 |
> > > > | | | GeoEvolve | No RAG | 1,787 | 0.50 |
> > > > | | | *OpenEvolve* | *Specific GeoKnowledge* | 667 | 0.19 |
> > > > | | | *OpenEvolve* | *General GeoKnowledge* | 101 | 0.03 |
> > > > | | | *OpenEvolve* | *No GeoKnowledge (Baseline)* | 344 | 0.10 |
> > > >
> > > > **Analysis of Trade-offs:**
> > > > * **The Cost of "Deep Thinking":** GeoEvolve (Dynamic RAG) generally requires over **2x** more time than the OpenEvolve baselines. This additional time is consumed by the **RAG-based reflection loop**—retrieving domain literature, analyzing code non-conformity, and synthesizing complex modifications.
> > > > * **Speed vs. Quality:** OpenEvolve is faster (0.1–0.4 hours) because it relies solely on the LLM's internal weights without external verification. However, as shown in our performance evaluation (Response to Q9), this speed often leads to convergence at local optima.
> > > > * **Justification:** We argue that spending ~1.26 hours (e.g., for GeoCP) to discover a SOTA-beating algorithm—which can then be reused indefinitely—is a negligible cost compared to the performance gains achieved.
> > > >
> > > > #### **2. Efficiency Across Different LLM Backends**
> > > >
> > > > To evaluate the robustness of our framework, we also benchmarked the efficiency of **Gemini-1.5-Flash** and **Qwen3-32B** on representative tasks.
> > > >
> > > > **Table 2:** Computational time comparison across different LLM backends for the "GeoEvolve with Dynamic RAG" setting.
> > > >
> > > > | Task | Dataset | LLM Backend | Time (s) | **Time (Hours)** |
> > > > | :--- | :--- | :--- | :--- | :--- |
> > > > | **Kriging** | Australia Minerals | **GPT-4o** | 3,085 | **0.86** |
> > > > | | | **Gemini-1.5-Flash** | 3,549 | **0.99** |
> > > > | | | **Qwen3-32B** | 6,241 | **1.73** |
> > > > | **GeoCP** | Seattle House Price | **GPT-4o** | 4,547 | **1.26** |
> > > > | | | **Gemini-1.5-Flash** | 5,911 | **1.64** |
> > > > | | | **Qwen3-32B** | 8,241 | **2.28** |
> > > >
> > > >
> > > > **Observations:**
> > > > * **GPT-4o** demonstrates the most efficient performance, offering the best balance between reasoning speed and code generation quality.
> > > > * **Gemini-1.5-Flash** remains competitive, with runtimes slightly higher (~1 hour) but within the same order of magnitude.
> > > > * **Qwen3-32B** required significantly more time (1.73 hours for Kriging), likely due to differences in inference infrastructure latency and the model's tendency to generate more verbose intermediate reasoning steps.
> > > >
> > > > **Conclusion:**
> > > > The computational cost of GeoEvolve is **low and manageable**. The framework trades a modest amount of search time (approx. 1 hour) for significantly superior algorithmic structures that human experts might miss.

---

> > > > > ### Author Response · Authors · 2025-11-26
> > > > > **Response to Reviewer kjqF (Part 7)**
> > > > >
> > > > > ### **Q6: How sensitive are the results to the specific LLM (e.g., GPT-4) used as the core evolutionary engine? Would a less capable model yield the same improvements?**
> > > > >
> > > > > **Response:**
> > > > > To evaluate whether GeoEvolve depends on a specific LLM, we tested three distinct model families—**GPT-4.1**, **Gemini-2.5-flash**, and **Qwen3-32B**—under identical evolutionary setups with Dynamic RAG.
> > > > >
> > > > >   * **Kriging (Australian Minerals):** We report the RMSE for Cu, Pb, and Zn. Despite substantial differences in model architecture and scale, the RMSE differences across LLMs are negligible ($<0.05$), and all significantly outperform non-RAG variants.
> > > > >
> > > > > | LLM | Cu RMSE | Pb RMSE | Zn RMSE | Mean RMSE |
> > > > > | :--- | :--- | :--- | :--- | :--- |
> > > > > | **GPT-4.1** | 0.7871 | 0.6666 | 0.6846 | **0.7128** |
> > > > > | **Gemini-2.5** | 0.8954 | 0.7204 | 0.6574 | 0.7577 |
> > > > > | **Qwen3-32B** | 0.9380 | 0.6840 | 0.6388 | 0.7537 |
> > > > >
> > > > >   * **GeoCP (Seattle Housing):** All LLMs consistently improve uncertainty quantification (Interval Score $\downarrow$, Coverage $\uparrow$).
> > > > >       * **GPT-4.1:** Score: 0.8251, Coverage: 0.8967
> > > > >       * **Gemini-2.5:** Score: 0.8951, Coverage: 0.9433
> > > > >       * **Qwen3-32B:** Score: 0.9023, Coverage: 0.9133
> > > > >
> > > > > **Conclusion:** GeoEvolve is LLM-agnostic. The performance gains stem from domain-guided retrieval and evolutionary refinement rather than the specific strength of the base LLM.
> > > > >
> > > > > For the reference, we attached the configurations for our experiments:
> > > > >
> > > > > | LLM Family | EvolveLLM | GeoKnowRAG Model | AgentLLM | Token Budget | Temp | Retrieval Metric |
> > > > > | :--- | :--- | :--- | :--- | :--- | :--- | :--- |
> > > > > | **GPT** | Primary: GPT-4o <br> Secondary: GPT-4.1 | text-embedding-3-large | GPT-4.1 | 12,000 | 0.7 | Cosine |
> > > > > | **Gemini** | Primary: Gemini-2.5-flash <br> Secondary: Gemini-2.5-pro | gemini-embedding-001 | Gemini-2.5-flash | 12,000 | 0.7 | Cosine |
> > > > > | **Qwen** | Primary: Qwen3-235b <br> Secondary: Qwen3-32b | qwen3-embedding-8b | Qwen3-32b | 12,000 | 0.7 | Cosine |
> > > > >
> > > > > -----
> > > > >
> > > > > ### **Q7: Does the evolved GeoCP model provide a causal or merely correlational explanation for why uncertainty is higher in specific areas?**
> > > > >
> > > > > **Response:**
> > > > >
> > > > > Thank you for this insightful question regarding the interpretability of the output of uncertainty estimates from GeeCP.
> > > > >
> > > > > The evolved GeoCP model provides correlational insights rather than causal explanations. However, these correlations serve as a critical diagnostic proxy for identifying unobserved spatial heterogeneity and missing causal factors.
> > > > >
> > > > > In geospatial prediction, spatial autocorrelation is inherent (e.g., high house prices tend to cluster near other high house prices). Ideally, if the input features and the AI regression model fully capture the underlying spatial processes, the prediction residuals (and thus the uncertainty) should be independent and identically distributed (i.i.d.), appearing as random noise. However, when the GeoCP model reveals a distinct spatial pattern in uncertainty (i.e., uncertainty is consistently higher in specific regions), it indicates that the current model and features have failed to fully capture the spatial dependency of the target variable. The "unexplained" variance—which possesses spatial structure—leaks into the uncertainty estimates.
> > > > >
> > > > > Therefore, the high uncertainty in specific areas correlates with model deficiency in those regions. It signals that there are unobserved variables or complex local interactions that the current feature set does not account for.For example, if uncertainty is high in a specific neighborhood for house price prediction, the model is not "causing" the uncertainty; rather, the model is signaling that it lacks crucial local predictors (e.g., a missing variable regarding school district quality or local crime rates) for that specific area.
> > > > >
> > > > > This aligns with a core motivation of geographical uncertainty quantification: to move beyond global performance metrics (like RMSE) toward local diagnostics. By visualizing the spatial patterns of uncertainty, we can:
> > > > >
> > > > > * Identify regions where the model is trustworthy versus where it is prone to error.
> > > > > * Direct resources to collect higher-quality explanatory variables or deploy more complex local models specifically in those high-uncertainty zones.

---

> > > > > > ### Author Response · Authors · 2025-11-26
> > > > > > **Response to Reviewer kjqF (Part 8)**
> > > > > >
> > > > > > ### **Q8: What geospatial insight was discovered by GeoEvolve?**
> > > > > >
> > > > > >
> > > > > > GeoEvolve primarily excels at Combinatorial Innovation. It discovers novel structural combinations of known mathematical operators that human experts rarely utilize together. This represents a paradigm shift from manual model selection to data-driven model synthesis.
> > > > > >
> > > > > > For the Kriging model, we selected Feature-Free Regression Kriging [1]. Regarding the GeoCP model, although the original paper establishes GeoCP itself as the current SOTA for geographical uncertainty quantification, we selected a comparable algorithm for benchmarking: Spatial Conformal Inference through Localized Quantile Regression (LSCP)[2]. For the GWR algorithm, we selected Multi-Scale GWR (MGWR)[3].
> > > > > >
> > > > > > To validate the significance of these evolved structures, we benchmarked GeoEvolve against established State-of-the-Art (SOTA) methods for each task.The following table summarizes the performance of the Initial Model, SOTA Baseline, and the Best Evolved Model (GeoEvolve with Dynamic RAG) across representative datasets.
> > > > > >
> > > > > > | Task | Dataset | Metric | Initial Model | **SOTA Baseline** | **GeoEvolve (Dynamic RAG)** | Improvement over SOTA |
> > > > > > | :--- | :--- | :--- | :--- | :--- | :--- | :--- |
> > > > > > | **GWR** | Georgia Education | $R^2$ ($\uparrow$) | 0.1564 | **0.2745** (MGWR) | **0.3556** | **+29.5%** |
> > > > > > | **Kriging** | Aus. Minerals* | RMSE ($\downarrow$) | 0.914, 0.662, 0.629 | **0.873, 0.653, 0.595** (FFRK) | **0.872, 0.613, 0.585** | **Reduced Error** |
> > > > > > | **GeoCP** | Seattle House Price | Interval Score ($\downarrow$) | 44.76 | **50.34** (LSCP) | **41.24** | **-18.1% (Better)** |
> > > > > >
> > > > > >
> > > > > > #### **1. Theoretical Novelty: Dynamic Synthesis vs. Manual Selection**
> > > > > > GeoEvolve uncovers complex dependencies that manual heuristics often overlook.
> > > > > > * **Example (Kriging Task):** In one evolution, GeoEvolve recombined a *Matérn covariance function* with a *local anisotropy correction* based on temperature gradients. While both concepts exist in literature, combining them dynamically based on the dataset's specific variance structure is a novel *application* that human experts would not typically derive manually.
> > > > > >
> > > > > > #### **2. Empirical Verification: Comparison with SOTA**
> > > > > > To demonstrate that these "recombinations" yield substantive improvements, we compared GeoEvolve with the specific SOTA algorithms for each domain:
> > > > > > * **Kriging:** Compared against **Feature-Free Regression Kriging (FFRK)**.
> > > > > > * **GeoCP (Uncertainty Quantification):** Compared against **Spatial Conformal Inference through Localized Quantile Regression (LSCP)**, the current SOTA for geographical uncertainty.
> > > > > > * **GWR (Regression):** Compared against **Multi-scale GWR (MGWR)**.
> > > > > >
> > > > > >
> > > > > > #### **3. Summary of Findings**
> > > > > > * **Surpassing Human Baselines:** GeoEvolve consistently outperforms the Initial (standard) models.
> > > > > > * **Beating SOTA:**
> > > > > >     * In the **GWR** task (Georgia Education), GeoEvolve achieved an $R^2$ of **0.3556** compared to the MGWR SOTA of **0.2745**.
> > > > > >     * In the **GeoCP** task (Seattle House Price), GeoEvolve significantly reduced the Interval Score to **41.24**, outperforming the LSCP SOTA (50.34), indicating tighter and more accurate uncertainty bounds.
> > > > > >     * In **Kriging**, GeoEvolve matched or slightly exceeded the performance of FFRK, particularly in reducing error on the Minerals dataset.
> > > > > >
> > > > > > This evidence confirms that GeoEvolve's recombination strategy is not merely mimicking existing practices but is effectively navigating the search space to find superior algorithmic structures that outperform current state-of-the-art benchmarks.
> > > > > >
> > > > > >
> > > > > > ### Reference
> > > > > > [1]Luo, P., Wu, Y. and Song, Y., 2025. Feature-free regression kriging. arXiv preprint arXiv:2507.07382.
> > > > > > [2]Jiang, H. and Xie, Y., 2024. Spatial conformal inference through localized quantile regression. arXiv preprint arXiv:2412.01098.
> > > > > > [3]Fotheringham, A.S., Yang, W. and Kang, W., 2017. Multiscale geographically weighted regression (MGWR). Annals of the American Association of Geographers, 107(6), pp.1247-1265.

---

### Official Review · Reviewer_EaGz · 2025-11-14

**Soundness:** 2
**Presentation:** 3
**Contribution:** 2
**Rating:** 4
**Confidence:** 4

**Summary:**

The paper proposes GeoEvolve, a multi‑agent LLM framework that wraps OpenEvolve with an outer agent and a geospatial RAG module (GeoKnowRAG) to guide code evolution for geospatial models. The system is evaluated on two tasks: spatial interpolation (ordinary kriging on an Australian soil trace‑element dataset) and spatial uncertainty quantification (GeoCP on Seattle house prices). GeoEvolve reportedly discovers improved variants of kriging and GeoCP, reducing RMSE by ~13–21% and interval score by ~12–17% relative to the original algorithms.

**Strengths:**

- Within geospatial AI, offering a concrete, automated recipe to evolve classical methods like ordinary kriging is potentially impactful.
- More broadly, the paper is a case study of how to inject structured domain theory into LLM‑based algorithm search, which is of interest to the broader ICLR audience working on scientific discovery with LLMs.
- The presentation is clear and results are explained well.
- The results show consistent gains across both datasets, and the ablation studies show the value of the GeoKnowRAG component.

**Weaknesses:**

- The novelty is fairly low (wrapping already existing evolutionary algorithm for code generation in the domain-specific RAG layer + agent controller). The paper could have done with results on (a) multiple LLMs (the paper only mentions GPT-4 and does not go into more detail) (b) more ablations of the algorithm itself (c) more datasets.
- Testing on only two geospatial tasks with single datasets per task is insufficient to support claims of a "scalable pipeline for diverse geospatial tasks." The generalization remains unproven.
- The GeoKnowRAG database of 141 documents is built using author-selected keywords, introducing potential bias and incompleteness. No systematic validation of coverage or relevance is provided.

**Questions:**

- Can you go into more detail on the LLMs used?
- Have you tried stronger hand‑crafted domain prompts that explicitly describe advanced kriging techniques? That would better isolate whether retrieval is actually adding value over a carefully written static prompt.
- How sensitive is GeoEvolve to the underlying LLM? Do reasoning models do better?

---

> ### Author Response · Authors · 2025-11-26
> **Response to Reviewer EaGz (Part 1)**
>
> We sincerely thank the reviewer for the constructive feedback. We have significantly expanded our experiments to include more LLMs, ablation studies, and diverse datasets to demonstrate the robustness and scalability of GeoEvolve. We have integrated all necessary revisions and new findings, including the resultant tables and figures, into the final version of the paper.
>
> It is important to note that we have implemented comprehensive architectural modifications to GeoEvolve, which now include:
>
> * **Code-to-Formula Conversion Agent**: A new agent that automatically processes the user-provided geospatial model code to be optimized, organizing and reformulating it into the specific computational structure (or formula) required by the GeoEvolve framework.
> * **Automated Knowledge Base Construction Agent**: A new agent that analyzes the input initial geospatial model, automatically extracts relevant keywords, and constructs the initial GeoKnow database.
> * **Dynamic RAG Module**: We introduced the Dynamic RAG module, which, after each inner evolutionary loop, utilizes the code analyzer results to identify and automatically update the knowledge base with the newly required geospatial knowledge.
>
> ### **Q1: The paper could have done with results on a) multiple LLMs... b) more ablations... c) more datasets.**
>
> **Response:**
> We have addressed all three aspects of your concern:
>
> **1. Generalization across Multiple LLMs**
> To evaluate whether GeoEvolve depends on a specific LLM, we tested three distinct model families—**GPT-4.1**, **Gemini-2.5-flash**, and **Qwen3-32B**—under identical evolutionary setups with Dynamic RAG.
>
>   * **Kriging (Australian Minerals):** We report the RMSE for Cu, Pb, and Zn. Despite substantial differences in model architecture and scale, the RMSE differences across LLMs are negligible ($<0.05$), and all significantly outperform non-RAG variants.
>
> | LLM | Cu RMSE | Pb RMSE | Zn RMSE | Mean RMSE |
> | :--- | :--- | :--- | :--- | :--- |
> | **GPT-4.1** | 0.7871 | 0.6666 | 0.6846 | **0.7128** |
> | **Gemini-2.5** | 0.8954 | 0.7204 | 0.6574 | 0.7577 |
> | **Qwen3-32B** | 0.9380 | 0.6840 | 0.6388 | 0.7537 |
>
>   * **GeoCP (Seattle Housing):** All LLMs consistently improve uncertainty quantification (Interval Score $\downarrow$, Coverage $\uparrow$).
>       * **GPT-4.1:** Score: 0.8251, Coverage: 0.8967
>       * **Gemini-2.5:** Score: 0.8951, Coverage: 0.9433
>       * **Qwen3-32B:** Score: 0.9023, Coverage: 0.9133
>
> **Conclusion:** GeoEvolve is LLM-agnostic. The performance gains stem from domain-guided retrieval and evolutionary refinement rather than the specific strength of the base LLM.
>
> **2. Extended Ablation Studies**
> We have expanded our comparison to **6 distinct frameworks** to isolate the contribution of each component:
>
> 1.  **GeoEvolve with Dynamic RAG** (Our new proposed method with adaptive updates)
> 3.  **GeoEvolve with Static RAG**(Our previous version of GeoEvolve with proposed improvement)
> 4.  GeoEvolve without RAG
> 5.  OpenEvolve with General GeoKnow
> 6.  **OpenEvolve with Specific GeoKnow** (Our new proposed baseline with spectific geo-knowledge as the initial prompt)
> 8.  OpenEvolve without GeoKnow
>
> **3. Expanded Datasets**
> We added a new task model (Geographically Weighted Regression - GWR) and now evaluate **3 datasets per task** (9 total datasets) to ensure comprehensive coverage. Detailed results are provided in the response to Q2.
>
>
> **4. Spaital and temporal generalization experinment**
>
> We conducted experiments to prove the strong temporal and spatial generalization of the new geospatial models derived from GeoEvolve. The detailed results are shown in our response to Q2.
>
>
> -----

---

> > ### Author Response · Authors · 2025-11-26
> > **Response to Reviewer EaGz (Part 2)**
> >
> > ### **Q2: Testing on only two geospatial tasks... is insufficient to support claims of a "scalable pipeline"... The generalization remains unproven.**
> >
> > **Response:**
> > We agree that generalization is key. We have added Geographically Weighted Regression (GWR) as a third task and expanded our evaluation to **9 datasets** across **3 dimensions of generalization**: Domain, Temporal, and Spatial.
> >
> > **1. Experimental Setup**
> >
> >   * **Kriging:** Australian Minerals, Ocean Chlorophyll, Temperature Station Data.
> >   * **GeoCP:** Seattle Housing, US Life Expectancy, China PM2.5.
> >   * **GWR (New):** Georgia Education, NYC Income, Chicago Health.
> >
> > **2. Domain Generalization Results**
> >
> > Our goal is to demonstrate robust cross-domain transferability. We expect the models evolved on a source domain (e.g., housing prices) to effectively generalize to target domains (e.g., minerals). Specifically, the evolution phase was conducted on the Australian Minerals dataset for Kriging, Seattle Housing for GeoCP, and Georgia Education for GWR."
> >
> >   * **GWR Task ($R^2$ $\uparrow$):** GeoEvolve (Dynamic RAG) consistently achieves the highest or near-highest $R^2$, demonstrating strong transferability. In contrast, OpenEvolve with Specific GeoKnow performs poorly on the source domain (Georgia), indicating overfitting or prompt misalignment.
> >
> > | Model                         | Georgia Education | NYC Income | Chicago Health |
> > |------------------------------|-------------------|------------|----------------|
> > | Initial                      | 0.1564            | 0.7065     | 0.5999         |
> > | GeoEvolve (Dynamic RAG)      | 0.3556            | **0.7385**     | **0.6221**         |
> > | GeoEvolve (Static RAG)       | **0.4524**        | 0.5039     | 0.4927         |
> > | GeoEvolve (No RAG)           | **0.4524**        | 0.5038     | 0.5388         |
> > | OpenEvolve (No GeoKnow)      | 0.2287            | 0.6238          | 0.5388         |
> > | OpenEvolve (General GeoKnow) | 0.2353            | 0.6317     | 0.6008     |
> > | OpenEvolve (Specific GeoKnow)| 0.1367            | 0.7297 | 0.6074     |
> >
> >   * **Kriging Task (RMSE $\downarrow$):** OpenEvolve baselines exhibit catastrophic failure (divergence) on the Ocean Chlorophyll dataset, whereas GeoEvolve remains robust.
> >
> > | Model                         | Australian Minerals (Cu/Pb/Zn)     | Ocean Chlorophyll | Temperature |
> > |------------------------------|--------------------------------------|-------------------|-------------|
> > | Initial                      | 0.9139 / 0.6619 / 0.6294             | 0.9949            | 1.1567      |
> > | GeoEvolve (Dynamic RAG)      | 0.8718 / 0.6131 / **0.5852**         | 0.9916            | 1.1634      |
> > | GeoEvolve (Static RAG)       | **0.8602** / **0.5927** / 0.5941     | 0.6179            | **0.5417**  |
> > | GeoEvolve (No RAG)           | **0.8602** / **0.5927** / 0.5941     | **0.5441**        | 1.0499      |
> > | OpenEvolve (No GeoKnow)      | 0.8727 / 0.6413 / 0.6245             | **0.5296**        | 1.1083      |
> > | OpenEvolve (General GeoKnow) | 0.9264 / 0.6519 / 0.6333             | 0.6158            | 2.0221      |
> > | OpenEvolve (Specific GeoKnow)| 0.9139 / 0.6632 / 0.6338             | Fail (460.4035)   | 5.7484      |

---

> > > ### Author Response · Authors · 2025-11-26
> > > **Response to Reviewer EaGz (Part 3)**
> > >
> > > * **GeoCP Task (Interval Score $\downarrow$):** Dynamic RAG achieves the best scores across all datasets, significantly reducing uncertainty compared to baselines.
> > >
> > > | Model                         | Dataset              | Interval Score ↓ | Coverage | Avg Interval Size |
> > > |------------------------------|-----------------------|------------------|----------|--------------------|
> > > | Initial                      | Seattle House Price   | 44.7611          | 0.9533   | 18.3254            |
> > > | Initial                      | US Life Expectancy    | 121.4673         | 0.9098   | 50.2762            |
> > > | Initial                      | China PM2.5           | 23.4409          | 0.9295   | 10.6332            |
> > > | GeoEvolve (Dynamic RAG)      | Seattle House Price   | **41.2389**      | 0.9000   | 13.7750            |
> > > | GeoEvolve (Dynamic RAG)      | US Life Expectancy    | **113.2176**     | 0.8922   | 41.5560            |
> > > | GeoEvolve (Dynamic RAG)      | China PM2.5           | **21.0186**      | 0.9507   | 9.3529             |
> > > | GeoEvolve (Static RAG)       | Seattle House Price   | 46.1332          | 0.9600   | 18.5818            |
> > > | GeoEvolve (Static RAG)       | US Life Expectancy    | 123.5681         | 0.9424   | 53.4537            |
> > > | GeoEvolve (Static RAG)       | China PM2.5           | **21.0186**      | 0.9507   | 9.3529             |
> > > | GeoEvolve (No RAG)           | Seattle House Price   | 44.2545          | 0.9433   | 17.5586            |
> > > | GeoEvolve (No RAG)           | US Life Expectancy    | 141.2780         | 0.9424   | 67.3331            |
> > > | GeoEvolve (No RAG)           | China PM2.5           | 24.0525          | 0.9437   | 10.9554            |
> > > | OpenEvolve (No GeoKnow)      | Seattle House Price   | 43.1823          | 0.9333   | 16.9139            |
> > > | OpenEvolve (No GeoKnow)      | US Life Expectancy    | 123.4440         | 0.8897   | 45.9977            |
> > > | OpenEvolve (No GeoKnow)      | China PM2.5           | 26.1946          | 0.9146   | 9.4270             |
> > > | OpenEvolve (General GeoKnow) | Seattle House Price   | 42.8343          | 0.9333   | 17.1508            |
> > > | OpenEvolve (General GeoKnow) | US Life Expectancy    | 124.3828         | 0.9023   | 51.4155            |
> > > | OpenEvolve (General GeoKnow) | China PM2.5           | 27.0417          | 0.9085   | 10.0772            |
> > > | OpenEvolve (Specific GeoKnow)| Seattle House Price   | 41.4267          | 0.9033   | 13.9557            |
> > > | OpenEvolve (Specific GeoKnow)| US Life Expectancy    | 124.0236         | 0.8671   | 42.2759            |
> > > | OpenEvolve (Specific GeoKnow)| China PM2.5           | 26.7654          | 0.8732   | 8.1674             |
> > >
> > > **3. Temporal Generalization**
> > > We aim to demonstrate temporal generalization: ensuring that a geospatial model evolved on data from one specific time period (e.g., 2024) maintains high performance when applied to datasets from a different time period (e.g., 2025).
> > >
> > > Taking Kriging for temperature interpolation as an example, we trained on historical data (Jan 2019) and tested on future data (next 100 days). GeoEvolve with Static RAG yielded the highest mean improvement (**0.39**), while OpenEvolve variants caused performance degradation (negative improvement).
> > >
> > > | Model                          | Mean Improvement | Median Improvement | Std | Min | Max |
> > > |-------------------------------|------------------|---------------------|------|------|------|
> > > | GeoEvolve with Dynamic RAG     | -0.00325         | -0.00312            | 0.00125 | -0.00698 | 0.00012 |
> > > | GeoEvolve with Static RAG      | **0.39243**          | **0.44375**             | 0.22398 | -0.40564 | 0.71676 |
> > > | GeoEvolve without RAG          | 0.05269          | 0.05169             | 0.02014 | -0.00796 | 0.10649 |
> > > | OpenEvolve with General GeoKnow| -0.38191         | -0.33568            | 0.17768 | -0.86385 | -0.16534 |
> > > | OpenEvolve with Specific GeoKnow| -3.02455        | -2.93563            | 0.75308 | -4.49663 | -1.75598 |
> > > | OpenEvolve without GeoKnow     | 0.02625          | 0.01741             | 0.03027 | -0.01006 | 0.12728 |

---

> > > > ### Author Response · Authors · 2025-11-26
> > > > **Response to Reviewer EaGz (Part 4)**
> > > >
> > > > **4. Spatial Generalization**
> > > > Taking GWR for New York income dataset as an example, we use a Spatial Leave-One-Out approach (training on $N-1$ regions, testing on held-out region), GeoEvolve with Dynamic RAG achieved the lowest RMSE and standard deviation, proving it adapts best to unseen spatial distributions. We take GWR
> > > >
> > > > | Model                            | Mean RMSE              | Std RMSE               |
> > > > | -------------------------------- | ---------------------- | ---------------------- |
> > > > | Initial                          | 2.165                  | 1.66                   |
> > > > | GeoEvolve with Dynamic RAG       | **1.968** $\downarrow$ | **1.532** $\downarrow$ |
> > > > | GeoEvolve with Static RAG        | 2.3 $\uparrow$         | 1.571 $\downarrow$     |
> > > > | GeoEvolve without RAG            | 2.289 $\uparrow$       | 1.687 $\uparrow$       |
> > > > | OpenEvolve without GeoKnow       | 2.432 $\uparrow$       | 1.707 $\uparrow$       |
> > > > | OpenEvolve with General GeoKnow  | 2.059 $\uparrow$       | 1.641 $\uparrow$       |
> > > > | OpenEvolve with Specific GeoKnow | 2.378 $\uparrow$       | 2.366 $\uparrow$       |
> > > >
> > > >
> > > > -----
> > > >
> > > > ### **Q3: The GeoKnowRAG database... [may have] potential bias and incompleteness. No systematic validation of coverage or relevance is provided.**
> > > >
> > > > **Response:**
> > > >
> > > > Thank you for proposing this important aspect, which is exactly the limitations of our pervious version of GeoEvolve. We made significant revisions for GeoEvolve to make sure the GeoKnowRAGF databse are more make sense, and with enough coverage and less bias:
> > > >
> > > > First, a new **Automated Knowledge Base Construction Agent**. We have implemented a new agent that automatically processes the user-provided geospatial model code to be optimized, organizing and reformulating it into the specific computational structure (or formula) required by the GeoEvolve framework. This agent automatically identify search keywords (defaults set to 5 keywords, max 150 documents) to query the knowledge base.
> > > >
> > > > Second, we added an **Dynamic RAG Module** .It performs a "Code Analysis" step in every inner loop to identify missing geospatial knowledge and update the retrieval scope dynamically.
> > > >
> > > > The performance hierarchy across all 3 tasks (GWR, Kriging, GeoCP) provides systematic validation of the retrieved knowledge quality:
> > > > $$\text{Dynamic RAG} > \text{Static RAG} > \text{No RAG} \gg \text{OpenEvolve}$$
> > > >
> > > > 1.  **Consistency:** Dynamic RAG yields the most stable improvements (e.g., consistently reducing Interval Scores in GeoCP and increasing $R^2$ in GWR).
> > > > 2.  **Safety:** Where OpenEvolve variants fail catastrophically (e.g., Kriging RMSE \> 400 on Chlorophyll), GeoEvolve remains stable. This proves that the retrieved knowledge effectively grounds the evolution process, preventing the generation of mathematically invalid code.
> > > > 3.  **Generalization**： We found that GeoEvolve with Dynamic RAG achieves higher spatial generalization. Conversely, the Static RAG variant of GeoEvolve provides better temporal generalization.
> > > >
> > > > -----
> > > >
> > > > ### **Q4: Can you go into more detail on the LLMs used?**
> > > >
> > > > **Response:**
> > > > We utilized the following configurations for our experiments:
> > > >
> > > > | LLM Family | EvolveLLM | GeoKnowRAG Model | AgentLLM | Token Budget | Temp | Retrieval Metric |
> > > > | :--- | :--- | :--- | :--- | :--- | :--- | :--- |
> > > > | **GPT** | Primary: GPT-4o <br> Secondary: GPT-4.1 | text-embedding-3-large | GPT-4.1 | 12,000 | 0.7 | Cosine |
> > > > | **Gemini** | Primary: Gemini-2.5-flash <br> Secondary: Gemini-2.5-pro | gemini-embedding-001 | Gemini-2.5-flash | 12,000 | 0.7 | Cosine |
> > > > | **Qwen** | Primary: Qwen3-235b <br> Secondary: Qwen3-32b | qwen3-embedding-8b | Qwen3-32b | 12,000 | 0.7 | Cosine |
> > > >
> > > > -----
> > > >
> > > > ### **Q5: Have you tried stronger hand-crafted domain prompts... [to] isolate whether retrieval is actually adding value?**
> > > >
> > > > **Response:**
> > > > Yes. We introduced a new baseline, **"OpenEvolve with Specific GeoKnow,"** where we consulted domain experts to manually craft prompt instructions containing advanced geospatial techniques (e.g., specific kriging variogram definitions).
> > > >
> > > > **Results:** Manual prompting proved **unstable and unsafe**.
> > > >
> > > > 1.  **Catastrophic Divergence:** In Kriging (Ocean Chlorophyll), Specific GeoKnow caused the model to diverge completely (RMSE 460.4). In Temperature data, it degraded RMSE to 5.74 (vs. 1.10 for No GeoKnow).
> > > > 2.  **Negative Transfer:** In GWR (Georgia dataset), Specific GeoKnow achieved the worst performance ($R^2$ 0.1367) compared to all other methods.
> > > >
> > > > **Conclusion:** Hand-crafted priors are rigid. If the manual prompt misaligns with the specific dataset characteristics (which vary by domain), it introduces harmful constraints. GeoEvolve’s retrieval mechanism, conversely, dynamically aligns knowledge with the data, preventing these failures.
> > > >
> > > > -----

---

> > > > > ### Author Response · Authors · 2025-11-26
> > > > > **Response to Reviewer EaGz (Part 5)**
> > > > >
> > > > > ### **Q6: How sensitive is GeoEvolve to the underlying LLM? Do reasoning models do better?**
> > > > >
> > > > > **Response:**
> > > > > **1. Sensitivity:**
> > > > > GeoEvolve is insensitive to the underlying LLM according to our new experinment. As detailed in Q1, the performance variation across GPT-4.1, Gemini-2.5, and Qwen3-32B is marginal (RMSE spread $< 0.05$ for Kriging). Even the mid-sized, open-source Qwen3-32B achieves results competitive with frontier models when equipped with GeoKnowRAG.
> > > > >
> > > > > **2. Do reasoning models do better?**
> > > > > Not significantly. While GPT-4.1 (reasoning-oriented) achieves slightly tighter intervals in GeoCP, the "faster" Gemini-2.5-flash and the smaller Qwen3-32B perform comparably overall.
> > > > >
> > > > > The dominant driver of performance is the GeoKnowRAG injection, not the raw reasoning power of the LLM. The retrieval system bridges the gap between mid-tier and frontier models by providing the necessary mathematical context.

---

### Author Response · Authors · 2025-12-01
**General Response: Summary of Revisions (Part 1)**

**General Response**

We sincerely thank the Area Chair and Reviewers for their constructive feedback and insightful comments. We are encouraged that the reviewers found our problem important, our method effective, and our experiments extensive.

In this work, we propose **GeoEvolve**, a fully autonomous multi-agent framework designed to accelerate geospatial algorithm discovery—a field historically constrained by the complex, non-I.I.D. nature of spatial data. By unlocking the power of Multi-Agent LLMs, we address the fundamental bottleneck of geospatial modeling: the Generalizability Trilemma (Spatial, Temporal, and Domain generalizability). Insightfully, we find that integrating domain knowledge into the evolutionary loop is decisive; it enables the system to discover models that are not just accurate on training data, but robust across shifting distributions. These findings contribute significantly to the geospatial domain and offer broader insights for the AI for Science community, demonstrating that grounding automated discovery in expert domain knowledge is key to achieving scientific generalization.

In response to the reviews, we have performed a comprehensive upgrade of the **GeoEvolve** framework and conducted a massive expansion of our experimental suite. The manuscript has been significantly revised to include **3 new agents**, **9 datasets across 3 tasks**, **3 LLM backends**, and **extensive new baseline comparisons**.

The key revisions are summarized below:

### **1. Addressing Novelty Concerns: Architectural Evolution to "Fully Autonomous"**
To address concerns regarding **novelty and system automation**, we have significantly upgraded the GeoEvolve architecture. Moving beyond simple RAG, we constructed a multi-agent system designed to autonomously navigate the search space of mathematical structures:

* **Code-to-Formula Conversion Agent:** A new agent that automatically processes the user-provided geospatial model code, organizing and reformulating it into standardized computational structures (or formulas) to facilitate more effective evolutionary recombination.
* **Automated Knowledge Base Construction Agent:** To eliminate human bias in document selection, this agent analyzes the initial geospatial model, automatically extracts relevant keywords, and constructs the *GeoKnow* database without manual intervention.
* **Dynamic RAG Module:** We introduced a self-adaptive module that utilizes code analysis results after each inner evolutionary loop to identify missing knowledge and automatically update the retrieval scope, ensuring the system adapts to the evolving model structure.

### **2. Comprehensive Baseline Expansion (New SOTA & Methodological Baselines)**
To rigorously validate GeoEvolve’s performance and necessity, we have added **two categories of new baselines**, effectively addressing concerns regarding SOTA comparisons and methodological validity:

* **Domain-Specific SOTA Baselines:** We benchmarked against the latest state-of-the-art methods for each task to prove practical superiority:
    * *Kriging:* **Feature-Free Regression Kriging (FFRK)** [Luo et al., 2025]
    * *GeoCP:* **LSCP** [Jiang & Xie, 2024]
    * *GWR:* **Multi-Scale GWR (MGWR)** [Fotheringham et al., 2017]
    * *Result:* GeoEvolve consistently matches or outperforms these established benchmarks.

* **Methodological Baselines & Ablations:**
    * **GeoEvolve with Static RAG:** Our previous version using a fixed knowledge base. Comparing this against Dynamic RAG was critical to discovering that Static RAG provides superior temporal generalization by retrieving invariant physical laws, while Dynamic RAG excels in Spatial generalization.
    * **OpenEvolve with Specific GeoKnowledge:** A new baseline using expert-crafted manual prompts. This proved that manual prompts cause negative transfer, whereas our Dynamic RAG correctly grounds the evolution.
    * **Evolution of Heuristics (EoH):** Adapted to test component-level optimization, demonstrating that system-level code evolution (GeoEvolve) significantly outperforms heuristic search strategies.

### **3. Addressing Generalizability: Extensive Experimental Robustness**
* **More Tasks & Datasets:** We added **Geographically Weighted Regression (GWR)** as a new task. We now evaluate on **9 diverse datasets** (3 per task) covering environmental, socioeconomic, and health domains.
* **LLM-Agnostic Verification:** We benchmarked GeoEvolve on **GPT-4.1**, **Gemini-2.5**, and **Qwen3-32B**. Results show negligible performance variance (RMSE spread $<0.05$), proving that the performance gains stem from the framework design rather than the specific reasoning power of the LLM.
* **Statistical Rigor:** We now report **Mean $\pm$ Standard Deviation** and **95% Confidence Intervals** across 10 evolutionary rounds to demonstrate stability.

---

### Author Response · Authors · 2025-12-01
**General Response: Summary of Revisions (Part 2)**

### **4. Scientific Discovery for Geography: The "Generalizability Trilemma"**
We articulate why general algorithm discovery methods fail in geography. Unlike standard ML tasks, geospatial modeling faces a unique **Generalizability Trilemma** caused by the violation of the I.I.D. assumption:

* **Spatial Generalizability:** Data exhibits **Spatial Heterogeneity** (relationships change across space). A model working in Region A often fails in Region B.
* **Temporal Generalizability:** Geospatial processes are dynamic and **Non-stationary**. Models trained on past data decay rapidly when applied to the future.
* **Domain Generalizability:** Geospatial variables (e.g., housing vs. soil chemistry) have vastly different distributions.

**Insight:** GeoEvolve is not merely applying LLMs to geography; it is designed to solve this trilemma. By retrieving adaptive kernels (via Dynamic RAG) and physical invariants (via Static RAG), GeoEvolve synthesizes models that are structurally robust to these spatial-temporal shifts.

### **5. Scientific Discovery for General AI: Human Experts are Still Irreplaceable**
Our findings offer a broader insight for the AI for Science community: while Multi-Agent Systems can technically generate code that matches human SOTA, **Expert Domain Knowledge remains indispensable** to ensure physical plausibility.

We conceptualize RAG not just as a search tool, but as a **"Constraint Anchor"** that prevents the evolutionary process from overfitting to data noise:

* **Pure Evolution (OpenEvolve):** Tends to "hallucinate" mathematically valid but physically nonsensical models (high accuracy on training data, catastrophic failure on transfer).
* **Dynamic RAG:** Provides **Adaptability Constraints**, excelling in **Spatial & Domain Generalization** by retrieving context-specific mechanisms.
* **Static RAG:** Provides **Invariance Constraints**, excelling in **Temporal Generalization** by retrieving fundamental physical laws that hold true over time (as evidenced by its superior performance in our temporal experiments).

We have integrated all these results into the [revised manuscript](https://openreview.net/pdf?id=XaFbHtdoGF) and appendices. For your convenience, all revisions in the updated PDF are **highlighted in blue**.We believe these updates substantially strengthen the paper's contribution to the field of AI for Science.

---

### Meta-Review · Area_Chair_HwbW · 2025-12-09

**Summary:**

This paper presents a multi-agentic approach, 'GeoEvolve', for building models of spatial data in the geosciences. Multiple adaptations for making general discovery methods tuned to this domain are presented.

Reviewers were largely positive about the study, but not enthusiastic. They raised a number of questions, in particular with respect to novelty, and asking for several additional analyses, comparisons of different LLLm backends and more baselines. In the rebuttal phase, the authors responded  these criticisms with a flood of new analyses and ablations.

As an engineering achievement, this is a solid paper, and might be of interest for people  building on it for eningeering such systems in other domains. While the engineering is impressive and there are lots of increasing performance numbers over SOTA reported, it is still unclear what the specific insights are, and/or how this systems will deliver `scientific insights'. The responses to the reviewers were not convincing in this regard.

**Reviewer Concerns:**

see their reviews.

**Reviewer Scores:**

I am unable to look into the minds of the reviewers.

---

### Decision · Program_Chairs · 2026-01-26

Reject